# Microglia regulate central nervous system myelin growth and integrity

Niamh B. McNamara[1,2,3], David A. D. Munro[1,4], Nadine Bestard-Cuche[5], Akiko Uyeda[6], Jeroen F. J. Bogie[7,8], Alana Hoffmann[1,2,3], Rebecca K. Holloway[1,2,3,9,10,11], Irene Molina-Gonzalez[1,2,3], Katharine E. Askew[2], Stephen Mitchell[12], William Mungall[13], Michael Dodds[13], Carsten Dittmayer[14], Jonathan Moss[5], Jamie Rose[1,2], Stefan Szymkowiak[1,2], Lukas Amann[15,16], Barry W. McColl[1,2], Marco Prinz[15,16], Tara L. Spires-Jones[1,2], Werner Stenzel[14], Karen Horsburgh[2], Jerome J. A. Hendriks[7,8], Clare Pridans[17,18,19], Rieko Muramatsu[6], Anna Williams[1,5], Josef Priller[1,4,20,21,22] & Veronique E. Miron[1,2,3,9,10,11,22 ✉]

Myelin is required for the function of neuronal axons in the central nervous system, but the mechanisms that support myelin health are unclear. Although macrophages in the central nervous system have been implicated in myelin health[1], it is unknown which macrophage populations are involved and which aspects they influence. Here we show that resident microglia are crucial for the maintenance of myelin health in adulthood in both mice and humans. We demonstrate that microglia are dispensable for developmental myelin ensheathment. However, they are required for subsequent regulation of myelin growth and associated cognitive function, and for preservation of myelin integrity by preventing its degeneration. We show that loss of myelin health due to the absence of microglia is associated with the appearance of a myelinating oligodendrocyte state with altered lipid metabolism. Moreover, this mechanism is regulated through disruption of the TGFβ1–TGFβR1 axis. Our findings highlight microglia as promising therapeutic targets for conditions in which myelin growth and integrity are dysregulated, such as in ageing and neurodegenerative disease[2,3].

Myelin ensheathes neuronal axons to ensure their health and rapid propagation of electrical impulses to support central nervous system (CNS) functions, for example, cognition. Learning and memory involve the formation of myelin and require myelin to be of good structural integrity. Myelin layers are compacted to a thickness proportional to the axon diameter[4]; however, with ageing and in neurodegenerative disease, disruption of these myelin properties occurs through hypermyelination. Enlarged areas of uncompacted myelin (where myelin grows) leads to myelin that is thicker, unravelling and forming protrusions (termed outfoldings), and loss of myelin integrity through degeneration also occurs in these contexts[2,3,5]. These myelin changes lead to impaired cognition in mice and predict poor cognitive performance in aged nonhuman primates and humans[6–9]. However, the fundamental mechanisms that coordinate appropriate formation, growth and integrity of CNS myelin are unclear. Recent research has implicated a population of CNS-resident macrophages, microglia, in this process.

Myelination and the generation of myelin-forming oligodendrocytes are impaired following microglial depletion through loss of function of the pro-survival colony stimulating factor 1 receptor (CSF1R)[1]. However, this approach also targets CNS-resident border-associated macrophages (including perivascular macrophages) and blood monocytes. Therefore, it is unclear which macrophage populations regulate myelin, and the specific involvement of microglia in myelin formation and health is unknown.

## Microglia are dispensable for myelination

To address these questions, we utilized a recently developed transgenic mouse model in which the Fms intronic regulatory element (*Fire*) super-enhancer of the *Csf1r* gene (*Fire*[Δ/Δ]; Fig. 1a) is deleted. This deletion leads to an absence of microglia from development (when they normally emerge) through to adulthood, whereas other CNS

[1]UK Dementia Research Institute at The University of Edinburgh, Edinburgh, UK. [2]Centre for Discovery Brain Sciences, Chancellor's Building, The University of Edinburgh, Edinburgh, UK. [3]Medical Research Council Centre for Reproductive Health, The Queen's Medical Research Institute, The University of Edinburgh, Edinburgh, UK. [4]Centre for Clinical Brain Sciences, Chancellor's Building, The University of Edinburgh, Edinburgh, UK. [5]Centre for Regenerative Medicine, Institute for Regeneration and Repair, The University of Edinburgh, Edinburgh, UK. [6]Departments of Molecular Pharmacology, National Institute of Neuroscience, National Center of Neurology and Psychiatry, Kodaira, Japan. [7]Department of Immunology and Infection, Biomedical Research Institute, Hasselt University, Hasselt, Belgium. [8]University MS Centre, Hasselt University, Hasselt, Belgium. [9]Barlo Multiple Sclerosis Centre, St Michael's Hospital, Toronto, Ontario, Canada. [10]Keenan Research Centre for Biomedical Science, St Michael's Hospital, Toronto, Ontario, Canada. [11]Department of Immunology, The University of Toronto, Toronto, Ontario, Canada. [12]Wellcome Trust Centre for Cell Biology, King's Buildings, The University of Edinburgh, Edinburgh, UK. [13]Biological and Veterinary Services, Chancellor's Building, The University of Edinburgh, Edinburgh, UK. [14]Department of Neuropathology and Neurocure Clinical Research Center, Charité-Universitätsmedizin Berlin, Berlin, Germany. [15]Institute of Neuropathology, Centre for Basics in NeuroModulation, Faculty of Medicine, University of Freiburg, Freiburg, Germany. [16]Signalling Research Centres BIOSS and CIBSS, University of Freiburg, Freiburg, Germany. [17]Centre for Inflammation Research, The Queen's Medical Research Institute, The University of Edinburgh, Edinburgh, UK. [18]Simons Initiative for the Developing Brain, Centre for Discovery Brain Sciences, University of Edinburgh, Edinburgh, UK. [19]Muir Maxwell Epilepsy Centre, University of Edinburgh, Edinburgh, UK. [20]Department of Psychiatry and Psychotherapy, Klinikum rechts der Isar, School of Medicine, Technical University of Munich, Munich, Germany. [21]Neuropsychiatry and Laboratory of Molecular Psychiatry, Charité-Universitätsmedizin Berlin and DZNE, Berlin, Germany. [22]These authors contributed equally: Josef Priller, Veronique E. Miron. ✉e-mail: Veronique.Miron@unityhealth.to

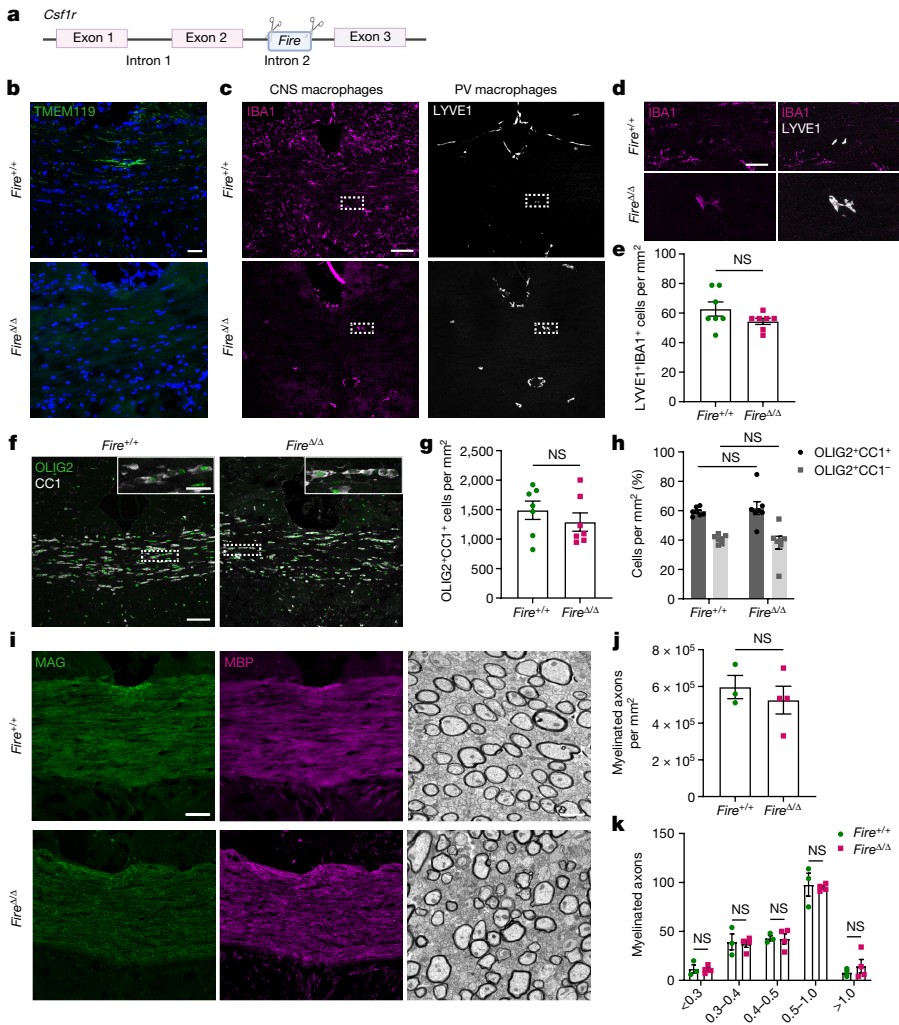

**Fig. 1 | Microglia are not required for oligodendrocyte maturation and myelination. a**, $Fire^{\Delta/\Delta}$ mice were generated using CRISPR9–Cas9 deletion of the $Fire$ super-enhancer located in intron 2 of $Csf1r$. **b**, Images of microglia (TMEM119[+]; green) in corpus callosum samples from $Fire^{+/+}$ and $Fire^{\Delta/\Delta}$ mice at 1 month of age, counterstained with Hoechst (blue). **c**, Images of CNS macrophages (IBA1[+]; magenta) and perivascular (PV) macrophages (LYVE1[+]; white) in $Fire^{+/+}$ and $Fire^{\Delta/\Delta}$ mice. **d**, Magnified images from **c** of IBA1[+]LYVE1[+] PV macrophages in $Fire^{+/+}$ and $Fire^{\Delta/\Delta}$ mice. **e**, Mean LYVE1[+]IBA1[+] cells per mm[2] ± s.e.m. in $Fire^{+/+}$ and $Fire^{\Delta/\Delta}$ mice. $n = 7$ mice per group. $P = 0.1411$, two-tailed unpaired Student's $t$-test. **f**, Images of mature oligodendrocytes expressing both OLIG2 (green) and CC1 (white) in $Fire^{+/+}$ and $Fire^{\Delta/\Delta}$ mice. Inset shows magnified view. **g**, Mean OLIG2[+]CC1[+] cells per mm[2] ± s.e.m. in $Fire^{+/+}$ and $Fire^{\Delta/\Delta}$ mice. $n = 7$ mice per group. $P = 0.1990$, two-tailed unpaired Student's $t$-test.

**h**, Mean proportion of cells of the oligodendrocyte lineage (OLIG2[+]), which are mature (CC1[+]; black) or immature (CC1[−]; grey) (±s.e.m.). $n = 7$ mice per group. CC1[+], $P = 0.9472$; CC1[−], $P = 0.9472$; one-way analysis of variance (ANOVA) with Tukey's multiple comparisons test. **i**, Images of corpus callosum from $Fire^{+/+}$ and $Fire^{\Delta/\Delta}$ mice stained for the myelin proteins MAG (green) and MBP (magenta) (left and middle) and imaged by electron microscopy (right). **j**, Mean number of myelinated axons per mm[2] ± s.e.m. in corpus callosum from $Fire^{+/+}$ and $Fire^{\Delta/\Delta}$ mice. $n = 3$ $Fire^{+/+}$ mice and 4 $Fire^{\Delta/\Delta}$ mice. $P = 0.5216$, two-tailed unpaired Student's $t$-test. **k**, Mean number of myelinated axons in corpus callosum from $Fire^{+/+}$ and $Fire^{\Delta/\Delta}$ mice per axon diameter. $n = 3$ $Fire^{+/+}$ mice and 4 $Fire^{\Delta/\Delta}$ mice. $P = 0.9139$, two-way ANOVA with Sidak's multiple comparisons test. Scale bars, 25 µm (**b,d,i** (left and middle)), 75 µm (**c,f**) and 1 µm (**i** (right)).

macrophages are present[10,11]. These mice do not have many of the confounding issues that occur in other microglia-deficient models, such as developmental death, bone abnormalities and absence of CNS perivascular macrophages and monocytes[10]. Using the microglia marker TMEM119, we confirmed that microglia were depleted in the largest white matter tract of the brain, the corpus callosum (Fig. 1b). The few IBA1[+] macrophages retained in $Fire^{\Delta/\Delta}$ mice were positive for the perivascular macrophage marker LYVE1 (Fig. 1c,d). Moreover, the presence of these cells was confirmed by the observation of CD206[+] cells adjacent to CD31[+] blood vessels (Extended Data Fig. 1a). Perivascular macrophage densities were not significantly altered in $Fire^{\Delta/\Delta}$ mice at this time point (Fig. 1e) or at older ages (Extended Data Fig. 1b,c) despite the reduced expression of CSF1R (Extended Data Fig. 1d–f).

Similar astrocyte numbers (GFAP[+]SOX9[+]) were observed in the corpus callosum of $Fire^{\Delta/\Delta}$ mice and $Fire^{+/+}$ littermates (GFAP[+]SOX9[+] cells; Extended Data Fig. 1g,h). Notably, at an age when myelination is underway (postnatal day 25 (P25) to P30), $Fire^{\Delta/\Delta}$ mice had generated mature oligodendrocytes (OLIG2[+]CC1[+]) (Fig. 1f,g), which formed a similar proportion of the oligodendrocyte lineage (OLIG2[+]) as in $Fire^{+/+}$ littermates (Fig. 1h). Myelin was formed in $Fire^{\Delta/\Delta}$ mice, as indicated by the expression of the myelin proteins MAG, MBP, CNPase, MOG and PLP in the corpus callosum (Fig. 1i and Extended Data Fig. 2a–c) and the cerebellum (Extended Data Fig. 2d,e). Ultrastructural analysis confirmed that myelination proceeded in $Fire^{\Delta/\Delta}$ mice (Fig. 1i), with no significant difference in the number of myelinated axons compared with $Fire^{+/+}$ mice (Fig. 1j) regardless of axon diameter (Fig. 1k).

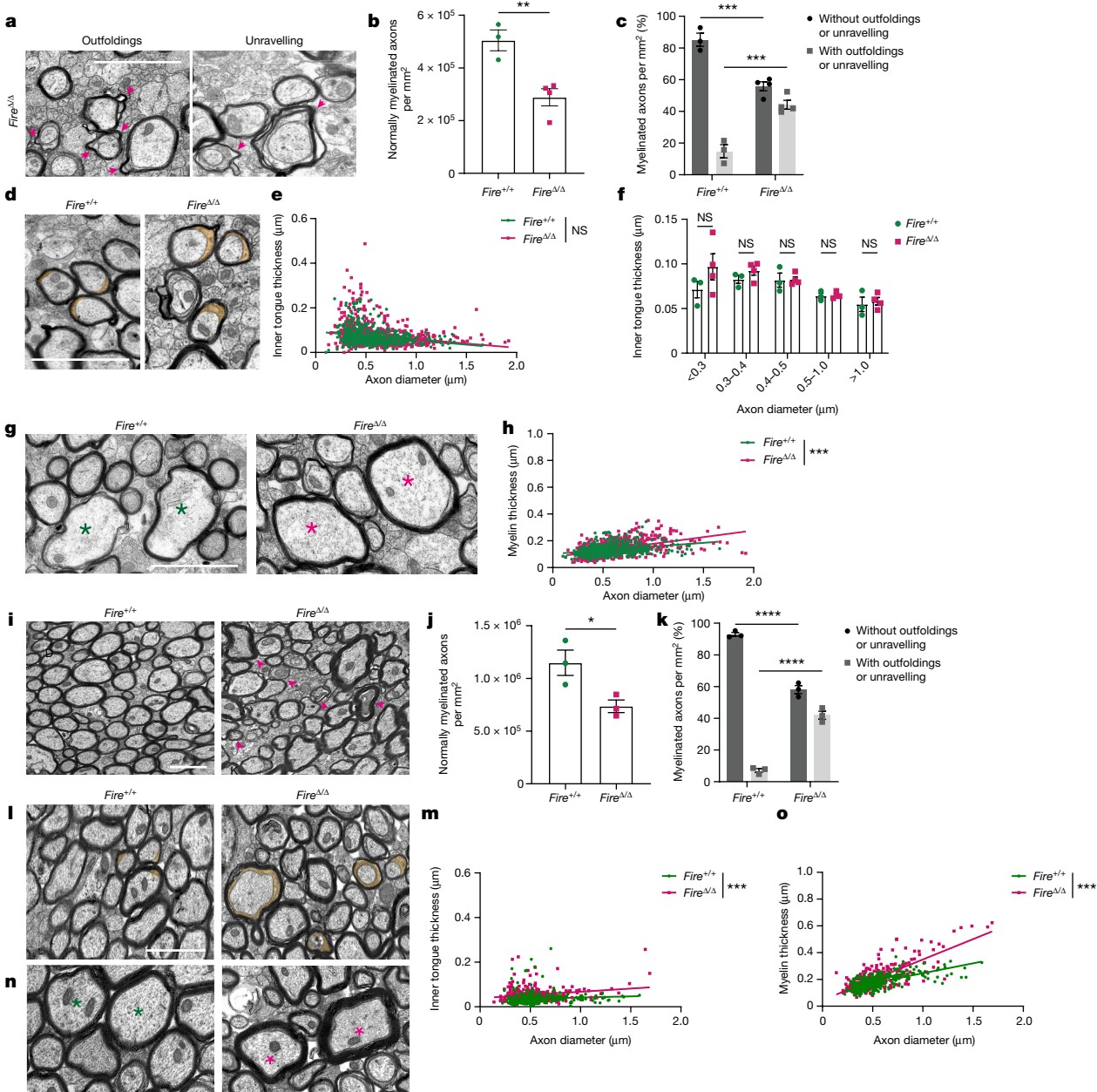

**Fig. 2 | Microglia regulate myelin growth.** Assessments at 1 month (**a**–**h**) and 3–4 months (**i**–**o**) of age in $Fire^{+/+}$ and $Fire^{\Delta/\Delta}$ mice. **a**, Images of myelin abnormalities (arrowheads) in $Fire^{\Delta/\Delta}$ mice. **b**, Mean number of normally myelinated axons per mm² ± s.e.m. $n = 3$ $Fire^{+/+}$ mice and 4 $Fire^{\Delta/\Delta}$ mice. **\*\*** $P = 0.008$, two-tailed unpaired Student's $t$-test. **c**, Mean proportion of axons with and without outfoldings and unravelling per mm² ± s.e.m. $n = 3$ $Fire^{+/+}$ mice and 4 $Fire^{\Delta/\Delta}$ mice. With, **\*\*\*** $P = 0.0006$; without, **\*\*\*** $P = 0.0006$; one-way ANOVA with Tukey's multiple comparisons test. **d**, Images of inner tongues (orange). **e**, Inner tongue thickness (μm) versus axon diameter. $n = 200$ axons per mouse, 3 $Fire^{+/+}$ mice and 4 $Fire^{\Delta/\Delta}$ mice. **f**, Mean inner tongue thickness per axon diameter. $n = 3$ $Fire^{+/+}$ mice and 4 $Fire^{\Delta/\Delta}$ mice. $P = 0.1099$ (<0.3 μm), 0.8933 (0.3–0.4 μm), >0.999 (0.4–0.5 μm), >0.999 (0.5–1.0 μm), 0.9989 (>1.0 μm), two-way ANOVA with Sidak's multiple comparisons test. **g**, Images of myelin thickness (asterisks indicate axons of similar size). **h**, Myelin thickness versus axon diameter. $n = 200$ axons per mouse, 3 $Fire^{+/+}$ mice and 4 $Fire^{\Delta/\Delta}$ mice. **\*\*\*** $P < 0.0001$, simple linear regression of slopes. **i**, Images of myelin abnormalities (arrowheads) in $Fire^{\Delta/\Delta}$ mice. **j**, Mean number of normally myelinated axons per mm² ± s.e.m. $n = 3$ mice per group. **\*** $P = 0.0372$, two-tailed unpaired Student's $t$-test. **k**, Mean proportion of axons with and without outfoldings and unravelling per mm² ± s.e.m. $n = 3$ mice per group. With, **\*\*\*\*** $P < 0.0001$; without, **\*\*\*\*** $P < 0.0001$; one-way ANOVA with Tukey's multiple comparisons test. **l**, Images of inner tongues (orange). **m**, Inner tongue thickness versus axon diameter. $n = 100$ axons per mouse, 3 mice per group. **\*\*\*** $P < 0.0001$, simple linear regression of intercepts. **n**, Images of myelin thickness (asterisks indicate axons of similar size). **o**, Myelin thickness versus axon diameter. $n = 100$ axons per mouse, 3 mice per group. **\*\*\*** $P < 0.0001$, simple linear regression of slopes. Scale bars, 1 μm (**a**,**d**,**g**,**i**,**l**,**n**).

Our results indicate that microglia are dispensable for oligodendrocyte maturation and developmental myelin ensheathment. This finding is in contrast to previous attributions of these functions to microglia following depletion of all CNS macrophage populations.

## Microglia prevent hypermyelination

However, $Fire^{\Delta/\Delta}$ mice showed abnormal myelin structure indicative of hypermyelination. At P25–P30, there was an increase in myelin

that was outfolding or unravelling in *Fire*$^{\Delta/\Delta}$ mice (Fig. 2a–c), which was documented in 44% of sheaths compared with only 14% in *Fire*$^{+/+}$ controls. *Fire*$^{\Delta/\Delta}$ mice showed a non-significant trend of enlarged areas of uncompacted myelin (inner tongue) on smaller diameter axons (Fig. 2d–f). As enlarged inner tongues precluded conventional *g* ratio analysis, we measured myelin thickness directly and observed increased myelin thickness in *Fire*$^{\Delta/\Delta}$ mice preferentially on large diameter axons (Fig. 2g,h). Axon diameter-dependent observations may reflect that myelination of larger diameter axons occurs before that of smaller ones, first involving growth at the inner tongue followed by compaction that thickens the myelin sheath. We assessed the impact of these myelin changes on axonal health in *Fire*$^{\Delta/\Delta}$ mice. Although expression of phosphorylated neurofilament was unaffected at 1 month of age, axonal spheroids, which are indicative of impaired axonal transport, were occasionally observed at later ages in <0.1% of myelinated axons (Extended Data Fig. 2f–h). Myelin outfoldings and unravelling persisted in *Fire*$^{\Delta/\Delta}$ mice at 3–4 months of age (Fig. 2i–k), and enlarged inner tongues and increased myelin thickness were observed across all axon diameters compared with *Fire*$^{+/+}$ mice (Fig. 2l–o) and younger *Fire*$^{\Delta/\Delta}$ mice (Extended Data Fig. 3). Therefore, hypermyelination occurs in the absence of microglia, which indicates that microglia are required for the regulation of myelin growth.

Given that these changes in myelin structure are sufficient to cause cognitive impairment in other models[12,13], we evaluated cognition in *Fire*$^{\Delta/\Delta}$ mice using the Barnes maze spatial learning and memory task (Extended Data Fig. 4a). Both *Fire*$^{\Delta/\Delta}$ and *Fire*$^{+/+}$ mice became progressively faster at locating the target hole with the underlying escape chamber (primary latency), which indicated spatial learning in these mice (Extended Data Fig. 4b,c). Following removal of the escape chamber, probes 1 h and 3 days later indicated no memory-encoding deficit, as indicated by the percentage of time spent in the target quadrant and the number of nose pokes in and around the target hole (Extended Data Fig. 4d–f). Next, we tested cognitive flexibility, which is learning to adjust thinking from an old to a new situation and is highly dependent on the structural integrity of myelin[6,14,15]. This experiment involves learning to locate an escape hole placed 180° from the original target (Extended Data Fig. 4g–k). Although the time taken to locate the new target was unimpaired in *Fire*$^{\Delta/\Delta}$ mice, significantly more errors were made before reaching it (Extended Data Fig. 4h,i), which indicated that these mice have poor cognitive flexibility. *Fire*$^{\Delta/\Delta}$ mice did not have confounding anxiety or motor deficits (Extended Data Fig. 4l–r). Therefore, the absence of microglia is associated with impaired cognitive flexibility.

Recent studies have indicated that learning and memory encoding require new oligodendrogenesis, and that long-term consolidation of this information involves increased myelination[16,17]. Therefore, we assessed whether these processes occur in the absence of microglia. The generation of new oligodendrocytes from proliferating progenitor cells was identified through the incorporation of 5-ethynyl-2′-deoxyuridine (EdU) provided during the cognitive testing stage. There was a similar number of newly generated oligodendrocytes in *Fire*$^{\Delta/\Delta}$ mice and *Fire*$^{+/+}$ mice (Extended Data Fig. 5a–d), which was consistent with the largely unimpaired learning and memory encoding observed in *Fire*$^{\Delta/\Delta}$ mice. However, whereas *Fire*$^{+/+}$ mice had a significantly increased number of myelinated axons in the corpus callosum 6 weeks after completion of the cognitive task, this did not differ between untrained and trained *Fire*$^{\Delta/\Delta}$ mice (Extended Data Fig. 5e–g). These findings suggest that the absence of microglia prevents the increase in myelination that normally occurs with consolidation of new spatial information. Of note, we did not find an association between the number of myelinated axons in a given mouse and its reversal cognitive performance (Extended Data Fig. 5h). This result suggests that either a threshold number of myelinated axons is required for cognitive flexibility or that myelin structural changes may be more relevant for this function.

## Microglia prevent demyelination

Assessment of myelin in *Fire*$^{\Delta/\Delta}$ mice at 6 months of age showed areas of substantial demyelination (Fig. 3a) and areas of patchy demyelination. This in turn resulted in a significant decrease in the number and proportion of myelinated axons compared with *Fire*$^{+/+}$ mice (Fig. 3b–d). The patchy nature of demyelination did not result in widespread loss of myelin protein across the corpus callosum (Extended Data Fig. 6a–c). Axonal spheroids were rarely observed (<0.1% of axons). Axons retaining myelin in 6-month-old *Fire*$^{\Delta/\Delta}$ mice had reduced inner tongue size and thinner myelin compared with *Fire*$^{+/+}$ mice (Fig. 3e,f and Extended Data Fig. 6d) and with 3–4-month-old *Fire*$^{\Delta/\Delta}$ mice (Extended Data Fig. 3). Demyelination was not associated with loss of oligodendrocytes at this age or younger (Extended Data Fig. 6e–h). Demyelination was initiated at 4.5 months of age in *Fire*$^{\Delta/\Delta}$ mice (Extended Data Fig. 6i–l), and unmyelinated axons were medium-to-large calibre (mean 0.73 μm ± 0.1 s.e.m.) in size (Extended Data Fig. 6l). This result indicated that these axons underwent demyelination first, as medium-to-large diameter axons showed hypermyelination immediately before demyelination at 3–4 months of age (Extended Data Fig. 6k); therefore, hypermyelination may precede demyelination. These findings demonstrate that the lack of microglia is sufficient to induce CNS demyelination with increasing age.

## Microglia maintain existing myelin

We next asked whether these changes in myelin growth and integrity reflect disruption of myelin formation or myelin maintenance. To that end, we depleted microglia in mice when developmental myelination is complete (2 months of age onwards) by providing the CSF1R inhibitor PLX5622 in the diet of adult *Fire*$^{+/+}$ mice for 1 month. This resulted in >50% reduction of IBA1$^+$ cells at 3 months of age (Extended Data Fig. 7a–c). Compared with mice fed the control diet, microglia depletion from 2 to 3 months of age resulted in enlarged inner tongues and thicker myelin (Extended Data Fig. 7d–g), whereas depletion from 5 to 6 months of age caused patchy demyelination (Extended Data Fig. 7j–l). Oligodendrocyte numbers were unchanged (Extended Data Fig. 7h,i). Therefore, microglia depletion in adult mice mirrored the hypermyelination and myelin degeneration observed at equivalent ages in *Fire*$^{\Delta/\Delta}$ mice, which indicated that microglia are required for myelin maintenance once it is already formed.

## Microglia deficits in humans affect myelin health

After demonstrating that microglia are required for myelin health in mice, we investigated the relevance of these findings in humans. We analysed samples from individuals with the rare leukoencephalopathy adult-onset leukoencephalopathy with axonal spheroids and pigmented glia (ALSP) (Extended Data Table 1). In ALSP, heterozygous *CSF1R* mutations lead to cognitive dysfunction in association with reduced IBA1$^+$ parenchymal cells, especially in frontal white matter, whereas those in the grey matter are relatively preserved[18]. In comparison to age-matched individuals who died of non-neurological causes, there was a significant decrease in IBA1$^+$ microglia and macrophages in the frontal white matter of individuals with ALSP (Fig. 4a,b). Moreover, there was a relative increase in the proportion of perivascular macrophages (IBA1$^+$LYVE1$^+$) (Extended Data Fig. 8a–d). Ultrastructural analysis of ALSP white matter revealed myelin outfoldings and unravelling (Fig. 4c), thicker myelin (Fig. 4c,d) and enlarged inner tongues (Fig. 4c,e and Extended Data Fig. 8e–g). Demyelination was also observed and progressively worsened with age (Fig. 4f). Larger axon diameters were noted in ALSP samples compared with unaffected samples (Fig. 4d,e). This result is consistent with axonal swelling being a typical pathological feature of this disorder; however, myelin was still thicker than would be expected of these axon diameters. Extra thick myelin in ALSP was

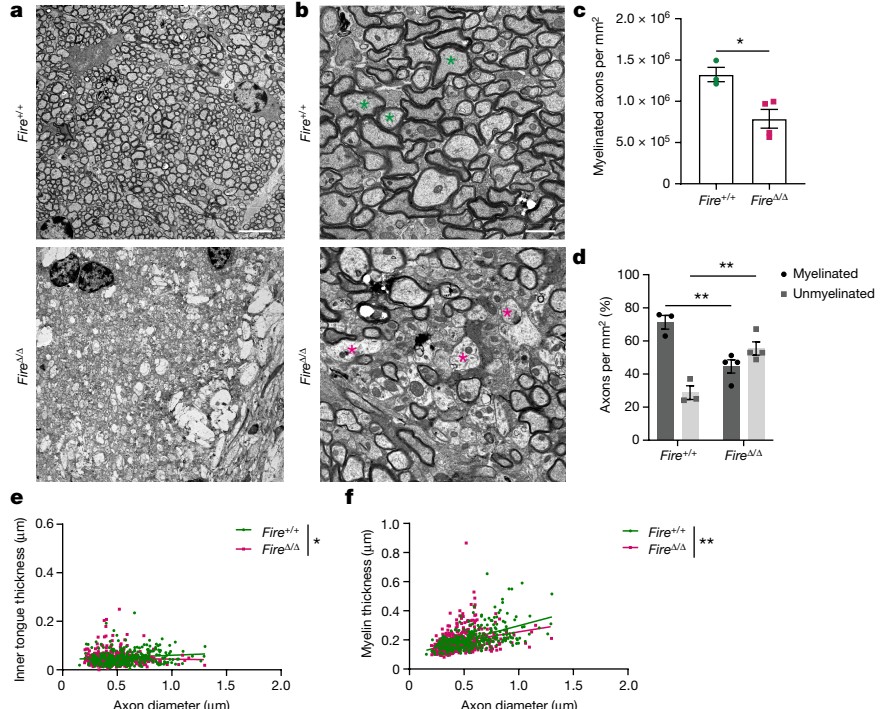

**Fig. 3 | Absence of microglia causes demyelination. a**, Images of substantial demyelination in the corpus callosum of 6-month-old $Fire^{\Delta/\Delta}$ mice compared with age-matched $Fire^{+/+}$ mice. Scale bar, 5 μm. **b**, Images of patchy demyelination in $Fire^{\Delta/\Delta}$ mice compared with age-matched $Fire^{+/+}$ mice. Asterisks indicate axons of similar size. Scale bar, 1 μm. **c**, Mean number of myelinated axons per mm² ± s.e.m. $n = 3$ $Fire^{+/+}$ mice and 4 $Fire^{\Delta/\Delta}$ mice. *$P = 0.0170$, two-tailed unpaired Student's $t$-test. **d**, Mean proportion of axons that are myelinated (black) and unmyelinated (grey) per mm² ± s.e.m. $n = 3$ $Fire^{+/+}$ mice and 4 $Fire^{\Delta/\Delta}$ mice. Myelinated, **$P = 0.0051$; unmyelinated, **$P = 0.0051$; one-way ANOVA with Tukey's multiple comparisons test. **e**, Inner tongue thickness versus axon diameter. $n = 100$ axons per mouse, 3 $Fire^{+/+}$ mice and 4 $Fire^{\Delta/\Delta}$ mice. *$P = 0.0332$, simple linear regression of slopes. **f**, Myelin thickness versus axon diameter. $n = 100$ axons per mouse, 3 $Fire^{+/+}$ mice and 4 $Fire^{\Delta/\Delta}$ mice. **$P = 0.0062$, simple linear regression of slopes.

associated with myelin unravelling (Fig. 4c), which may indicate early stages of a transition from hypermyelination to demyelination. These findings show that a reduction in white matter microglia in humans is associated with hypermyelination and eventual demyelination.

## Microglia suppress oligodendrocyte state

We next sought to determine the cellular and molecular mechanisms that underpin the loss of myelin health in the absence of microglia. To that end, we performed single-cell RNA sequencing of brain samples from $Fire^{\Delta/\Delta}$ and $Fire^{+/+}$ mice at 1 month of age (Extended Data Fig. 9a–j). Mature oligodendrocytes were identified through the expression of myelin genes (*Plp*, *Mog*, *Mag* and *Mbp*) and the absence of expression of markers for other cell types (Extended Data Fig. 9a–d). The oligodendrocytes were clustered into four states (Oligo1 to Oligo4) (Fig. 5a and Extended Data Fig. 9k,l). Notably, we observed that cluster 1 oligodendrocytes (Oligo1) were almost exclusively found in $Fire^{\Delta/\Delta}$ mice (Fig. 5b,c) and distinguished by the high expression of genes (Supplementary Table 1 and Fig. 5d) including *Serpina3n* and *C4b* (Fig. 5d,e). We confirmed the presence of SERPINA3N⁺OLIG2⁺ cells almost exclusively in $Fire^{\Delta/\Delta}$ mouse white matter, whereas these cells were undetectable in grey matter of either genotype (Fig. 5f,g). Analysis of differentially expressed genes in the Oligo1 cluster revealed that the top canonical pathways were related to lipid synthesis and metabolism. Specifically, ingenuity pathway analysis highlighted the following significant pathways: superpathway of cholesterol biosynthesis ($P = 8 \times 10^{-12}$) and cholesterol biosynthesis I–III ($P = 1.23 \times 10^{-7}$). Analysis using the DAVID bioinformatics resource highlighted the following pathways: lipid biosynthesis ($P = 3.8 \times 10^{-8}$); lipid metabolism ($P = 5.1–7.0 \times 10^{-6}$); and cholesterol metabolism ($P = 9.2 \times 10^{-4}$). Of note, cholesterol is enriched in myelin and required for myelin growth[19]. Lipidomics analysis of $Fire^{\Delta/\Delta}$

mouse white matter revealed an increase in cholesterol esters and a decrease in triglycerides (Fig. 5h), which was indicative of excess cholesterol and impaired lipid export, respectively. This result is consistent with the observed surplus in myelin membrane formation in $Fire^{\Delta/\Delta}$ mice. Moreover, dysregulation of genes associated with cholesterol transport has been reported in ALSP[20].

## TGFβ1 signalling regulates myelin health

To determine how the absence of microglia contributes to these findings, we assessed the predicted upstream regulators of genes in the Oligo1 cluster. TGFβ1 was identified as a prime candidate ($P = 7.7 \times 10^{-13}$; Supplementary Table 2), as it is predominantly expressed by microglia in both mouse and human brain (https://www.brainrnaseq.org)[21,22], and it is known to influence lipid metabolism[23]. Accordingly, TGFβ1 levels were significantly downregulated in $Fire^{\Delta/\Delta}$ mouse white matter (Fig. 6a). We next assessed the capacity of oligodendrocytes to respond to TGFβ1. Although there were abundant TGFβR1⁺OLIG2⁺ cells in $Fire^{+/+}$ mice, the number and percentage of these cells were significantly reduced in $Fire^{\Delta/\Delta}$ mice (Fig. 6b–d). These findings led us to ask whether elimination of TGFβR1 signalling in oligodendrocytes is sufficient to cause myelin pathology. As *Tgfb1* knockout in the CNS results in a confounding decrease in microglia number, loss of microglia homeostasis and monocyte infiltration[24], we utilized a conditional knockout of *Tgfbr1* in mature oligodendrocytes ($Plp^{creERT}$;$Tgfbr1^{fl/fl}$). Following tamoxifen administration from P14 to P18 (Fig. 6e), TGFβR1 expression by oligodendrocyte lineage cells was significantly reduced in $Plp^{creERT}$;$Tgfbr1^{fl/fl}$ mice at 1 month of age compared with controls (Extended Data Fig. 10a,b). There was no significant impact on the number of myelinated axons (Extended Data Fig. 10c). However, by P28, conditional knockout mice had enlarged

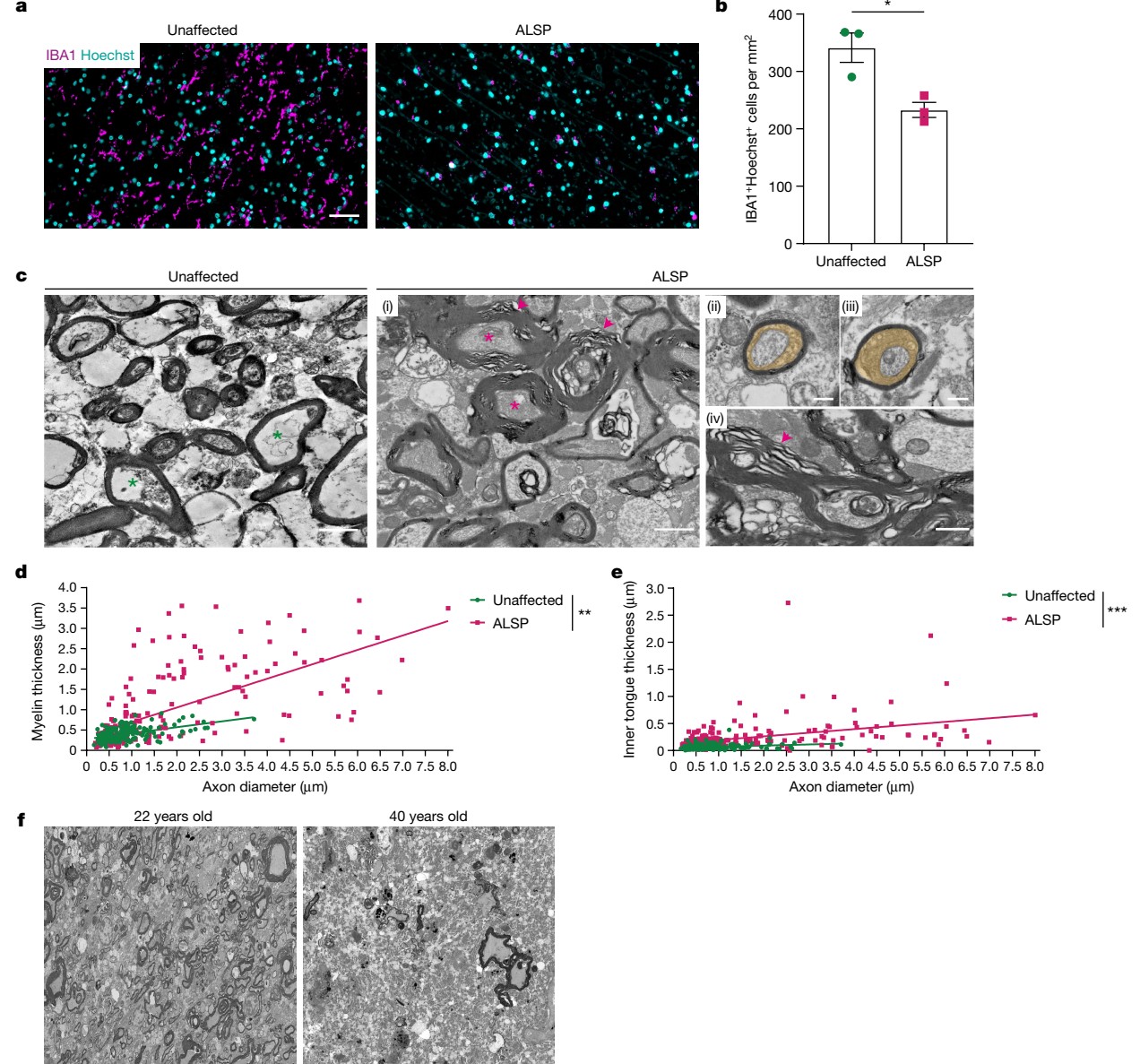

**Fig. 4 | Reduction of microglia in human white matter is associated with hypermyelination and demyelination. a**, Images of IBA1[+] macrophages (magenta) in human frontal white matter from individuals with ALSP and unaffected age-matched individuals, counterstained with Hoechst (cyan). **b**, Mean IBA1[+] cells per mm[2] ± s.e.m. in samples from unaffected individuals and individuals with ALSP. $n = 3$ samples per group. *$P = 0.0197$, two-tailed unpaired Student's $t$-test. **c**, Images of frontal white matter in unaffected and ALSP samples (**i**). Asterisks indicate axons of similar size, arrowheads indicate myelin abnormalities. Panels (ii) and (iii) show enlarged inner tongues in ALSP (orange). Panel (iv) shows myelin outfoldings and unravelling in ALSP (arrow). **d**, Myelin thickness versus axon diameter. $n = 100$ axons per sample, $n = 2$ samples per group. **$P = 0.002$, simple linear regression of slopes. **e**, Inner tongue thickness (µm) versus axon diameter. $n = 100$ axons per sample, $n = 2$ samples per group. ***$P < 0.0001$, simple linear regression of intercepts. **f**, Images of extent of demyelination of frontal white matter in individuals with ALSP; 22-year-old individual and 40-year-old individual. Scale bars, 0.5 µm (**c** (ii)–(iv)), 1 µm (**c** (i)). 10 µm (**f**) or 50 µm (**a**).

inner tongues on smaller diameter axons (Fig. 6f–h and Extended Data Fig. 10d) and thicker myelin on larger diameter axons (Fig. 6f,g,i and Extended Data Fig. 10e) relative to tamoxifen-treated floxed and wild-type controls. This result mimics the observations in $Fire^{\Delta/\Delta}$ mice at 1 month of age.

We next asked whether stimulating TGFβR1 signalling in $Fire^{\Delta/\Delta}$ mice could rescue myelin health. In the absence of sufficient TGFβR1 expression by oligodendrocytes in $Fire^{\Delta/\Delta}$ mice, we used a small-molecule activator of downstream TGFβ signalling, SRI-011381 hydrochloride. This approach bypasses the need to stimulate the receptor by activating the SMAD2–SMAD3 pathway[25]. We administered SRI-011381 hydrochloride to $Fire^{\Delta/\Delta}$ mice from 2 months of age to observe the potential

impact on the significant myelin pathology observed by 3 months of age (Fig. 6j). SRI-011381 had no influence on myelinated axon number (Extended Data Fig. 10f) but significantly reduced inner tongue thickness (Fig. 6k–m and Extended Data Fig. 10g) and myelin thickness (Fig. 6k,l,n and Extended Data Fig. 10h) compared with vehicle-treated $Fire^{\Delta/\Delta}$ mice. Following SRI-011381 hydrochloride treatment, these parameters overlapped with those in age-matched $Fire^{+/+}$ mice (Fig. 6m,n). This result suggested that disrupted TGFβR1 signalling is the primary mechanism by which myelin is dysregulated in $Fire^{\Delta/\Delta}$ mice. Altogether, these findings reveal the importance of the TGFβ1–TGFβR1 axis in microglia–oligodendrocyte communication for the regulation of myelin health.

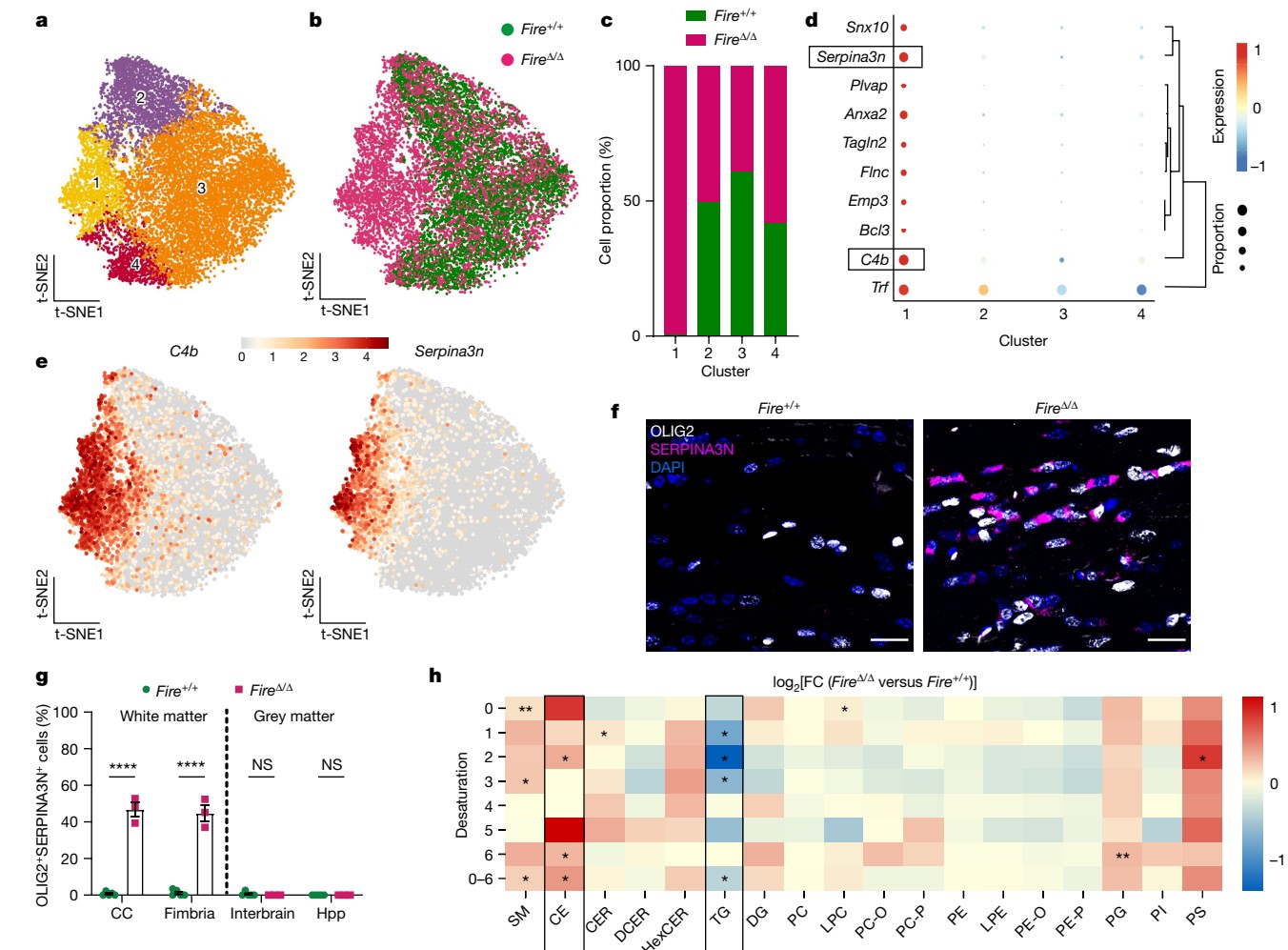

**Fig. 5 | An oligodendrocyte state is enriched in *Fire*^Δ/Δ mice. a**, T-distributed stochastic neighbour embedding (t-SNE) plots of oligodendrocyte clusters in *Fire*^+/+ and *Fire*^Δ/Δ mice: Oligo1 (yellow), Oligo2 (purple), Oligo3 (orange) and Oligo4 (red). **b**, Distribution of oligodendrocyte clusters between *Fire*^+/+ (green) and *Fire*^Δ/Δ (magenta) mice. **c**, Oligodendrocyte cluster proportions in *Fire*^+/+ and *Fire*^Δ/Δ mice. *n* = 4 mice per group. **d**, Top differentially expressed genes in Oligo1 compared with Oligo2 to Oligo4. Level of normalized expression of each gene is indicated by the heatmap: 1, red, −1, blue. Proportion of cells expressing each gene is indicated by the size of the circles in the plot. **e**, t-SNE projection of expression of *C4b* and *Serpina3n*. **f**, Images of OLIG2^+ cells (white) expressing SERPINA3N (magenta) and counterstained with DAPI (blue) in the corpus callosum. **g**, Mean percentage of OLIG2^+ cells expressing SERPINA3N ± s.e.m. in white matter (corpus callosum (CC) and fimbria) and grey matter (interbrain and hippocampus (Hpp)). *n* = 5 *Fire*^+/+ mice and 3 *Fire*^Δ/Δ mice. CC and fimbria, ****P < 0.0001; interbrain and Hpp, *P* = 0.9989 and *P* > 0.9999, respectively; two-way ANOVA with Sidak's multiple comparisons test. **h**, Lipidomics analysis represented as log$_2$(fold change (FC)) in *Fire*^Δ/Δ mice versus *Fire*^+/+ mice, with upregulated lipid species indicated in red and downregulated lipid species indicated in blue, ordered based on desaturation of fatty acids (double bonds) and total class value (0–6). Boxes indicate lipid species of interest. SM, sphingomyelins; CE, cholesterol esters; CER, ceramides; DCER, dihydroceramides; HexCER, hexosylceramides; TG, triaglycerides; DG, diacylglycerides; PC, phosphatidylcholine; LPC, lysophosphatidylcholine; PC-O, 1-alkyl,2-acylphosphatidylcholines; PC-P, 1-alkenyl,2-acylphosphatidylcholines; PE, phosphatidylethanolamine; LPE, lysophosphatidylethanolamine; PE-O, 1-alkyl,2-acylphosphatidylethanolamines; PE-P, 1-alkenyl,2-acylphosphatidylethanolamines; PG, phosphatidylglycerol; P I, phosphatidylinositol; PS, phosphatidylserine. *n* = 3 mice per group. One-sample *t*-test of log$_2$(FC) against a value of 0: SM-0, **P* = 0.0058; SM-3, **P* = 0.0202; SM-0–6, **P* = 0.0347; CE-2, **P* = 0.0384; CE-6, **P* = 0.0474; CE-0–6, **P* = 0.0276; CER-1, **P* = 0.0171; TG-1, **P* = 0.0301; TG-2, **P* = 0.0116; TG-3, **P* = 0.0432; TG-0–6, **P* = 0.0412; LPC-0, **P* = 0.0415; PG-6, **P* = 0.0055; PS-2, **P* = 0.0459.

## Discussion

Here we identified the requirement for microglia in the maintenance of healthy CNS myelin. Our use of a new transgenic model in which microglia are lacking while other CNS macrophages are present revealed that microglia are not required for developmental oligodendrocyte maturation or myelin ensheathment. Rather, microglia preserve the structural integrity of myelin. We demonstrated the role of microglia in limiting hypermyelination and preventing demyelination of existing myelin sheaths in adulthood. Our results complement recent work implicating microglia in the regulation of myelin

sheath number during myelin formation in embryonic development[26] and the inhibition of myelination of regenerated axons after optic nerve injury[27]. This raises the question of whether other macrophage populations, such as perivascular and/or peripheral macrophages, influence developmental myelination or whether other glial cell types, such as astrocytes, have a compensatory role in the absence of microglia. Our work indicates that a threshold number of microglia is needed to maintain myelin health, as even just a 50–60% decrease in white matter microglia in mice or humans is associated with loss of myelin integrity. Altogether, these findings suggest prudence in the current trials of CSF1R inhibitors to deplete microglia in cancer

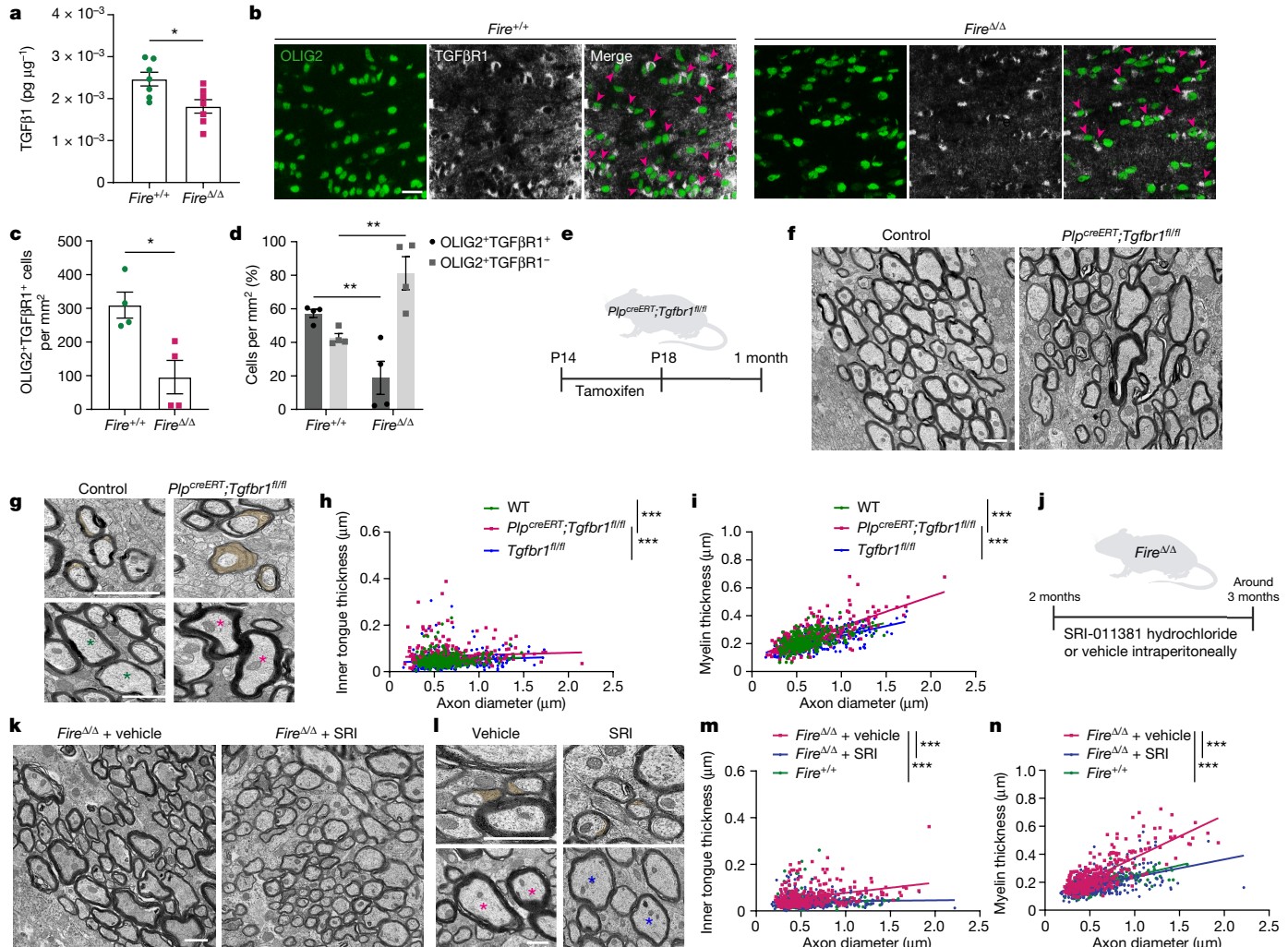

**Fig. 6 | TGFβR1 signalling regulates myelin integrity. a**, TGFβ1 levels (pg μg⁻¹ ± s.e.m.) in corpus callosum normalized to the respective total protein. *n* = 7 mice per group. **P* = 0.0150, two-tailed unpaired Student's *t*-test. **b**, Images of OLIG2⁺ cells (green) expressing TGFβR1 (white), (arrowheads) in 1-month-old mice. **c**, Mean OLIG2⁺TGFβR1⁺ cells ± s.e.m. *n* = 4 mice per group. **P* = 0.0144, two-tailed unpaired Student's *t*-test. **d**, Mean percentage of OLIG2⁺ cells that are TGFβR1⁺ (black) or TGFβR1⁻ (grey) (± s.e.m.), **P* = 0.0120 and **P* = 0.0120, respectively, *n* = 3 mice per group. One-way ANOVA with Tukey's multiple comparisons test. **e**, Mice were treated with tamoxifen from P14 to P18 then assessed at 1 month. **f**, Images of *Plp^creERT;Tgfbr1^fl/fl* mice versus control. **g**, Images of *Plp^creERT;Tgfbr1^fl/fl* mice, indicating enlarged inner tongues (orange) and hypermyelination (asterisks) compared with control. **h**, Inner tongue thickness versus axon diameter. *n* = 100 axons per mouse, 3 mice per group. ****P* < 0.0001, simple linear regression of intercepts. *Plp^creERT;Tgfbr1^fl/fl* versus wild type and versus *Tgfbr1^fl/fl*, ****P* < 0.0001; wild type versus *Tgfbr1^fl/fl*, *P* = 0.3228; Kruskal–Wallis with Dunn's multiple comparisons test. **i**, Myelin thickness versus axon

diameter. *n* = 100 axons per mouse, 3 mice per group. ****P* < 0.0001, simple linear regression of slopes. *Plp^creERT;Tgfbr1^fl/fl* versus wild type, ****P* = 0.0008 and versus *Tgfbr1^fl/fl*, ****P* = 0.0001; wild type versus *Tgfbr1^fl/fl*, *P* > 0.9999 Kruskal–Wallis with Dunn's multiple comparisons test. **j**, *Fire^Δ/Δ* mice were treated with SRI-011381 hydrochloride (SRI; 30 mg kg⁻¹) or vehicle from 2–3 months of age. **k**, Images of *Fire^Δ/Δ* mice treated with vehicle or SRI-011381. **l**, Images of *Fire^Δ/Δ* mice treated with vehicle or SRI-011381 indicating inner tongue size (orange) and myelin thickness (asterisks). **m**, Inner tongue thickness versus axon diameter. *n* = 100 axons per mouse, and *n* = 3 vehicle treated and 4 SRI treated. ****P* < 0.0001, simple linear regression of slopes. Vehicle versus SRI and versus *Fire^+/+*, ****P* < 0.0001; SRI versus *Fire^+/+*, *P* = 0.0519, Kruskal–Wallis with Dunn's multiple comparisons test. **n**, Myelin thickness versus axon diameter. *n* = 100 axons per mouse, and *n* = 3 vehicle treated and 4 SRI treated. ****P* < 0.0001, simple linear regression of slopes. Vehicle versus SRI and versus *Fire^+/+*, ****P* < 0.0001; SRI versus *Fire^+/+*, *P* > 0.9999, Kruskal–Wallis with Dunn's multiple comparisons test. Scale bars, 1 μm (**f,g,k,l**) or 25 μm (**b**).

or neurological conditions, and warrant the monitoring of potential off-target effects on myelin health.

We associated structural changes in myelin in the absence of microglia with poor cognitive flexibility, along with impaired de novo myelination that normally underpins long-term memory consolidation[17,28]. This builds on previous work revealing that microglia dysregulation (in response to chemotherapy) is sufficient to disrupt myelin structure and cognitive function[9,13], as we have uncovered the requirement for healthy microglia in the prevention of these pathologies. Given the close relationship between myelin structure and neuronal activity[29], our findings also raise the possibility of a role for microglia in influencing

adaptive myelination to reinforce cognitive circuits. However, understanding the impact of the absence of microglia on neuronal activity and synaptic health requires further investigation. In addition, our work has important implications for understanding cellular networks that contribute to cognitive decline with ageing, in which there is prominent hypermyelination[3,12], demyelination and impaired new myelination[5,28], alongside microglia dysfunction[21,30,31]. This may also be relevant to dementia-associated neurodegenerative disease, given that in a mouse model of Alzheimer's disease, there are opposite changes in gene modules primarily associated with microglia (and astrocytes) compared with those related to oligodendrocytes and myelination[32]. That study

also identified that the oligodendrocyte-associated module is initially upregulated followed by a downregulation in microenvironments with the highest β-amyloid accumulation. Whether this represents initial hypermyelination followed by demyelination remains to be determined.

The worsening myelin pathology we observed with age in response to microglia depletion points to an increasing dependence on healthy microglia for myelin integrity with ageing. Microglia dysfunction may therefore initiate myelin damage with ageing and in neurodegenerative disorders. Previous work has implicated the peripheral immune system or primary oligodendrocyte dysregulation in inducing demyelination. Here we propose that the contribution of microglia now also needs to be considered. Our data suggest that hypermyelination may precede demyelination, which raises the question of whether this sequence of events underpins myelin damage in ageing and neurodegenerative disease. Notably, we identified parallels in dysregulated cellular profiles and molecular mechanisms between microglia-deficient mice and other neurological injury models. These results implicate microglia as crucial regulators of myelin pathology in these contexts. For instance, our data show that microglia normally suppress the appearance of a dysregulated oligodendrocyte state (expressing *Serpina3n* and *C4b*), similar to that recently documented in mouse models of demyelination, ageing and Alzheimer's disease[33–37]. This implies that in pathological contexts, microglia dysregulation may permit these oligodendrocytes to appear. We have made data-mining of this (and other) oligodendrocyte populations in the *Fire*$^{\Delta/\Delta}$ mouse model publicly available on the following website: https://annawilliams.shinyapps.io/shinyApp_oligos_VM/. The functional consequences of the appearance of these oligodendrocytes, and the molecular mechanisms involved in their function, have hitherto been unclear. Our association of these oligodendrocytes with an altered lipid profile (for example, increased cholesterol esters) is consistent with the hypermyelination in *Fire*$^{\Delta/\Delta}$ mice. This may shed light on potential pathological mechanisms in human neurological conditions in which cholesterol esters are increased, such as Huntington's disease and multiple sclerosis[38,39]. The finding that microglia regulate the lipid profile of mature myelinating oligodendrocytes complements their recently discovered role in promoting oligodendrocyte progenitor maturation during myelin regeneration through the supply of a cholesterol pathway intermediate[40]. Our pathway analysis of these oligodendrocytes identified a dysregulated TGFβ−TGFβR1 axis, which we showed is a mediator of myelin pathology. Transcriptomic analyses of ALSP brain samples has indicated dysregulation of the TGFβ pathway specifically in the white matter[20], and an upregulation of *TGFB1* in the least affected white matter region[41]. Signalling downstream of TGFβ receptors is also reduced in ageing and neurodegenerative disease[42,43].

We previously discovered that a subset of microglia expressing a TGFβ superfamily member, activin-A, regulates remyelination efficiency[44,45]. Taken together with the transcriptomic heterogeneity of microglia during development, homeostasis, demyelination, remyelination and ageing[30,44,46–50], we are now poised to ask whether specific microglia states are required to regulate myelin growth and integrity. Recently, studies have identified microglia states associated with white matter, with roles in phagocytosis of dying cells in development[50] or myelin debris in ageing[30], and we found that a shift in functional microglial states underpins their capacity to support remyelination[44,46]. Whether altered heterogeneity with ageing and disease is associated with a loss in supportive microglia states that then contributes to progressive myelin pathology needs to be investigated. Altogether, our study uncovered the role of microglia in preserving myelin health and integrity in adulthood, and highlights microglia as key therapeutic targets in the context of disrupted myelin in ageing and disease.

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

## Methods

### Animals

All experiments were performed under project licences approved by the UK Home Office and issued under the Animals (Scientific Procedures) Act. This study used $Csf1r^{Fire\Delta/\Delta}$ mice on a mixed CBA:C57Bl6J background, wild-type controls from $Csf1r^{Fire\Delta/+}$ crossings, $Plp^{creERT}$ mice (Jackson Laboratories) and $Tgfbr1^{fl/fl}$ mice (provided by S. Karlsson, Lund University). Recombination was induced in $Plp^{creERT};Tgfbr1^{fl/fl}$ mice through the administration of 4-hydroxytamoxifen (100 mg kg$^{-1}$, intraperitoneally; Sigma-Aldrich) dissolved in ethanol and corn oil (1:9) mixture for 5 consecutive days from P14 to P18, then killed at P28. All animals were housed at a maximum number of 6 animals per cage in a 12 h light–dark cycle with unrestricted access to food and water. For animal experiments, the sample size was determined by power analysis calculated by two-sided 95% confidence interval through the normal approximation method using OpenEpi software (Openepi.com), and reached >80% power for all experiments. Both males and females were used throughout the study, except for open-field experiments, for which only male mice were used. Animals were randomized to time points analysed. ARRIVE2 guidelines were followed in providing details of experiments, quantifications and reporting.

### Genotyping

Genomic DNA was extracted from ear biopsy tissue using a Wizard SV genomic purification system (Promega) according to the manufacturer's instructions. $Csf1r^{Fire\Delta/\Delta}$ mice were genotyped using PCR strategies as previously described[10]. Genotyping of $Plp^{creERT};Tgfbr1^{fl/fl}$ mice was performed with genomic DNA extracted from the tail, as previously described for $Plp^{creERT}$ mice[51] and $Tgfbr1^{fl/fl}$ mice[52].

### Immunofluorescence staining of mouse tissue

Mice were intracardially perfused with 4% paraformaldehyde (PFA; w/v; Sigma), and brains were post-fixed overnight and cryoprotected in sucrose before embedding in OCT (Tissue-Tech) and storage at −80 °C. Cryosections (10 µm) were air-dried, permeabilized and blocked for 1 h with 5% normal horse serum (Gibco) and 0.3% Triton-X-100 (Fisher Scientific) in PBS. For myelin protein staining, sections were permeabilized in methanol at −20 °C for 10 min. For EdU visualization, an AlexaFluor-555 Click-iT EdU Cell Proliferation Assay kit (Invitrogen) was applied before immunostaining. Sections were permeabilized with 0.5% Triton X-100 in PBS for 20 min at room temperature (20–25 °C) then incubated in Click-iT reaction cocktail in the dark at room temperature for 30 min and then washed in PBS. Heat-induced antigen retrieval was performed before primary antibody application. Primary antibodies were applied overnight at 4 °C in a humid chamber and included the following: MBP (AbD Serotec, 1:250; MCA409S, clone 12); MAG (Millipore, 1:100; MAB1567, clone 513); MOG (Millipore, 1:100; MAB5680, clone 8-18C5); CNPase (Sigma-Aldrich, 1:100; AMAB91072, clone CL2887); TMEM119 (Abcam, 1:100; ab209064, clone 28-3); IBA1 (Abcam, 1:500; ab5076); CD206 (Abcam, 1:100; ab64693); CD31 (R&D Systems, 1:100; AF3628); LYVE1 (Abcam, 1:100; ab14917); OLIG2 (Millipore, 1:100; AB9610, clone 211F1.1); APC/CC1 (Abcam, 1:100; ab16794, clone CC1); SOX9 (Millipore, 1:500; AB5535); GFAP (Cambridge Bioscience, 1:500; 829401); neurofilament-H (BioLegend, 1:100,000; Covance, PCK-592P); and PLP (Abcam, 1:100; ab28486). For SERPINA3N immunostaining, 6 µm-thick formalin-fixed paraffin-embedded sections were deparaffinized, rehydrated and then placed in a water bath at 85 °C for 30 min in a citrate-based antigen unmasking solution (H-3300-250, Vector Laboratories). Sections were subsequently rinsed in PBS and blocked for 1 h using PBS with 10% donkey serum (D9663, Sigma Aldrich) and 0.2% Triton X-100 (T8787, Sigma Aldrich). Sections were then incubated overnight at 4 °C in anti-SERPINA3N (R&D Systems, 1:100; AF4709) and anti-OLIG2 (Millipore, 1:500; AB9610) in PBS with 5% donkey serum and 0.1% Triton X-100, then washed three times in PBS

with 0.05% Tween-20 (P1379, Sigma Aldrich). Fluorescently conjugated secondary antibodies were applied for 1–2 h at room temperature in a humid chamber (1:500, Life Technologies–Molecular Probes). Following counterstaining with Hoechst or DAPI, slides were coverslipped with Fluoromount-G (Southern Biotech). For TGFβR1 immunostaining of $Fire^{\Delta/\Delta}$ mice, following a wash in TBS with 0.001% Triton X-100 (Sigma), sections were microwaved in antigen unmasking solution (pH 6 citrate buffer, Vector Laboratories) for 10 min, then heated at 60 °C for 30 min. After cooling, sections were washed once with TBS and 0.001% Triton X-100, and endogenous phosphatase and peroxidase activity was blocked with Bloxall (Vector) for 10 min. Blocking was performed for 1 h with 10% heat-inactivated horse serum (Gibco) and 0.5% Triton X-100 in TBS. Primary antibody diluted in blocking solution was applied overnight in a humid chamber at 4 °C. Antibodies used included TGFβR1 (Abcam, 1:100; ab31013) and OLIG2 (Millipore, 1:100; MABN50). Following three washes in TBS with 0.001% Triton X-100, peroxidase-conjugated secondary antibody (Vector) was applied for 1 h at room temperature in a humid chamber. Following further washes, sections were developed using Opal 520 (Akoya) at 1:100 in Plus Amplification Diluent (Akoya) for 10 min in a humid chamber. Slides were washed, and residual peroxidase activity was quenched by applying Bloxall (Vector) for 10 min. For co-staining, another primary antibody was then applied and developed using peroxidase-conjugated secondary antibody and Opal 650 (Akoya) at 1:100 in Plus Amplification Diluent as described above. Following washes in TBS, the sections were counterstained with Hoechst (1:10,000) and mounted with Fluoromount-G (Invitrogen).

Sections were imaged on a Leica SPE or Zeiss LSM 510 confocal microscope. Cell counts were calculated from a measured area based on assumption of circularity using Fiji/ImageJ (Fiji.sc), with three regions of interest quantified per section. Colocalization analysis of SOX9 and GFAP was performed using Imaris software v.9.7.

### Flow cytometry

An enzyme-free brain dissociation protocol was used to gather a myeloid cell-enriched cell suspension to explore CSF1R (CD115) surface protein levels. After transcardially perfusing 10–11-week-old female mice with ice-cold PBS, brains were dissected and minced with a 22A scalpel in HBSS (without Ca$^{2+}$ and Mg$^{2+}$; 14175-053, Gibco) with 25 mM HEPES (10041703, Fisher Scientific). Brains were then homogenized using a Dounce homogenizer (D9938, Kimble) in HBSS (without Ca$^{2+}$ and Mg$^{2+}$) with 25 mM HEPES. Brain homogenates were separated using a 35% Percoll gradient, with centrifugation at 800$g$ for 20 min at 4 °C (with no brake). Cell pellets were collected and washed in PBS (without Ca$^{2+}$ and Mg$^{2+}$; 14190-094, Gibco) with 0.1% low endotoxin BSA (A8806, Sigma Aldrich). Fc receptors were blocked (1:100; 101302, BioLegend) for 15 min at 4 °C on a shaker. Cells were then stained with primary antibodies directed against CD11b (PE; 1:200; 101207; BioLegend, clone M1/70), CD45 (PECy7; 1:200; 103114; BioLegend, clone 30-F11) and CD115 (APC; 1:200; 135510; BioLegend, clone AFS98) for 30 min at 4 °C on a shaker. Samples were then washed and resuspended in PBS (without Ca$^{2+}$ and Mg$^{2+}$) with 0.1% low endotoxin BSA. Single-fluorochrome stained beads, unstained samples and fluorescence minus one samples were used as controls. DAPI was used for cell viability gating. Data were acquired using a BD LSRFortessa flow cytometer. FCS express 7 was used for post-acquisition data analysis.

### Electron microscopy of mouse tissue

Mice were intracardially perfused with 4% PFA (w/v) and 2% glutaraldehyde (v/v; TAAB Laboratories) in 0.1 M phosphate buffer. Tissue was post-fixed overnight at 4 °C and transferred to 1% glutaraldehyde (v/v) until embedding. Tissue sections (1 mm) were post-fixed in 1% osmium tetroxide and dehydrated before processing into araldite resin blocks. Next, 1 µm microtome-cut sections were stained with a 1% toluidine blue/2% sodium borate solution before bright-field imaging

using a Zeiss Axio microscope. Ultrathin sections (60 nm) were cut from corpus callosum samples, stained with uranyl acetate and lead citrate, and grids imaged on a JEOL transmission electron microscope. Axon diameter, myelin and inner tongue thickness were calculated from a measured area based on assumption of circularity using Fiji/ImageJ (Fiji.sc) (diameter = 2 × √[area/π]), with 100–200 axons per animal analysed.

## Behavioural testing

Experimenters were blinded to genotype during behavioural testing and data analyses. All experiments were performed in a behaviour testing room maintained at a constant temperature of 20 °C. The open-field test was performed on male mice at 4–8 weeks and 11–13 months of age to assess locomotor activity and anxiety-associated behaviours. Handling was carried out 3–4 days before testing. Mice were placed in the open field (47 × 47 cm) to freely explore the arena for 10 min. Equipment was cleaned with 70% ethanol between each test to remove odours. The total ambulatory distance travelled (in metres) and the time spent in the edges (9 cm from the wall) and centre (29 × 29 cm) were automatically quantified using the video tracking software Any-Maze (Stoelting Europe, v.6.3). The Barnes maze test was performed in adult mice 2–4 months of age to assess spatial learning, memory and cognitive flexibility. The maze consisted of one white circular platform with 20 circular holes around the outside edge, 91.5 cm in diameter and 115 cm in height (San Diego Instruments). A dark escape chamber was attached to one of the holes, and the location of the escape chamber remained constant for each mouse but was shifted 90° clockwise between consecutive mice to avoid carryover of olfactory cues. Lamps and overhead lights (450 lux) were used to light the maze. Once the trial started, an aversive white noise stimulus at 85 dB was played until the mouse entered the escape chamber. Visual cues were present on the curtains and walls around the maze. Animals were retained within a white holding cylinder (diameter of 10.5 cm) at the beginning of each trial. The maze and escape chamber were cleaned with ethanol between each trial to avoid carryover of olfactory cues between animals. All trials were recorded using the video-based automated tracking software Any-Maze (Stoelting Europe, v.4.99). Before testing, mice were handled for 3–5 min per day for 6–7 days by the experimenter. Animals were brought into the testing room and placed in the holding cylinder to acclimate to the testing environment for 10 s for 2 days before habituation. Mice were habituated to the maze and escape chamber 1 day before the start of the learning phase, whereby each mouse was placed in the holding cylinder for 10 s then allowed to freely explore the maze with no aversive stimuli for 3 min. Mice were then guided to the escape chamber and retained inside for 1 min. During the learning phase (T1–T6), mice were trained to locate the escape chamber over 6 consecutive days with 2 trials per day (1 h inter-trial interval); data per mouse were averaged per day. If the mouse failed to enter the escape chamber during the 3-min trial period, the experimenter guided it to the chamber. Spatial learning was assessed by the total time taken to locate the escape chamber in each trial (primary latency; defined by the head entering the chamber), and spatial working memory was assessed by the number of errors made before locating the escape chamber (primary errors; defined by the nose deliberately entering a hole with some extension of the head and neck or hindpaws). The total distance travelled and speed during the trials were additionally measured. Exclusion criteria were defined before data analysis as follows: mice must enter a minimum of three quadrants of the maze within two of the first five trials and must enter the escape chamber during the first three trials. One wild-type and one knockout mouse were excluded from analysis owing to refusal to enter the chamber. At 1 h and 3 days following the last trial, probe tests were performed whereby the mouse was allowed to explore the maze for 1 min with the escape chamber removed. The time spent in the target quadrant of the maze and the number of nose pokes in each hole

were recorded to assess memory of the escape chamber location. To assess cognitive flexibility, mice underwent the reversal learning phase (R1–R3), whereby the escape box was moved to 180° from the original location, and measurements were taken as described above. A probe test was also performed 3 days after the final reversal trial. The median age of mice assessed that were trained and untrained were comparable between genotypes at the time of euthanasia for immunofluorescence or electron microscopy analysis: untrained $Fire^{+/+}$ and $Fire^{Δ/Δ}$ mice were 118 days old, trained $Fire^{+/+}$ mice were 119 days old and trained $Fire^{Δ/Δ}$ mice were 120 days old.

## EdU incorporation

EdU was dissolved in the drinking water at 0.2 mg ml$^{-1}$ for a period of 14 days from the end of trial day 1 until the end of the experiment. The water was exchanged every other day, and intake was monitored to assess whether consistent volumes were consumed.

## Microglia depletion in adulthood

The CSF1R inhibitor PLX5622 (Chemgood, C-1521) was formulated into chow at a concentration of 1,200 ppm (Research Diets, D11100404i) and fed to 2-month-old and 5-month-old wild-type ($Fire^{+/+}$) mice for 1 month and euthanized as described above.

## SRI-011381 hydrochloride administration

The TGFβ signalling agonist SRI-011381 hydrochloride (HY-100347A, Cambridge Bioscience/MedChem Express), dissolved in PBS containing DMSO (10%) and PEG300 (40%), was injected intraperitoneally into $Fire^{Δ/Δ}$ mice at a dose of 30 mg kg$^{-1}$ 3 times per week for 3 weeks (9 injections in total) and killed as described above.

## Measuring TGFβ1 levels by ELISA

Following perfusion with PBS, the corpus callosum was dissected from 2 mm coronal sections of $Fire^{Δ/Δ}$ and wild-type brains and snap-frozen in liquid nitrogen. Samples were homogenized in RIPA buffer (Millipore, 20–188) containing phosphatase and protease inhibitors (Sigma-Aldrich, 4906845001 and 11836170001). Corpus callosum lysate samples were activated and TGFβ1 protein levels were measured by ELISA (BioLegend, 436707) according to the manufacturer's instructions for serum and plasma samples at a final dilution of 1:10. BCA assays were performed to measure total protein according to the manufacturer's instructions (Thermo Fisher Scientific, 23225), and values were normalized to this for each sample.

## Brain dissociation and cell sorting for single-cell RNA sequencing

Brains were collected from 6–7-week-old female mice at the same time of day for each animal. Mice were culled by cervical dislocation and brains were dissected, with the olfactory bulbs and cerebellum removed. Hippocampi from both hemispheres and the remainder of the left hemisphere (without the hippocampus) were collected in ice-cold HBSS (without Ca$^{2+}$ and Mg$^{2+}$; 14175-053, Gibco) with 5% trehalose (T0167, Sigma Aldrich) and 30 μM actinomycin D (A1410, Sigma Aldrich) and were finely minced using a 22A scalpel. Brains were digested using the an Adult Brain Dissociation kit (130-107-677, Miltenyi Biotec) with the following modifications: (1) tissues were dissociated as described in the "manual dissociation" section of the Neural Tissue Dissociation kit protocol (130-092-628, Miltenyi Biotec); (2) enzymatic digestions were performed at 35 °C; (3) half the concentration of enzyme P was used; (4) actinomycin D was used to limit dissociation-induced transcriptional changes; (5) 5% trehalose was added in all buffers to increase cellular viability; (6) cell clusters were removed by filtration through pre-moistened 70 μm (352350, Falcon) and 40 μm (352340; Falcon) cell strainers; (7) erythrocyte and myelin debris removal steps were omitted during dissociation steps; and (8) all centrifugations were performed at 200g at 4 °C. After dissociation, cells were collected in

PBS with 0.2% BSA before being sorted on a Sony SH800 cell sorter. Gates were chosen based on forward and side scatter to exclude myelin debris, erythrocytes and doublets. Non-viable cells were excluded based on DRAQ7 and/or DAPI staining (DRAQ7$^{high}$ and/or DAPI$^{high}$ cells were classified as non-viable). After confirming the viability of cells after sorting, using trypan blue and a haemocytometer, single cells were processed through the Chromium Single Cell Platform using a Chromium Next GEM Single Cell 3′ GEM Library and Gel Bead kit (v.3.1 chemistry, PN-1000121, 10x genomics) and a Chromium Next GEM Chip G kit (PN-1000120) and processed following the manufacturer's instructions. Libraries were sequenced using a NovaSeq 6000 sequencing system (PE150 (HiSeq), Illumina).

## Pre-processing of sequencing data and single-cell RNA sequencing analysis

Alignment to the reference genome, feature counting and cell calling were performed following the 10x Genomics CellRanger (v.5.0.0) pipeline, using the default mm10 genome supplied by 10x Genomics (https://cf.10xgenomics.com/supp/cell-exp/refdata-gex-mm10-2020-A.tar.gz). From the output, the filtered matrices were used for downstream analyses. Pre-processing was performed on the University of Edinburgh's compute cluster Eddie. The analysis was performed with R v.4.1.1. Full details to replicate the analysis pipelines described below can be found in code scripts available on GitHub (https://github.com/Anna-Williams/Veronique-Firemice). SingleCellEXperiment v.1.14.1 was used to handle the single-cell experiment objects in R. Cells were filtered using dataset-specific parameters on the basis of genes and unique molecular identifiers (UMIs) per cell, the ratio between these two parameters and the percentage of mitochondrial gene reads per cell. Thresholds were computed with the isOutlier function from scater (v.1.20.1), as batch 6 was of poorer quality than the other batches, with outlier values, the subset argument was used. Only genes that were detected in at least two cells were kept. Using scran (v.1.20.1), the data were normalized by deconvolution, and the top 15% highly variable genes were selected. Following principal component analysis (PCA), 25 principal components (PCs) were kept for downstream analysis (cut-off selected by examination of an Elbowplot). Nonlinear dimensional dimension representation and t-distributed stochastic neighbour embedding (t-SNE) and gene expression variance explained by batch (computed with scater) revealed the need for batch correction. Batch correction was performed using mutual nearest neighbours with fastMNN, batchelor (v.1.8.0). Finally, a graph-based clustering approach was used to cluster the cells using the clusterCells function from scran, with $k = 60$. Clusters with the highest expression of oligodendrocyte markers (*Plp1*, *Mog*, *Mag* and *Mbp*) and that did not express other cell type markers (for example, astrocyte, oligodendrocyte progenitor cell or microglia markers) were subsetted to be analysed separately. Cell and gene quality control were further adjusted, setting a stricter minimum UMI count threshold (5,000 UMIs) and maximum percentage of mitochondrial gene reads per cell (10%). A small cluster of cells of lower quality based on the new thresholds was also excluded from the analysis (Extended Data Fig. 10). Ultimately, we included a total of 19,506 genes and 13,583 cells. The normalization, feature selection, dimensional reduction and batch correction were repeated with the subset dataset as described above. Clustering was performed at different resolutions, and after examination with clustree (v.0.4.4), $k = 100$ was selected and then merged into four clusters (Fig. 6a and Extended Data Fig. 10). Differential gene expression between cluster 1 (specific to the *Fire*$^{Δ/Δ}$ mice) and the mean expression of all other cells was performed using FindMarkers from Seurat (v.4.1.0) (Supplementary Table 1). ShinyCell v.2.1.0 was used to produce an interactive application, and org.Mm.eg.db v.3.13.0 was used to annotate genes. ggplot2 v.3.3.5 was used to perform custom plots, here v.1.0.1 was used to ensure reproducible paths, and Matrix v.1.3.4 was used for handling sparse matrices.

## Lipidomics

Corpus callosum samples from *Fire*$^{+/+}$ and *Fire*$^{Δ/Δ}$ mice were diluted in 700 µl of PBS with 800 µl 1 N HCl:CH$_3$OH 1:8 (v/v), 900 µl ChCl$_3$ and 200 µg ml$^{-1}$ of the antioxidant 2,6-*di*-tert-butyl-4-methylphenol (BHT; Sigma-Aldrich). Splash Lipidomix Mass Spec standard (3 µl; Avanti Polar Lipids) was added into the extract mix. The organic fraction was evaporated at room temperature using a Savant Speedvac spd111v (Thermo Fisher), and the remaining lipid pellet was reconstituted in 100% ethanol. Lipid species were analysed by liquid chromatography electrospray ionization tandem mass spectrometry (LC–ESI–MS/MS) on a Nexera X2 UHPLC system (Shimadzu) coupled with hybrid triple quadrupole/linear ion trap mass spectrometer (6500+ QTRAP system; AB SCIEX). Chromatographic separation was performed on a XBridge amide column (150 × 4.6 mm, 3 × 5 µm; Waters) maintained at 35 °C using mobile phase A (1 mM ammonium acetate in water and acetonitrile 5:95 (v/v)) and mobile phase B (1 mM ammonium acetate in water and acetonitrile 50:50 (v/v)) in the following gradient: 0–6 min: 0% B; 6% B; 6–10 min: 6% B; 25% B; 10–11 min: 25% B; 98% B; 11–13 min: 98% B; 100% B; 13–19 min: 100% B; 19–24 min: 0% B. The flow rate was set at 0.7 ml min$^{-1}$, which was increased to 1.5 ml min$^{-1}$ from 13 min onwards. Sphingomyelin and cholesteryl esters were measured in positive ion mode with a precursor scan of 184.1, 369.4. Triglycerides, diglycerides and monoglycerides were measured in positive ion mode with a neutral loss scan for one of the fatty acyl moieties. Phosphatidylcholine, phosphatidylethanolamine, phosphatidylglycerol, phosphatidylinositides and phosphatidylserines were measured in negative ion mode by fatty acyl fragment ions. Lipid quantification was performed by scheduled multiple reactions monitoring, with the transitions based on the neutral losses or the typical product ions as described above. The instrument parameters were set as follows: curtain gas = 35 psi; collision gas = 8 a.u. (medium); IonSpray 16 voltage = 5,500 V and −4,500 V; temperature = 550 °C; ion source gas 1 = 50 psi; ion source gas 2 = 60 psi; declustering potential = 60 V and −80 V; entrance potential = 10 V and −10 V; and collision cell exit potential = 15 V and −15 V. Peak integration was performed using the Multiquant TM software v.3.0.3. Lipid species signals were corrected for isotopic contributions (calculated using Python Molmass 2019.1.1) and were quantified on the basis of internal standard signals and adhere to the guidelines of the Lipidomics Standards Initiative.

## Immunofluorescence staining of human tissue

Post-mortem tissue from individuals with ALSP and from unaffected individuals who had died of non-neurological causes (Extended Data Table 1) were obtained with full ethical approval from the Queen Square Brain Bank for Neurological Disorders, UCL Queen Square Institute of Neurology, and their use was in accord with the terms of the informed consents for use of donor tissue and information. Ethical approval was granted to the Queen Square Brain Bank for the use of tissue by the National Health Service Health Research Authority through the London Central Research Ethics Committee. Diagnosis of ALSP was confirmed on the basis of mutations in *CSF1R* and by neuropathological means, and clinical history was provided by Z. Jaunmuktane (University College London). Formalin-fixed paraffin embedded tissue blocks were cut to 10 µm thickness. Sections were placed in the oven at 60 °C for 10 min and deparaffinized by a series of washes in HistoClear (2× 10 min) and ethanol (100% twice, 95%, 70%, 50%; 5 min each). Following washes in PBS, slides were placed in a pressure cooker in Vector unmasking solution for 20 min, cooled, washed once and blocked for 1 h with 5% normal horse serum (Gibco) and 0.3% Triton X-100 (Fisher Scientific) in PBS. Tissue was incubated with primary antibodies in a humid chamber overnight. Sections were then washed in PBS, and fluorescently conjugated secondary antibodies were applied for 2 h at room temperature in a humid chamber (1:500, Life Technologies–Molecular Probes). Following washes in PBS and then water, the sections were counterstained

with Hoechst and washed with distilled water for 20 min. Slides were then coverslipped with Fluoromount-G (Southern Biotech). Primary antibodies included IBA1 (Abcam, 1:500; ab5076) and LYVE1 (Abcam, 1:100; ab14917). Entire tissue sections were imaged using a Zeiss Axi-oScan Z.1 SlideScanner and Zeiss Zen2 software (blue edition). Three fields of $150 \times 150\ \mu m$ were counted per region of interest in each case and counts were multiplied to determine the density of immunopositive cells per $mm^2$.

### Electron microscopy of human tissue
ALSP tissue was acquired from the Department of Neuropathology at Charité-Universitätsmedizin Berlin, and tissue from unaffected individuals were obtained from the Medical Research Council Edinburgh Brain and Tissue Bank, approved by the respective ethical review boards. ALSP tissue samples of the white matter were fixed in 2.5% glutaraldehyde in 0.1 M sodium cacodylate buffer for 48 h at 4 °C. Samples were post-fixed in 1% osmium tetroxide in 0.05 M sodium cacodylate buffer for 3 h, dehydrated in graded acetone series including en bloc staining with 1% uranyl acetate and 0.1% phosphotungstic acid in the 70% acetone step for 60 min and embedded in araldite resin. For tissue from unaffected individuals, ultrathin sections were stained with uranyl acetate and lead citrate and imaged with a Zeiss P902 electron microscope. For ALSP tissue, ultrathin sections with virtual absence of limiting artefacts were prepared and entirely digitized using a Zeiss Gemini 300 scanning electron microscope with a scanning transmission electron microscopy. In brief, we used 29 kV acceleration voltage, 5 nm pixel size and 1.5 µs beam dwell time for digitization and Fiji/TrakEM2 for stitching to allow for in-depth analysis with QuPath 0.3.0.

### Statistics and reproducibility
All manual cell counts were performed in a blinded manner. Data are presented as the mean ± s.e.m. All micrographs are representative images of the result, which were independently repeated a minimum of three times; exact $n$ values are presented in the corresponding graphs of quantification. Before statistical testing, data were assessed for normality of distribution using Shapiro–Wilk test. Statistical tests included two-tailed Student's $t$-test for normally distributed data or Mann–Whitney test for nonparametric data, two-way analysis of variance (ANOVA) with Sidak's multiple comparisons test for comparing more than two groups, and a one-sample $t$-test for comparison of $log_2$(fold change) to a value of 0. For proportion graphs, one-way ANOVA with Tukey's multiple comparisons test was used. For Barnes maze analysis of primary latency and errors, repeated measures two-way ANOVA with Sidak's multiple comparisons test was used. Before testing for inter-group differences in the probe test data, each genotype was tested against chance (25%) using a one sample $t$-test. Slopes of myelin and inner tongue thickness versus axon diameter were compared using simple linear regression analysis, and for comparison of more than two groups, Kruskal–Wallis with Dunn's multiple comparisons test. A $P$ value of <0.05 was considered significant. Data handling and statistical processing were performed using Microsoft Excel and GraphPad Prism Software.

### Reporting summary
Further information on research design is available in the Nature Portfolio Reporting Summary linked to this article.

### Data availability
Raw single-cell RNA sequence datasets have been deposited in the Gene Expression Omnibus with accession code GSE215440. Analysed oligodendrocyte sequencing data are perusable on the following shiny application: https://annawilliams.shinyapps.io/shinyApp_oligos_VM/.

Full details to replicate the analysis pipelines can be found in code scripts available on GitHub (https://github.com/Anna-Williams/Veronique-Firemice). Alignment to the reference genome, feature counting and cell calling were performed following the 10x Genomics CellRanger (v.5.0.0) pipeline, using the default mm10 genome supplied by 10x Genomics (https://cf.10xgenomics.com/supp/cell-exp/refdata-gex-mm10-2020-A.tar.gz). Source data are provided with this paper.

### Code availability
Full details to replicate the analysis pipelines described below can be found in code scripts available on GitHub (https://github.com/Anna-Williams/Veronique-Firemice).

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

**Acknowledgements** We acknowledge staff at the Queen Square Brain Bank for Neurological Disorders, University College London Queen Square Institute of Neurology (Z. Jaunmuktane) for providing ALSP tissue and tissue from unaffected individuals for immunofluorescence analysis; staff at the School of Biological Sciences electron microscopy unit (Wellcome Multi-User Equipment Grant WT104915MA); staff at the Core Facility for Electron Microscopy of the Charité; M. Mohammad for support in data acquisition; and A. Corsinotti and F. Rossi at the CRM Flow Cytometry Core Facility (University of Edinburgh) for their assistance with cell processing for single-cell RNA sequencing. The Queen Square Brain Bank is supported by the Reta Lila Weston Institute of Neurological Studies, University College London Queen Square Institute of Neurology. This work was supported by the Wellcome Trust-funded four-year Ph.D. programme in Tissue Repair at The University of Edinburgh (to N.B.M., 108906/Z/15/Z), a Career Development Award from the Medical Research Council and the United Kingdom Multiple Sclerosis Society (to V.E.M., MR/M020827/1), a Senior Non-Clinical Research Fellowship from the Medical Research Council (to V.E.M., MR/V031260/1), the John David Eaton Chair in Multiple Sclerosis Research at the Barlo Multiple Sclerosis Centre (to V.E.M.), the Medical Research Council Centre for Reproductive Health (MR/N02256/1), the Simons Initiative for the Developing Brain (to C.P.), the United Kingdom Dementia Research Institute (UK DRI) (which receives its funding from UK DRI Ltd, the United Kingdom Medical Research Council, the Alzheimer's Society and Alzheimer's Research United Kingdom) (to J.P., ARUK-PG2016B-6), Alzheimer's Research United Kingdom (to K.H., ARUK-PG2016B-6), the European Research Council (ERC) under the European Union's Horizon 2020 research and innovation programme (to T.L.S.-J. number 681181), a Grant-in-Aid of Scientific Research (B) from the Japan Society for the Promotion of Sciences (to R.M., number 22H02962), the Research Foundation of Flanders (FWO Vlaanderen, to J.F.J.B. and J.J.A.H., numbers 1S15519N, G099618FWO and 12J9119N), Interreg V-A EMR (EURLIPIDS, to J.F.J.B. and J.J.A.H., number EMR23), the Belgian Charcot Foundation (to J.F.J.B. and J.J.A.H.), and a MRC/MS Society UK Award (to A.W. and N.B.-C., number MR/T015594/1). Diagrams were created using BioRender.com under a publication licence. For the purpose of open access, the author has applied a Creative Commons Attribution (CC BY) licence to any Author Accepted Manuscript version arising from this submission.

**Author contributions** N.B.M. designed the study, performed experiments, analysed and interpreted the data and co-wrote the manuscript. D.A.D.M. processed tissue for single-cell RNA sequencing, performed validation studies and a subset of behavioural analysis. N.B.-C. and A.W. performed bioinformatics analysis of single-cell RNA sequencing data and created the shiny application. A.U. and R.M. generated the *Tgfbr1* conditional knockout mouse model. J.F.J.B. and J.J.A.H. performed the lipidomics analysis. A.H., R.K.H., L.A., S.S., B.W.M. and M.P. assisted in the mouse experiments. I.M.-G. performed analysis on astrocytes. K.E.A. and K.H. supported cognitive task training, data analysis and interpretation. S.M., J.M., A.W., C.D., W.S., J.R. and T.L.S.-J. processed tissue and assisted in electron microscopy. W.M. and M.D. performed perfusions and brain extractions. C.P. generated the *Fire$^{Δ/Δ}$* mice, co-supervised the mouse project and contributed to study direction. A.W. and J.P. assisted in single-cell RNA sequencing interpretation, provided guidance and supervision of the study and participated in manuscript editing. V.E.M. supervised the study, guided experimental design and interpretation and co-wrote the manuscript.

**Competing interests** The authors declare no competing interests.

**Additional information**
**Correspondence and requests for materials** should be addressed to Veronique E. Miron.

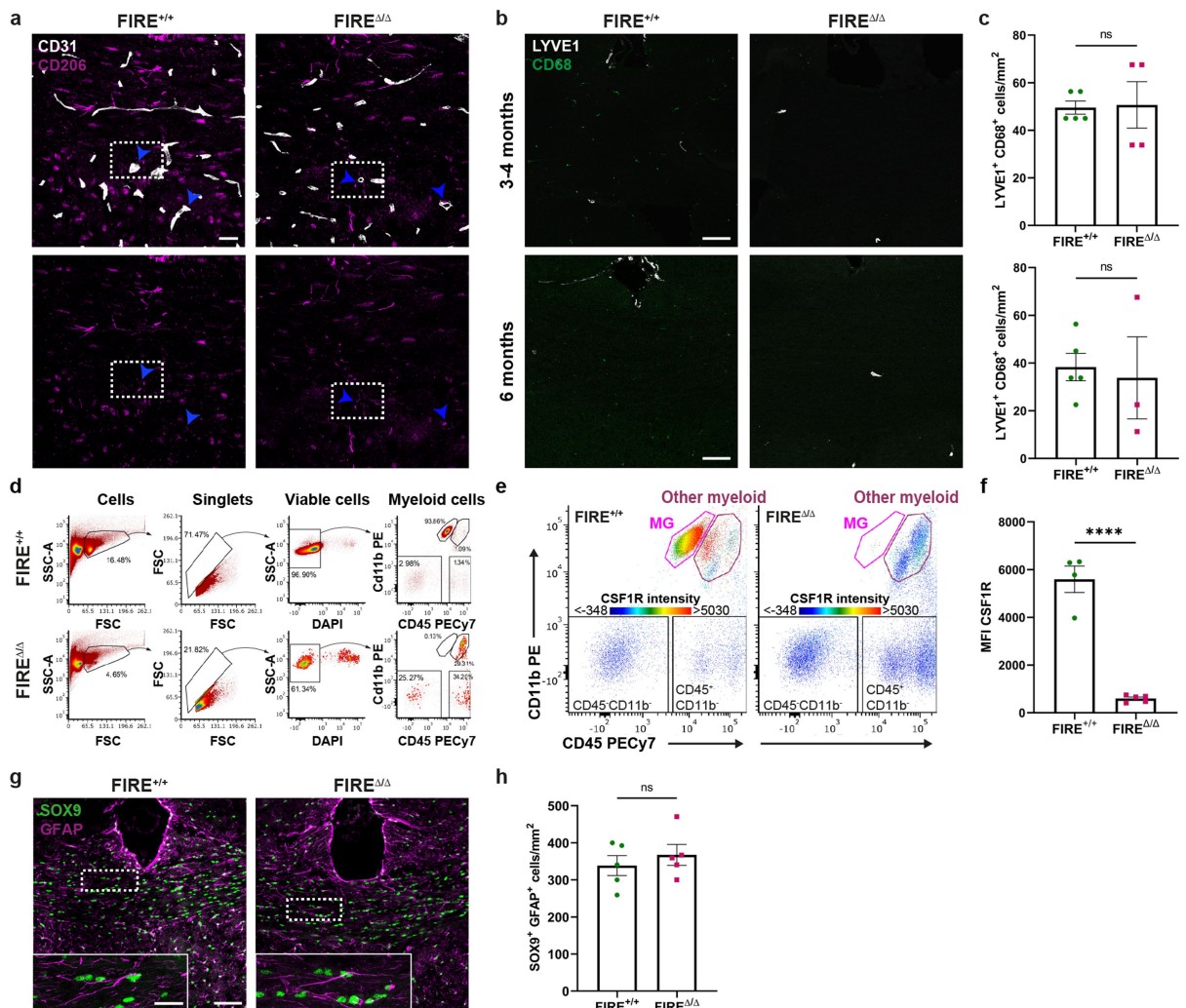

**Extended Data Fig. 1 | Characterisation of FIRE<sup>Δ/Δ</sup> mouse. a)** Images of perivascular macrophages (CD206+; magenta) in association with blood vessels (CD31+; white)(arrows) in FIRE⁺/⁺ and FIREᐞ/ᐞ mouse corpus callosum at 1 month of age. Scale bar, 25 µm. **b)** Images of perivascular macrophages (LYVE1+ CD68+) at 3-4 months and 6 months of age in corpus callosum. Scale bar, 75 µm. **c)** Mean density of LYVE1+ CD68+ cells/mm² ± s.e.m. at 3-4 months and 6 months of age. Non-significant, P > 0.9999 and 0.7698, Mann Whitney and 2-tailed unpaired Student's t-test, respectively. At 3-4 months, n = 5 FIRE⁺/⁺ and 4 FIREᐞ/ᐞ mice, at 6 months, n = 5 FIRE⁺/⁺ and 3 FIREᐞ/ᐞ mice. **d)** Flow cytometry gating strategy for assessment of non-microglial myeloid cells (including BAMs; CD11b⁺ CD45ʰⁱ) for panels (**e**) and (**f**). **e)** Intensity of expression of CSF1R in FIRE⁺/⁺ and FIREᐞ/ᐞ mice in CD11b⁺CD45ˡᵒ microglia (MG) versus CD11b⁺CD45ʰⁱ myeloid cells. **f)** Mean Fluorescence Intensity (MFI) of CSF1R ± s.e.m. in non-microglial myeloid cells (including BAMs) in FIRE⁺/⁺ and FIREᐞ/ᐞ mice. ****P < 0.0001, 2-tailed unpaired Student's t-test. n = 4 FIRE⁺/⁺ and 5 FIREᐞ/ᐞ mice **g)** Images of astrocytes (SOX9; green; GFAP; magenta) at 1 month of age in the corpus callosum. Scale bar, 75 µm. Inset shows magnified view of double positive cells. Scale bar, 25 µm. **h)** Mean SOX9+ GFAP+ cells/mm² ± s.e.m. at 1 month of age in the corpus callosum. n = 5 mice per group. Non-significant, P = 0.4799, 2-tailed unpaired Student's t-test.

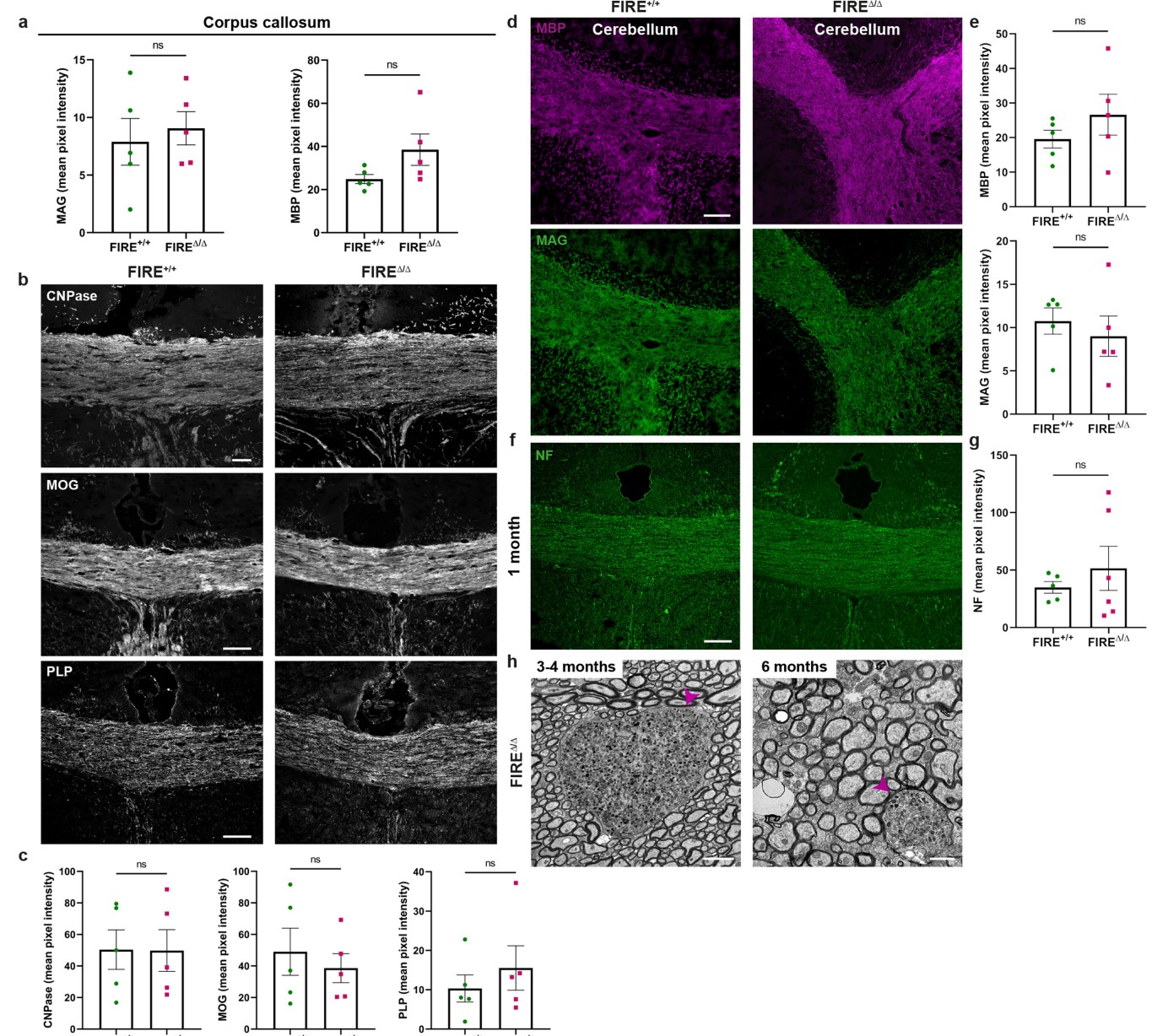

**Extended Data Fig. 2 | Myelin protein and axonal assessment in the FIRE$^{\Delta/\Delta}$ mouse. a**) Mean pixel intensity of MAG and MBP ± s.e.m. in corpus callosum at 1 month of age. Non-significant, $P = 0.6482$ and 0.1097 respectively, 2-tailed unpaired Student's t-test. n = 5 mice/group. **b**) Images of CNPase, MOG, and PLP (white) in corpus callosum at 1 month of age. Scale bar, 75 µm. **c**) Mean pixel intensity of CNPase, MOG, and PLP ± s.e.m. in corpus callosum at 1 month of age. Non-significant, $P = 0.9758$, 0.5699 and 0.4551 respectively, 2-tailed unpaired Student's t-test. n = 5 mice/group. **d**) Images of MBP (magenta) and MAG (green) in cerebellum at 1 month of age. Scale bar, 75 µm. **e**) Mean pixel intensity of MBP and MAG ± s.e.m. in cerebellum at 1 month of age. Non-significant, $P = 0.3067$ and 0.5462 respectively, 2-tailed unpaired Student's t-test. n = 5 mice/group. **f**) Images of neurofilament (NF; green) in corpus callosum at 1 month of age. Scale bar, 75 µm. **g**) Mean pixel intensity of NF ± s.e.m. in corpus callosum at 1 month of age. Non-significant, $P = 0.4623$, 2-tailed unpaired Student's t-test. n = 5 FIRE$^{+/+}$ and 6 FIRE$^{\Delta/\Delta}$. **h**) Images of axonal spheroids in corpus callosum of FIRE$^{\Delta/\Delta}$ mice at 3-4 months and 6 months, indicated with arrows. Scale bar, 2 µm and 1 µm, respectively.

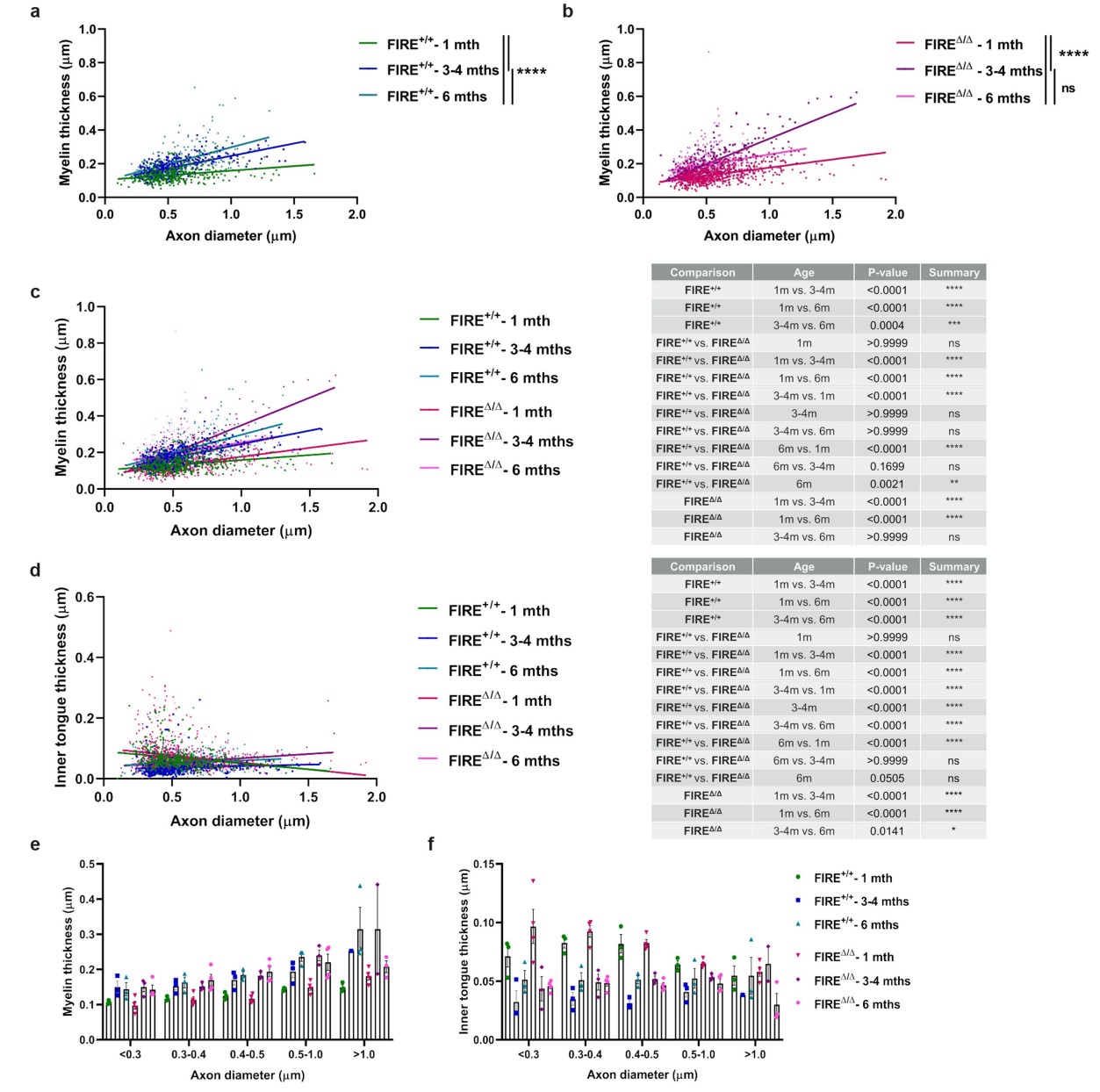

**Extended Data Fig. 3 | Comparison of myelin across time points. a)** Myelin thickness/axon diameter (μm) in FIRE^+/+ mice at 1, 3-4, and 6 months of age. ****P < 0.0001, Kruskal-Wallis with Dunn's multiple comparisons test. n = 3 mice/group. **b)** Myelin thickness/axon diameter (μm) in FIRE^Δ/Δ mice at 1, 3-4, and 6 months of age. ****P < 0.0001, ns P = 0.9232, Kruskal-Wallis with Dunn's multiple comparisons test. 1 month: n = 3 FIRE^+/+ and 4 FIRE^Δ/Δ; 3-4 months: n = 3 mice/group; 6 months: n = 3 FIRE^+/+ and 4 FIRE^Δ/Δ. **c)** Myelin thickness/axon diameter (μm) in FIRE^+/+ and FIRE^Δ/Δ mice at 1, 3-4, and 6 months of age. Table of P values from Kruskal-Wallis with Dunn's multiple comparisons test, 1 month: n = 3 FIRE^+/+ and 4 FIRE^Δ/Δ mice; 3-4 months: n = 3 mice/group; 6 months: n = 3

FIRE^+/+ and 4 FIRE^Δ/Δ. **d)** Inner tongue thickness/axon diameter (μm) in FIRE^+/+ and FIRE^Δ/Δ mice at 1, 3-4, and 6 months of age. Table of P values from Kruskal-Wallis with Dunn's multiple comparisons test, 1 month: n = 3 FIRE^+/+ and 4 FIRE^Δ/Δ mice; 3-4 months: n = 3 mice/group; 6 months: n = 3 FIRE^+/+ and 4 FIRE^Δ/Δ. **e)** Mean myelin thickness (μm)/axon diameter bin ± s.e.m. in FIRE^+/+ and FIRE^Δ/Δ mice at 1, 3-4, and 6 months of age. 1 month: n = 3 FIRE^+/+ and 4 FIRE^Δ/Δ mice; 3-4 months: n = 3 mice/group; 6 months: n = 3 FIRE^+/+ and 4 FIRE^Δ/Δ. **f)** Mean inner tongue thickness (μm)/axon diameter bin ± s.e.m. in FIRE^+/+ and FIRE^Δ/Δ mice at 1, 3-4, and 6 months of age. 1 month: n = 3 FIRE^+/+ and 4 FIRE^Δ/Δ mice; 3-4 months: n = 3 mice/group; 6 months: n = 3 FIRE^+/+ and 4 FIRE^Δ/Δ.

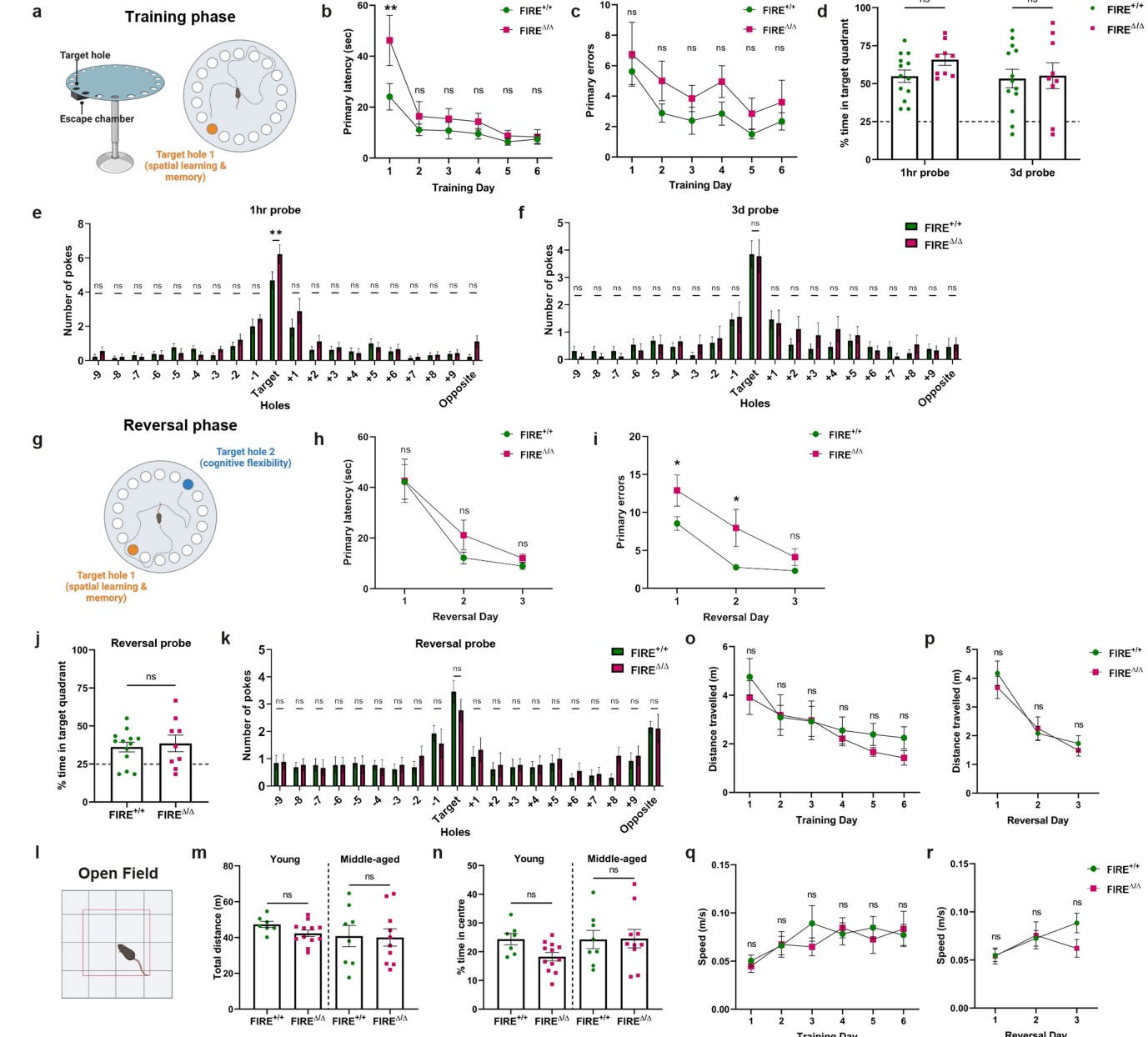

**Extended Data Fig. 4** | See next page for caption.

**Extended Data Fig. 4 | Learning and memory encoding unimpaired, but cognitive flexibility is compromised, in the FIRE$^{\Delta/\Delta}$ mouse. a**) Training phase of Barnes maze: mice learn to locate Target hole 1 (orange) with underlying escape chamber in spatial learning and memory retrieval trials. Mice tested were 2–4 months of age, with a median age of 119 days old (FIRE$^{+/+}$) and 120 days old (FIRE$^{\Delta/\Delta}$). **b**) Mean primary latency (sec) ± s.e.m. over 6 training days. n = 13 FIRE$^{+/+}$ mice and 10 FIRE$^{\Delta/\Delta}$. Non-significant between genotypes across all training days, Day 1: **$P$ = 0.0011; Day 2: $P$ = 0.9787; Day 3: $P$ = 0.9627; Day 4: $P$ = 0.9574; Day 5: $P$ = 0.9990; Day 6: $P$ > 0.9999, Repeated measures 2-way ANOVA with Sidak's multiple comparisons test. **c**) Mean primary errors ± s.e.m. during training phase. n = 13 FIRE$^{+/+}$ mice and 10 FIRE$^{\Delta/\Delta}$ mice. Non-significant, Day 1: $P$ = 0.9649; Day 2: $P$ = 0.5968; Day 3: $P$ = 0.8884; Day 4: $P$ = 0.6028; Day 5: $P$ = 0.9215; Day 6: $P$ = 0.9437, Repeated measures 2-way ANOVA with Sidak's multiple comparisons test. **d**) Mean percentage of time spent in the target quadrant ± s.e.m., n = 13 FIRE$^{+/+}$ and 10 FIRE$^{\Delta/\Delta}$. Dotted line indicates 25% chance. Non-significant between genotypes, 1hr: $P$ = 0.7337; 3d: $P$ > 0.9999, 2-way ANOVA with Sidak's multiple comparisons test. **e**) Mean number of nose pokes into holes during 1hr probe test ± s.e.m., n = 13 FIRE$^{+/+}$ mice and 9 FIRE$^{\Delta/\Delta}$ mice. Target **$P$ = 0.0019, 2-way ANOVA with Sidak's multiple comparisons test. **f**) Mean number of nose pokes into holes during 3d probe test ± s.e.m., n = 13 FIRE$^{+/+}$ mice and 9 FIRE$^{\Delta/\Delta}$ mice. Non-significant, $P$ = 0.9329, 2-way ANOVA with Sidak's multiple comparisons test. **g**) Reversal phase: Target hole 2 (blue) is 180° from the original target; mice require cognitive flexibility to adapt to the new target. **h**) Mean primary latency (sec) ± s.e.m. over 3 training days. n = 13 FIRE$^{+/+}$ and 9 FIRE$^{\Delta/\Delta}$. Non-significant across all training days between genotypes, Day 1: $P$ = 0.9999; Day 2: $P$ = 0.5186; Day 3: $P$ = 0.9622, Repeated measures 2-way ANOVA with Sidak's multiple comparisons test. **i**) Mean primary errors ± s.e.m. during reversal days. n = 13 FIRE$^{+/+}$ and 9 FIRE$^{\Delta/\Delta}$. Day 1: *$P$ = 0.0488, Day 2: *$P$ = 0.0142, Day 3: $P$ = 0.6727, Repeated measures 2-way ANOVA with Sidak's multiple comparisons test. **j**) Mean percentage of time spent in the target quadrant during reversal probe test ± s.e.m. n = 13 FIRE$^{+/+}$ mice and 9 FIRE$^{\Delta/\Delta}$ mice. Dotted line indicates 25% chance. Non-significant, $P$ = 0.6933, 2-tailed unpaired Student's t-test. **k**) Mean number of nose pokes into holes during reversal probe test ± s.e.m. n = 13 FIRE$^{+/+}$ mice and 9 FIRE$^{\Delta/\Delta}$ mice. Non-significant $P$ = 0.9657, 2-way ANOVA with Sidak's multiple comparisons test. **l**) Schematic of Open Field test. **m**) Total distance travelled (m) during 10-min Open Field test ± s.e.m. n = 37 mice: n = 15 FIRE$^{+/+}$ mice (7 young, 8 middle-aged); n = 22 FIRE$^{\Delta/\Delta}$ mice (12 young, 10 middle-aged). Young mice = 4–8 weeks of age, Middle aged mice = 11–13 months of age. Non-significant, Young: $P$ = 0.9997; Middle-aged: $P$ > 0.999, 2-way ANOVA. **n**) Percentage (%) of time spent in the centre of the arena during Open Field test ± s.e.m. n = 37 mice: n = 15 FIRE$^{+/+}$ mice (7 young, 8 middle-aged); n = 22 FIRE$^{\Delta/\Delta}$ mice (12 young, 10 middle-aged). Young mice = 4–8 weeks of age, Middle aged mice = 11–13 months of age. Non-significant, Young: $P$ = 0.7899; Middle-aged: $P$ > 0.9995, 2-way ANOVA. **o**) Mean distance travelled (m) during training days ± s.e.m. n = 13 FIRE$^{+/+}$ mice and 9 FIRE$^{\Delta/\Delta}$ mice. Non-significant, $P$ = 0.9607, >0.9999, >0.9999, 0.9955, 0.6583, 0.6034, Repeated measures 2-way ANOVA with Sidak's multiple comparisons test. **p**) Mean distance travelled (m) during reversal days ± s.e.m. n = 13 FIRE$^{+/+}$ mice and 9 FIRE$^{\Delta/\Delta}$ mice. Non-significant, $P$ = 0.7856, 0.9784, 0.8626, Repeated measures 2-way ANOVA with Sidak's multiple comparisons test. **q**) Mean speed (m/s) during training days ± s.e.m. n = 13 FIRE$^{+/+}$ mice and 9 FIRE$^{\Delta/\Delta}$ mice. Non-significant, $P$ = 0.9998, >0.9999, 0.6667, 0.9995, 0.9831, 0.9995, Repeated measures 2-way ANOVA with Sidak's multiple comparisons test. **r**) Mean speed (m/s) during reversal days ± s.e.m. n = 13 FIRE$^{+/+}$ mice and 9 FIRE$^{\Delta/\Delta}$ mice. Non-significant, $P$ > 0.9999, 0.9940, 0.1561, Repeated measures 2-way ANOVA with Sidak's multiple comparisons test.

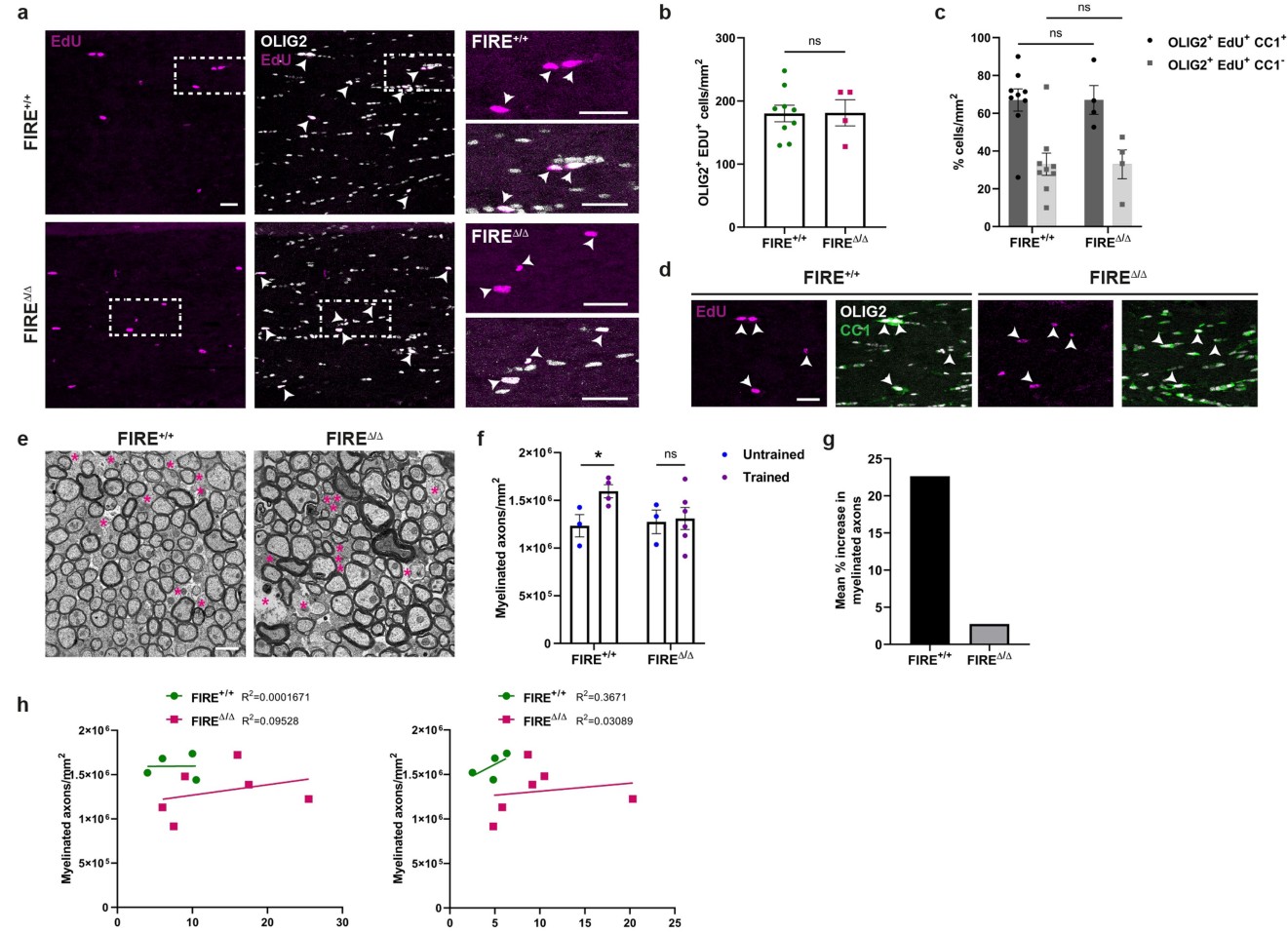

**Extended Data Fig. 5 | Post-learning oligodendrogenesis and myelination.**
**a**) Representative images of EdU (magenta) and OLIG2 (white) double positive cells (arrows), with magnified view of dotted outline. Scale bars, 25 μm. **b**) Mean OLIG2+ EdU+ cells/mm$^2$ ± s.e.m. n = 9 FIRE$^{+/+}$ mice and 4 FIRE$^{Δ/Δ}$ mice. Non-significant, P = 0.9696, 2-tailed unpaired Student's t-test. **c**) Mean proportion of OLIG2+ EdU+ cells which are mature oligodendrocytes (CC1+; black) or immature lineage cells (CC1-; grey) ± s.e.m. n = 9 FIRE$^{+/+}$ mice and 4 FIRE$^{Δ/Δ}$ mice. Non-significant, CC1+: P > 0.9999; CC1-: P > 0.9999, 1-way ANOVA with Tukey's multiple comparisons test. **d**) Representative images of EdU+ (magenta) CC1+ (green) OLIG2+ (white) triple positive cells (arrows). Scale bar, 25 μm. **e**) Images

of corpus callosum 6 weeks post cognitive testing. Scale bar, 1 μm. **f**) Mean number of myelinated axons/mm$^2$ ± s.e.m. in untrained versus trained mice. n = 3 mice/group in untrained category, 4 trained FIRE$^{+/+}$ mice and 6 trained FIRE$^{Δ/Δ}$ mice. FIRE$^{+/+}$ mice *P = 0.0364, FIRE$^{Δ/Δ}$ mice non-significant, P = 0.8537, 2-tailed unpaired Student's t-test. **g**) Mean percentage increase in myelinated axons in trained FIRE$^{+/+}$ and FIRE$^{Δ/Δ}$ mice. n = 3 mice per group in untrained category, 4 trained FIRE$^{+/+}$ mice and 6 trained FIRE$^{Δ/Δ}$ mice. **h**) Correlation between mean myelinated axons per mm$^2$ and reversal day 1 (RD1) primary errors or average primary errors in FIRE$^{+/+}$ and FIRE$^{Δ/Δ}$ mice. Each data point represents an individual mouse. n = 4 FIRE$^{+/+}$ and 6 trained FIRE$^{Δ/Δ}$ mice.

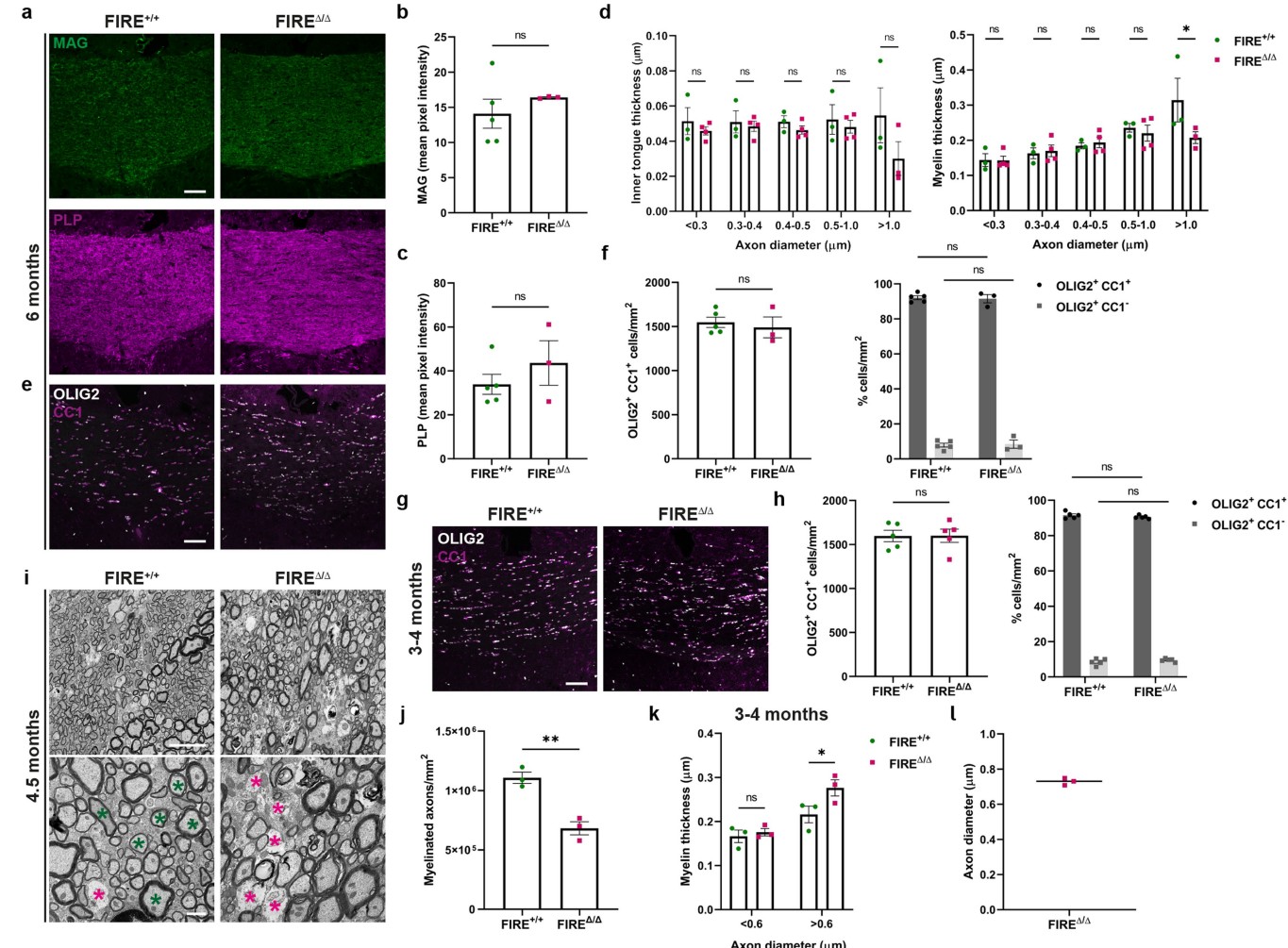

**Extended Data Fig. 6 | Demyelination in FIRE$^{\Delta/\Delta}$ mice. a)** Images of MAG (green) and PLP (magenta) at 6 months of age. Scale bar, 75 µm. **b)** Mean pixel intensity of MAG ± s.e.m. in corpus callosum at 6 months. Non-significant, $P = 0.4335$, 2-tailed unpaired Student's t-test. n = 5 FIRE$^{+/+}$ and 3 FIRE$^{\Delta/\Delta}$. **c)** Mean pixel intensity of PLP ± s.e.m. in corpus callosum at 6 months. Non-significant, $P = 0.3471$, 2-tailed unpaired Student's t-test. n = 5 FIRE$^{+/+}$ mice and 3 FIRE$^{\Delta/\Delta}$ mice. **d)** Mean inner tongue thickness (µm)/axon diameter bin ± s.e.m. at 6 months of age. n = 3 FIRE$^{+/+}$ and 4 FIRE$^{\Delta/\Delta}$ mice. Non-significant, $P = 0.4837$, 2-tailed unpaired Student's t-test. Mean myelin thickness (µm)/axon diameter bin ± s.e.m. at 6 months. n = 3 FIRE$^{+/+}$ and 4 FIRE$^{\Delta/\Delta}$ mice. >1 µm: *$P = 0.0263$, 2-tailed unpaired Student's t-test. **e)** Images of mature oligodendrocytes co-expressing OLIG2 (white) and CC1 (magenta) at 6 months of age. Scale bar, 75 µm. **f)** Mean OLIG2+ CC1+ cells/mm$^2$ ± s.e.m. n = 5 FIRE$^{+/+}$ and 3 FIRE$^{\Delta/\Delta}$. Non-significant, $P = 0.6400$, 2-tailed unpaired Student's t-test. Mean proportion of cells of the oligodendrocyte lineage (OLIG2+) which are mature (CC1+; black) or immature (CC1-; grey) ± s.e.m. n = 5 FIRE$^{+/+}$ and 3 FIRE$^{\Delta/\Delta}$. Non-significant, CC1+: $P = 0.9938$; CC1-: ns $P = 0.9938$, 1-way ANOVA with

Tukey's multiple comparisons test. **g)** Images of mature oligodendrocytes co-expressing OLIG2 (white) and CC1 (magenta) at 3-4 months of age. Scale bar, 75 µm. **h)** Mean OLIG2+ CC1+ cells/mm$^2$ ± s.e.m. n = 5 mice/group. Non-significant, $P = 0.9825$, 2-tailed unpaired Student's t-test. Mean proportion of cells of the oligodendrocyte lineage (OLIG2+) which are mature (CC1+; black) or immature (CC1-; grey) ± s.e.m. n = 5 mice per group. Non-significant, CC1+: $P = 0.7076$; CC1-: $P = 0.7076$, 1-way ANOVA with Tukey's multiple comparisons test. **i)** Images of FIRE$^{+/+}$ and FIRE$^{\Delta/\Delta}$ mouse corpus callosum at 4.5 months of age indicating onset of demyelination in the latter (magenta asterisks), examples of myelinated axons of medium-large calibre indicated by green asterisks. Scale bars, 5 µm and 1 µm. **j)** Mean number of myelinated axons/mm$^2$ ± s.e.m. n = 3 mice/group. **$P = 0.0042$, 2-tailed unpaired Student's t-test. **k)** Mean myelin thickness per small (<0.6 µm) or medium-large (>0.6 µm) axon diameter bin ± s.e.m. at 3-4 months of age. n = 3 mice/group. <0.6 µm: non-significant, $P = 0.8978$; >0.6 µm: *$P = 0.0491$, 2-way ANOVA with Sidak's multiple comparisons test. **l)** Mean of diameters (0.7290 µm) of demyelinated axons per representative image ± s.e.m. in FIRE$^{\Delta/\Delta}$ mice. n = 3 mice.

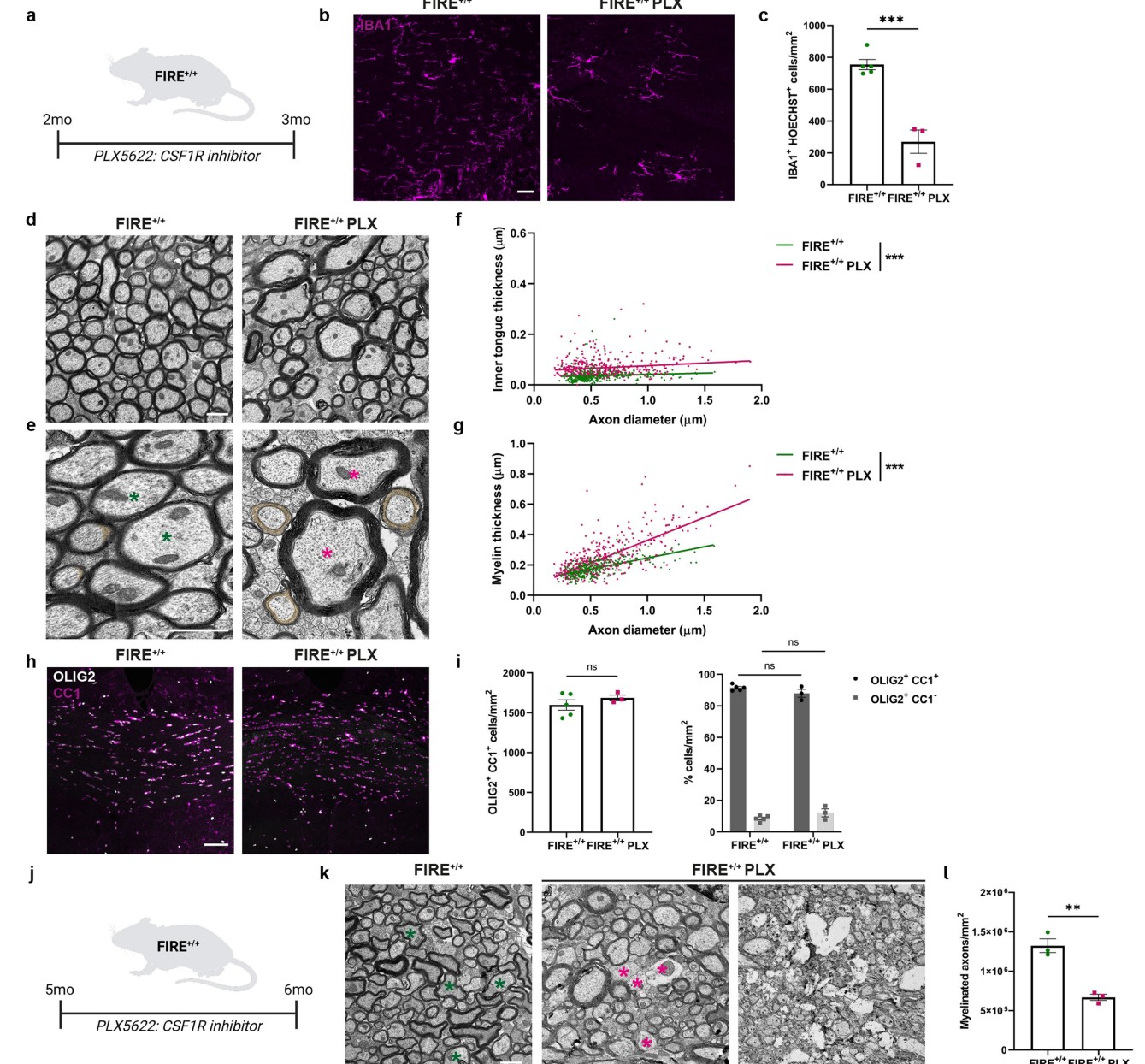

**Extended Data Fig. 7 | Microglia depletion in adulthood causes hypermyelination and demyelination. a)** Adult FIRE[+/+] mice were fed the CSF1R inhibitor PLX5622 in the diet from 2 to 3 months of age. **b)** Representative images of IBA1+ cells (magenta) in the corpus callosum of 3-month old FIRE[+/+] mice on normal diet versus PLX5622 (PLX) diet from 2 to 3 months of age. Scale bar, 25 μm. **c)** Density of IBA1+ cells/mm² ± s.e.m. in FIRE[+/+] mice on normal diet versus PLX diet at 3 months of age. n = 5 mice on normal diet and 3 mice on PLX diet, ***P = 0.0004, 2-tailed unpaired Student's t-test, **d)** Representative images of hypermyelination in the corpus callosum following PLX administration from 2 to 3 months of age. Scale bar, 1 μm. **e)** Representative images of enlarged inner tongues (orange) and thicker myelin (asterisks) following PLX administration from 2 to 3 months of age. Scale bar, 1 μm. **f)** Inner tongue thickness (μm) versus axon diameter in FIRE[+/+] mice on normal diet or PLX diet from 2 to 3 months of age. n = 100 axons/mouse. n = 3 mice on normal diet and 4 mice on PLX diet. ***P < 0.0001, simple linear regression of intercepts. **g)** Myelin thickness (μm) versus axon diameter in FIRE[+/+] mice on normal diet or PLX diet from 2 to

3 months of age. n = 100 axons per mouse. n = 5 mice on normal diet and 3 mice on PLX diet. ***P < 0.0001, simple linear regression of slopes. **h)** Images of mature oligodendrocytes co-expressing OLIG2 (white) and CC1 (magenta) in FIRE[+/+] mice on normal diet or PLX diet from 2 to 3 months of age. Scale bar, 75 μm. **i)** Mean OLIG2+ CC1+ cells per mm² ± s.e.m. in FIRE[+/+] mice on normal diet or PLX diet from 2 to 3 months of age. Non-significant, P = 0.3645, 2-tailed unpaired Student's t-test. Mean proportion of cells of the oligodendrocyte lineage (OLIG2+) which are mature (CC1+; black) or immature (CC1-; grey) ± s.e.m at 3 months. Non-significant, CC1+: P = 0.3779; CC1-: P = 0.3779, 1-way ANOVA with Tukey's multiple comparisons test. n = 5 mice on normal diet and 3 mice on PLX diet. **j)** Adult FIRE[+/+] mice were fed the CSF1R inhibitor PLX5622 in the diet from 5 to 6 months of age. **k)** Images of demyelination in the corpus callosum of 6-month-old FIRE[+/+] mice (asterisks) following PLX administration from 5 to 6-months of age. Scale bar, 1 μm. **l)** Mean number of myelinated axons ± s.e.m. in of FIRE[+/+] mice on normal diet versus PLX diet from 5 to 6-months of age. n = 3 mice/group. **P = 0.0024, 2-tailed unpaired Student's t-test.

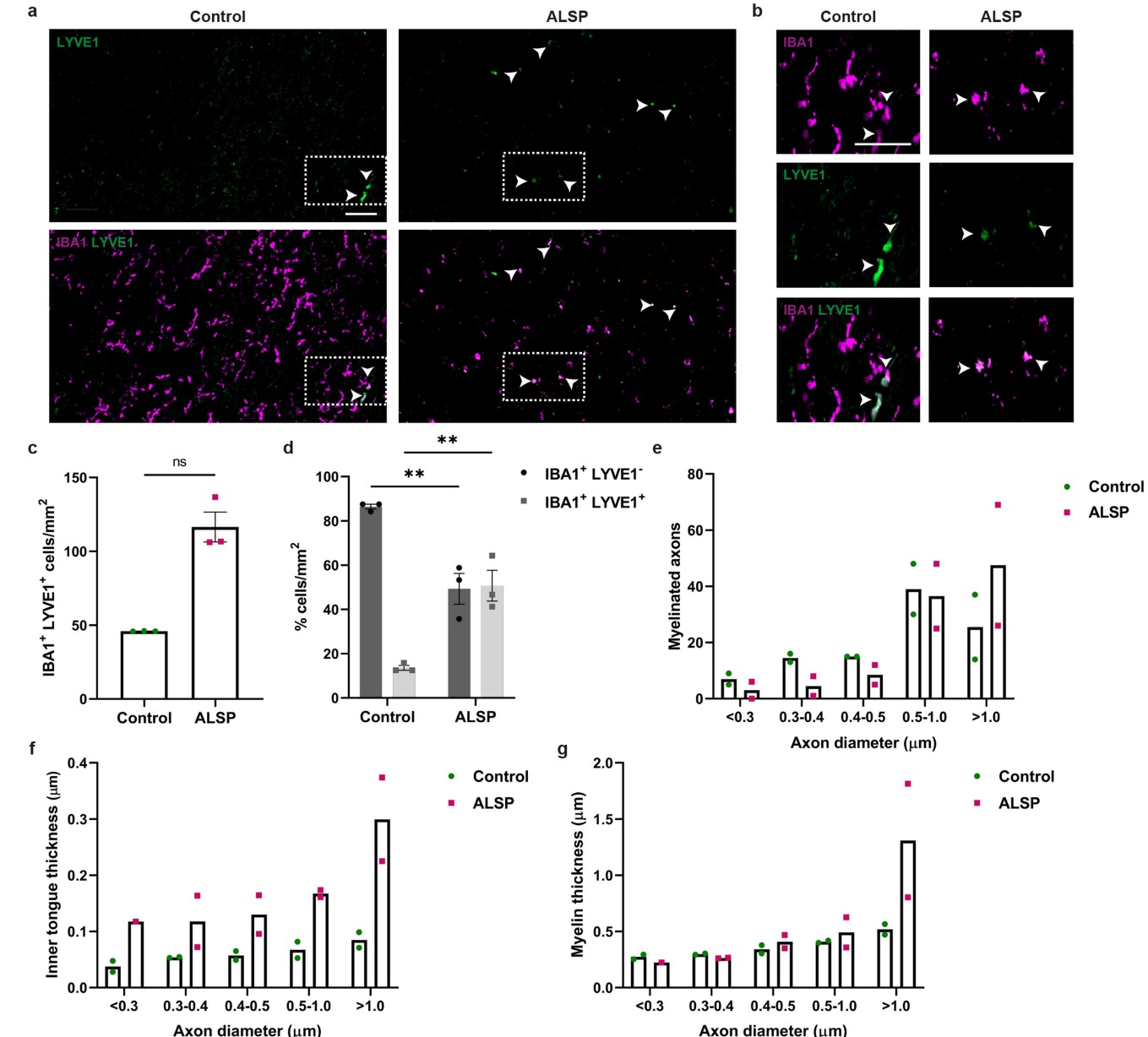

**Extended Data Fig. 8 | Characterization of ALSP tissue. a**) Images of LYVE1+ (green; top) and IBA1+ (magenta) LYVE1+ macrophages (bottom) in human ALSP and age-matched unaffected control frontal white matter. Scale bar, 50 μm. **b**) Magnified images from dotted outlines in (A) of IBA1+ LYVE1+ perivascular macrophages in human ALSP and unaffected age-matched control frontal white matter. Scale bar, 50 μm. **c**) Mean IBA1+ LYVE1+ cells/mm² ± s.e.m. in ALSP and unaffected control. n = 3 cases/group. *P* = 0.1000, Mann Whitney test. **d**) Mean proportion of IBA1+ cells which are perivascular macrophages

(LYVE1+; grey) or microglia (LYVE1-; black) ± s.e.m. in ALSP and unaffected control. n = 3 cases/group. **P* = 0.0034, 1-way ANOVA with Tukey's multiple comparisons test. **e**) Mean number of myelinated axons/axon diameter bin in unaffected control and ALSP patients. n = 2 cases/group. **f**) Mean inner tongue thickness (μm)/axon diameter bin in Control and ALSP patients. n = 2 cases/group. **g**) Mean myelin thickness (μm)/axon diameter bin in unaffected control and ALSP patients. n = 2 cases/group.

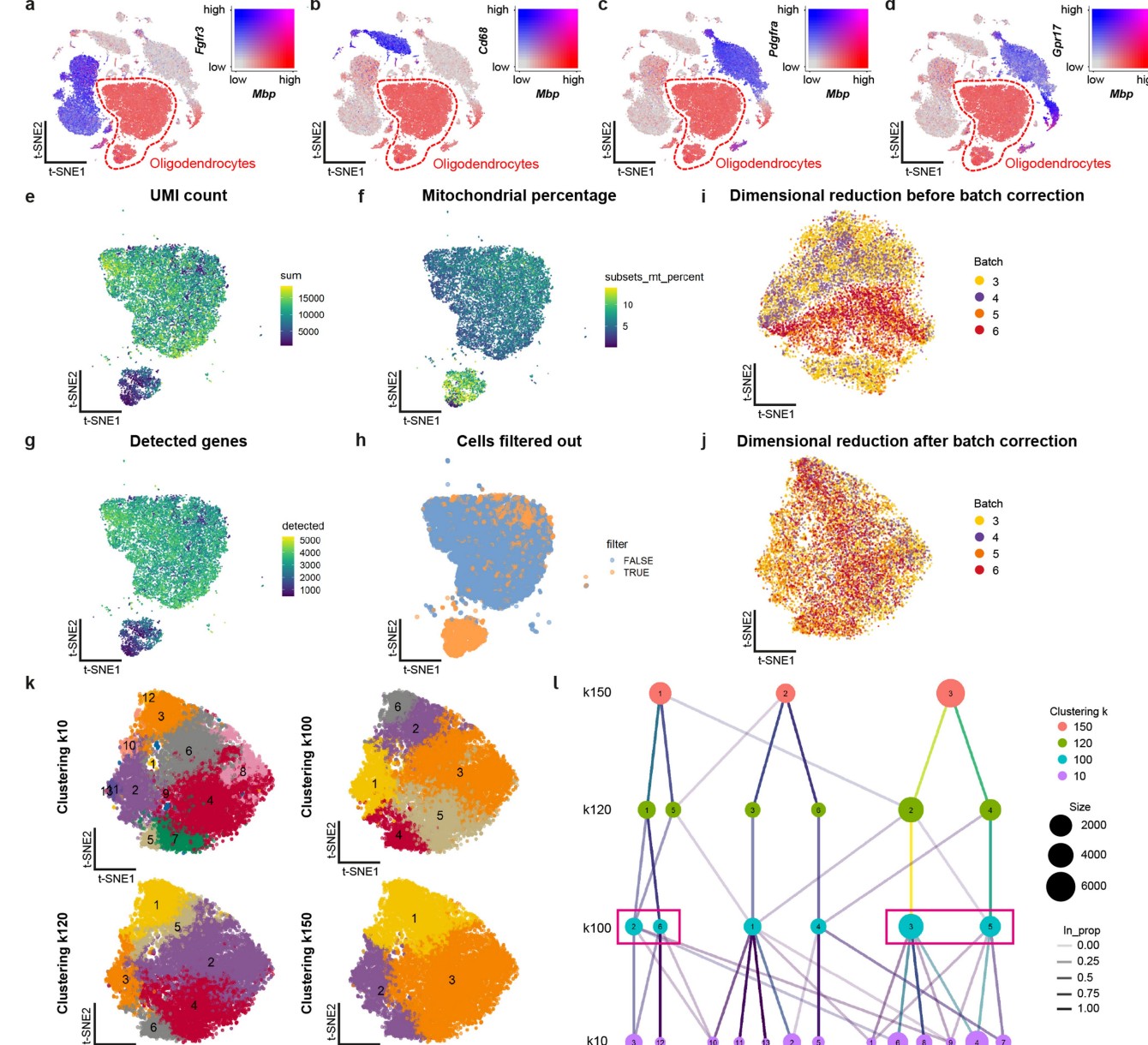

**Extended Data Fig. 9 | Single-cell RNA sequencing analysis of FIRE$^{\Delta/\Delta}$ mice.**
(**a**-**d**) t-SNE (t-distributed stochastic neighbour embedding) plots of pre-filtered
dataset showing oligodendrocytes (*Mbp*) versus other cell types: **a**) *Fgfr3*
(astrocytes), **b**) *Cd68* (microglia/macrophages), **c**) *Pdgfra* (OPCs) and **d**) *Gpr17*
(committed oligodendrocyte precursors), with outline around subsetted
oligodendrocytes. (**e**-**h**) t-SNE plots where each point is a cell coloured according
to: (**e**) Sum of unique molecular identifiers (UMI) counted, (**f**) Percentage of
mitochondrial genes, (**g**) Number of detected genes, (**h**) Filtering: In blue are
the cells that were kept for subsequent analysis, and in orange the ones filtered
out on the basis of UMIs and percentage of mitochondrial genes, see methods
for thresholds. (**i**) t-SNE plot without any batch correction. Each point is a cell

that is coloured according to its batch of origin (yellow = 3, purple = 4,
orange = 5, red = 6). (**j**) t-SNE plot after batch correction with fastMNN. Each
point is a cell that is coloured according to its batch of origin (yellow = 3,
purple = 4, orange = 5, red = 6). (**k**) Clustering with four different resolutions,
obtained by modifying the number of nearest neighbours (*k*). Each resolution
is represented in a separate t-SNE plot. (**l**) Clustree plot created using clustree
v0.4.4. Each level in this plot denotes a different resolution used for clustering
cells. Lines denote contributions of cells from previous clusters, and the size of
the circles represents the number of cells. The final clusters were obtained by
merging clusters 2 and 6 and clusters 3 and 5 (pink boxes) from the clustering
with k = 100 (light blue).

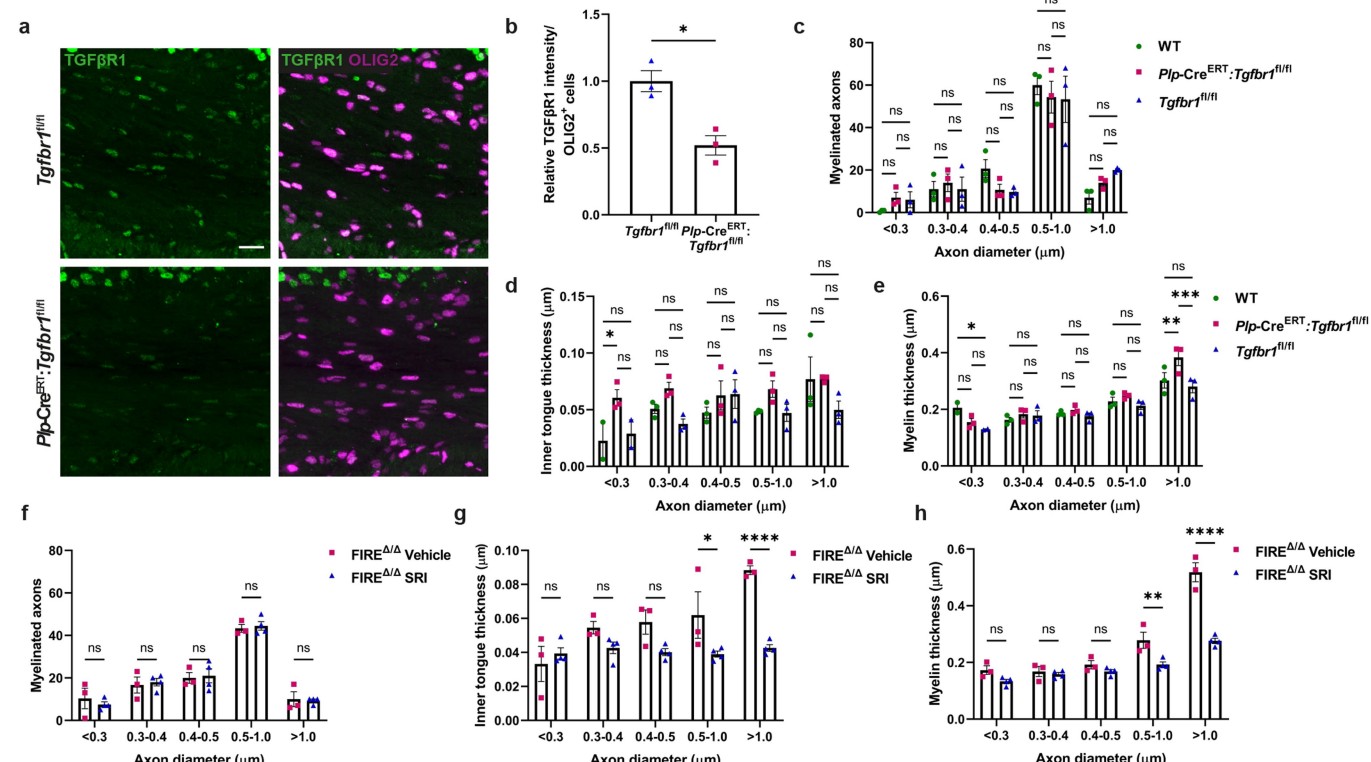

**Extended Data Fig. 10 | Influence of TGFβ receptor manipulation on myelin health. a**) Images of TGFβR1 expression (green) by OLIG2+ cells (magenta) in corpus callosum of *Tgfbr1*[fl/fl] and *Plp*-creERT;*Tgfbr1*[fl/fl] mice, who received 4-hydroxy tamoxifen (OHT; 100 mg/kg/day, P14–18) and were sacrificed at P28. Scale bar, 20 μm. **b**) Mean relative TGFβR1 intensity ± s.e.m. per OLIG2+ cell in *Tgfbr1*[fl/fl] and *Plp*-creER::*Tgfbr1*[fl/fl] mice. n = 3 mice/group. *P = 0.011, 2-tailed unpaired Student's t-test. **c**) Mean number of myelinated axons/axon diameter bin ± s.e.m. in wildtype mice (WT; green), *Plp*-creERT;*Tgfbr1*[fl/fl] mice (magenta) and *Tgfbr1*[fl/fl] mice (blue). n = 3 mice/group. Non-significant, P = 0.3001, 2-way ANOVA with Sidak's multiple comparisons test. **d**) Mean inner tongue thickness (μm)/axon diameter bin ± s.e.m. in wildtype mice (WT; green), *Plp*-creERT;*Tgfbr1*[fl/fl] mice (magenta) and *Tgfbr1*[fl/fl] mice (blue). n-3 mice/group. <0.3 μm WT vs. *Plp*-creERT;*Tgfbr1*[fl/fl]: *P = 0.03950, 2-way ANOVA with Sidak's multiple comparisons test. **e**) Mean myelin tongue thickness (μm)/axon diameter bin ± s.e.m. in wildtype mice (WT; green), *Plp*-creERT;*Tgfbr1*[fl/fl] mice (magenta) and *Tgfbr1*[fl/fl] mice (blue). n = 3 mice/group. <0.3 μm WT vs. *Tgfbr1*[fl/fl]:

*P = 0.0291; >1 μm WT vs. *Plp*-creERT;*Tgfbr1*[fl/fl] **P = 0.0037 and *Plp*-creERT; *Tgfbr1*[fl/fl] vs. *Tgfbr1*[fl/fl] ***P = 0.0003, 2-way ANOVA with Sidak's multiple comparisons test. **f**) Mean number of myelinated axons ± s.e.m./axon diameter bin in 3-month-old FIRE[Δ/Δ] mice following treatment with vehicle control or SRI-011381 hydrochloride from 2 to 3 months of age. n = 3 Vehicle-treated and 4 SRI-treated mice. Non-significant, P = 0.9202, 2-way ANOVA with Sidak's multiple comparisons test. **g**) Mean inner tongue thickness (μm) ± s.e.m. per axon diameter bin in 3-month-old FIRE[Δ/Δ] mice following treatment with vehicle control or SRI-011381 hydrochloride from 2 to 3 months of age. n = 3 Vehicle-treated and 4 SRI-treated mice. 0.5-1.0 μm: *P = 0.0334, >1μm: ****P < 0.0001, 2-way ANOVA with Sidak's multiple comparisons test. **h**) Mean myelin thickness (μm) ± s.e.m./axon diameter bin in 3-month-old FIRE[Δ/Δ] mice following treatment with vehicle control or SRI-011381 hydrochloride from 2 to 3 months of age. n = 3 Vehicle-treated and 4 SRI-treated mice. 0.5-1.0 μm: **P = 0.0027, >1 μm: ****P < 0.0001, 2-way ANOVA with Sidak's multiple comparisons test.

**Extended Data Table 1 | Clinical information from human tissue analysis**

| Experiment | Case | CSF1R Mutation | Age | Sex | Cause of Death |
|---|---|---|---|---|---|
| IHC | ALSP 1 | c.2381T>C; p.(I794T) | 58 years old | M | ALSP |
| IHC | ALSP 2 | c.1786G>A; p.(V596M) | 36 years old | F | Pulmonary embolism |
| IHC | ALSP 3 | c.2541G>C; p.(E847D) | 53 years old | F | ALSP |
| EM | ALSP 4 | c.2330G>A; p.(R777Q) | 40 years old | M | ALSP |
| EM | ALSP 5 | c.2330G>A; p.(R777Q) | 22 years old | F | Bacterial pneumonia |
| IHC | Control 1 | N/A | 58 years old | M | Rectosigmoid adenocarcinoma |
| IHC | Control 2 | N/A | 39 years old | M | Metastatic gastric carcinoma |
| IHC | Control 3 | N/A | 56 years old | F | Non-neurological |
| EM | Control 4 | N/A | 34 years old | M | Ischaemic heart disease & coronary artery atherosclerosis |
| EM | Control 5 | N/A | 40 years old | M | Myocardial infarction & coronary artery atherosclerosis |

EM, electron microscopy; F, female; IHC, immunohistochemistry; M, male; N/A, not applicable.

# Reporting Summary

## Statistics

For all statistical analyses, confirm that the following items are present in the figure legend, table legend, main text, or Methods section.

| n/a | Confirmed | |
|---|---|---|
| ☐ | ☒ | The exact sample size (*n*) for each experimental group/condition, given as a discrete number and unit of measurement |
| ☐ | ☒ | A statement on whether measurements were taken from distinct samples or whether the same sample was measured repeatedly |
| ☐ | ☒ | The statistical test(s) used AND whether they are one- or two-sided *Only common tests should be described solely by name; describe more complex techniques in the Methods section.* |
| ☒ | ☐ | A description of all covariates tested |
| ☐ | ☒ | A description of any assumptions or corrections, such as tests of normality and adjustment for multiple comparisons |
| ☐ | ☒ | A full description of the statistical parameters including central tendency (e.g. means) or other basic estimates (e.g. regression coefficient) AND variation (e.g. standard deviation) or associated estimates of uncertainty (e.g. confidence intervals) |
| ☐ | ☒ | For null hypothesis testing, the test statistic (e.g. *F*, *t*, *r*) with confidence intervals, effect sizes, degrees of freedom and *P* value noted *Give P values as exact values whenever suitable.* |
| ☒ | ☐ | For Bayesian analysis, information on the choice of priors and Markov chain Monte Carlo settings |
| ☒ | ☐ | For hierarchical and complex designs, identification of the appropriate level for tests and full reporting of outcomes |
| ☒ | ☐ | Estimates of effect sizes (e.g. Cohen's *d*, Pearson's *r*), indicating how they were calculated |

*Our web collection on statistics for biologists contains articles on many of the points above.*

## Software and code

Policy information about availability of computer code

| | |
|---|---|
| Data collection | OpenEpi.com was used for power calculations for sample size. Leica SPE or Zeiss LSM 510 microscopes/software were used for confocal imaging. Human tissue was acquired using the Zeiss Zen2 (blue edition) software on a Zeiss AxioScan Z.1 SlideScanner. Electron micrographs were acquired on a Zeiss 9P02 electron microscope or Zeiss Gemini 300 scanning electron microscope. Single cells were processed on the 10X Chromium single cell platform and RNA sequences were read on a NovaSeq6000. Lipidomics were performed using the Nexera X2 UHPLC system and hybrid triple quadruple/linear ion trap mass spectrometer (6500+ QTRAP system; AB SCIEX). Flow cytometry was performed using a BD LSR Fortessa. Behavioural data was acquired using a Barnes Maze (San Diego instruments). |
| Data analysis | Cell counts were performed using Fiji/Image J software (Fiji.sc) and colocalization performed using Imaris software v9.7. Data handling was performed using Microsoft Excel 2016 and GraphPad Prism v.8 and v.9 were used for statistical analyses. Human electron micrograph images were stitched together using Fiji and QuPath 0.3.0 softwares. RNA sequencing analysis involved use of 10X Genomics Cell Ranger v5.0.0 pipeline to align the reads to the reference genome, using the mm10 genome supplied by 10X Genomics (refdata-gex-mm10-2020-A), R v4.1.1 to run the code, , SingleCellExperiment v1.14.1 to handle single cell experiment objects in R, scater v1.20.1 for quality control and producing plots, scran v1.20.1 for normalisation, batchelor v1.8.0 for the batch correction with fastMNN, clustree v0.4.4 to visualise the clustering, Seurat v4.1.0 for differential gene expression, ShinyCell v2.1.0 to produce an interactive app, org.Mm.eg.db v3.13.0 to annotate genes, ggplot2 v3.3.5 to perform custom plots, here v1.0.1 to ensure reproducible paths, and Matrix v1.3.4 for handling sparse matrices. Flow cytometry data was analysed on FCS express v.7. Behavioural data was assessed using Anymaze for video tracking (Stoelting Europe v4.9.9). |

For manuscripts utilizing custom algorithms or software that are central to the research but not yet described in published literature, software must be made available to editors and reviewers. We strongly encourage code deposition in a community repository (e.g. GitHub). See the Nature Portfolio guidelines for submitting code & software for further information.

## Data

Policy information about availability of data

All manuscripts must include a data availability statement. This statement should provide the following information, where applicable:

- Accession codes, unique identifiers, or web links for publicly available datasets
- A description of any restrictions on data availability
- For clinical datasets or third party data, please ensure that the statement adheres to our policy

The data that support the findings from this study are available from the corresponding author upon request as source data associated with this manuscript. Raw single cell RNA sequences datasets have been deposited in the Gene Expression Omnibus, accession code GSE215440. Sequencing Analyzed oligodendrocyte sequencing data is perusable on the following shiny app: https://annawilliams.shinyapps.io/shinyApp_oligos_VM/. Full details to replicate the analysis pipelines can be found in code scripts available on GitHub (https://github.com/Anna-Williams/Veronique-Firemice). Alignment to the reference genome, feature counting and cell calling was performed following the 10X Genomics CellRanger (v5.0.0) pipeline, using the default mm10 genome supplied by 10X Genomics (https://cf.10xgenomics.com/supp/cell-exp/refdata-gex-mm10-2020-A.tar.gz).

## Human research participants

Policy information about studies involving human research participants and Sex and Gender in Research.

| | |
|---|---|
| Reporting on sex and gender | This study used post-mortem tissue of human brain from non-neurological controls and people with ALSP. For controls, this consisted 4 males and 1 female. The ALSP cases consisted of 2 males and 3 females. All information pertaining to these tissues is listed in Extended Data Table 1. |
| Population characteristics | 5 non-neurological controls died of rectosigmoid adenocarcinoma, metastatic gastric carcinoma, ischaemic heart disease/coronary artery atherosclerosis, and myocardial infarction/coronary artery atherosclerosis, and ranged from 34 to 58 years old. ALSP cases consisted of 3 who died of ALSP, 1 who died of pulmonary embolism, and 1 of bacterial pneumonia, and ranged from 22 to 58 years old. All information pertaining to these tissues is listed in Extended Data Table 1. |
| Recruitment | The tissue was obtained with full ethical approval from the Queen Square Brain Bank for Neurological Disorders, UCL Queen Square Institute of Neurology, the Department of Neuropathology at Charité-Universitätsmedizin Berlin and the Medical Research Council Edinburgh Brain and Tissue Bank (EBTB), and their use was in accord with the terms of the informed consents. |
| Ethics oversight | Ethical approval was granted to the Queen Square Brain Bank for the use of tissue by the National Health Service Health Research Authority through the London Central Research Ethics Committee. |

Note that full information on the approval of the study protocol must also be provided in the manuscript.

# Field-specific reporting

Please select the one below that is the best fit for your research. If you are not sure, read the appropriate sections before making your selection.

☒ Life sciences  ☐ Behavioural & social sciences  ☐ Ecological, evolutionary & environmental sciences

For a reference copy of the document with all sections, see nature.com/documents/nr-reporting-summary-flat.pdf

# Life sciences study design

All studies must disclose on these points even when the disclosure is negative.

| | |
|---|---|
| Sample size | For in vivo experiments, n=3-7 mice per genotype per time point. For human tissue, n=2-3 cases per condition. For mouse studies, sample size was determined using power calculations using OpenEpi.com, to reach a power of >80% at 0.05 significance. For human studies, no sample size calculation was performed as ALSP is an extremely rare disorder and we obtained as many cases as were available for our purposes. |
| Data exclusions | In the Barnes Maze data, 1 wildtype and 1 FIRE knockout mouse were excluded due to refusal to enter the escape hole during the training phase. This is indicated in the methods section. |
| Replication | All attempts at replication were successful |
| Randomization | Mice were randomly assigned to time points of assessment. For human studies, randomization was not involved nor applicable, as all tissue samples of a group were submitted to the same analyses (e.g. all resin blocks were submitted to EM processing, all frozen tissue to immunostaining). |
| Blinding | All manual counts were performed in a blinded manner. All behaviour experiments were performed in a blinded manner from data collection through to completion of data analysis. All analyses on human tissue were performed in a blinded manner. |

# Reporting for specific materials, systems and methods

We require information from authors about some types of materials, experimental systems and methods used in many studies. Here, indicate whether each material, system or method listed is relevant to your study. If you are not sure if a list item applies to your research, read the appropriate section before selecting a response.

## Materials & experimental systems

| n/a | Involved in the study |
|-----|-----------------------|
| ☐ | ☒ Antibodies |
| ☒ | ☐ Eukaryotic cell lines |
| ☒ | ☐ Palaeontology and archaeology |
| ☐ | ☒ Animals and other organisms |
| ☒ | ☐ Clinical data |
| ☒ | ☐ Dual use research of concern |

## Methods

| n/a | Involved in the study |
|-----|-----------------------|
| ☒ | ☐ ChIP-seq |
| ☐ | ☒ Flow cytometry |
| ☒ | ☐ MRI-based neuroimaging |

## Antibodies

| | |
|---|---|
| Antibodies used | The following antibodies were used: α-MBP (AbD Serotec, 1:250; MCA409S, clone 12), α-MAG (Millipore, 1:100; MAB1567, clone 513), α-MOG (Millipore, 1:100; MAB5680, clone 8-18C5), α-CNPase (Sigma-Aldrich, 1:100; AMAB91072, clone CL2887), α-TMEM119 (Abcam, 1:100; ab209064, clone 28-3), α-IBA1 (Abcam, 1:500; ab5076),  α-CD206 (Abcam, 1:100; ab64693), α-CD31 (R&D Systems, 1:100; AF3628), α-LYVE1 (Abcam, 1:100; ab14917), α-OLIG2 (Millipore, 1:100; AB9610), α-OLIG2 (Millipore, 1:100; MABN50, clone 211F1.1), α-APC ('CC1'; Abcam, 1:100; ab16794, clone CC1), α-SOX9 (Millipore, 1:500; AB5535), α-GFAP (Cambridge Bioscience, 1:500; 829401), α-Neurofilament-H (Biolegend, 1:100,000; PCK-592P), and α-PLP (Abcam, 1:100; ab28486), α-SERPINA3N (R&D Systems, 1:100; AF4709), α-TGFβR1 (Abcam, 1:100; ab31013). Flow cytometry: anti CD11b (PE; 1:200; 101207; BioLegend, clone M1/70), CD45 (PECy7; 1:200; 103114; BioLegend, clone 30-F11), and CD115 (APC; 1:200; 135510; BioLegend, clone AFS98) |
| Validation | Validation for the antibodies used for our indications are stated on the supplier websites as follows. 1) Positive signal in positive controls for α-MAG (human spleen and rat hippocampus), α-IBA (rat brain), α-CD206 (mouse lung), α-APC/CC1 (cerebellum), α-SOX9 (HepG2 cell lysate), α-GFAP (rat brain), α-TGFBR1 (human placenta). 2) α-TMEM119 was validated by absence of signal in knockout tissue. 3) References provided on supplier website for published literature using antibodies, for α-SERPINA3N, α-MBP, α-CD11b, α-CD45, α-CSF1R. 4) Validation of target by supplier via experimentation, for α-MOG (immunoblot), α-NFH (Western Blotting). For two antibodies, α-CNPase and α-PLP, the suppliers did not provide validation information. However, we have seen the same pattern of immunohistochemical signal for these myelin protein stains as for MOG, MAG, and MBP, which are all well validated. All secondary antibodies used were tested in Miron et al., 2013, Nature Neuroscience, Dillenburg et al., 2018, Acta Neuropathologica and Lloyd et al., 2019, Nature Neuroscience. |

## Animals and other research organisms

Policy information about studies involving animals; ARRIVE guidelines recommended for reporting animal research, and Sex and Gender in Research

| | |
|---|---|
| Laboratory animals | This study used Csf1r-FIREd/d mice, Csf1r-FIRE+/+ mice, both of mixed background (B6CBAF1/J and C57Bl6). Mice were examined at 1, 3-4, and 6 months of age. The Plp-CreERT2 mice and Tgfbr1 fl/fl mice were both on C57Bl6J background. The offspring were assessed at 28 days of age. All mice were housed at room temperature (18-23C)  at 40-60% humidity.<br>ARRIVE2 guidelines have been followed for reporting and included in the manuscript, such as ethical permissions, animal strains used, methods of termination, all commercial providers for reagents, sex of animals, exact n numbers used, statistical information and p values. |
| Wild animals | No wild animals were used in this study. |
| Reporting on sex | Both sexes were used for analysis for all animal experiments. |
| Field-collected samples | No field collected samples were used in this study. |
| Ethics oversight | Experiments were performed under a UK Home Office project licence, approved by the UK Home Office and issued under the Animals (Scientific Procedures) Act. |

Note that full information on the approval of the study protocol must also be provided in the manuscript.

# Flow Cytometry

## Plots

Confirm that:

☒ The axis labels state the marker and fluorochrome used (e.g. CD4-FITC).

☒ The axis scales are clearly visible. Include numbers along axes only for bottom left plot of group (a 'group' is an analysis of identical markers).

☒ All plots are contour plots with outliers or pseudocolor plots.

☒ A numerical value for number of cells or percentage (with statistics) is provided.

## Methodology

| | |
|---|---|
| Sample preparation | After transcardially perfusing 10–11-week-old female mice with ice-cold PBS, brains were dissected and minced with a 22A scalpel in HBSS (without Ca2+ and Mg2+; 14175-053; Gibco) with 25 mM HEPES (10041703; Fisher Scientific). Brains were then homogenised using a Dounce homogeniser (D9938; Kimble) in HBSS (w/o Ca2+ and Mg2+) with 25 mM HEPES. Brain homogenates were separated using a 35% Percoll gradient, with centrifugation at 800 g for 20 mins at 4°C (with no brake). Cell pellets were collected and washed in PBS (w/o Ca2+ and Mg2+; 14190-094; Gibco) with 0.1% low endotoxin BSA (A8806; Sigma Aldrich). Fc receptors were blocked (1:100; 101302; BioLegend) for 15 min at 4°C on a shaker. Cells were then stained with primary antibodies. |
| Instrument | Data were acquired using a BD LSRFortessa™ Flow Cytometer. |
| Software | FCS express 7 was used for post-acquisition data analysis. |
| Cell population abundance | All post sort cell abundance are indicated in Extended Data Figure 1d,e. |
| Gating strategy | Gating strategy is provided in Extended Data Figure 1d |

☒ Tick this box to confirm that a figure exemplifying the gating strategy is provided in the Supplementary Information.

