## [Peer Review File · Nature]

Manuscript Title: Microglia regulate central nervous system myelin growth and integrity

Reviewer Comments & Author Rebuttals

Reviewer Reports on the Initial Version:

Referees' comments:

Referee #1 (Remarks to the Author):

The study from the Miron lab demonstrates that microglia, a subset of CNS myeloid cells, are required for maintenance of myelin integrity. This leverages a new genetic mouse model with a deletion in an enhancer region of the CSF1R gene to which some but not all myeloid cells are susceptible. A learning task induced oligodendrogenesis but failed to increase number of myelinated axons in FIRE mice, indicating a role for microglia in successful activity-regulated/learning-induced myelination. Strikingly, in humans with ALSP leukoencephalopathy syndrome, selective loss of white matter microglia is associated with histological evidence of loss of myelin integrity, myelin outfoldings and enlarged inner tongues, together with increasing demyelination with age, reminiscent of the FIRE mouse findings.

This is a beautiful and important paper. It adds a great deal to our understanding of myeloid cell functional diversity in the CNS and underscores a newly appreciated critical role for microglia in myelin integrity homeostasis and plasticity. It also elucidates the pathophysiology of ALSP leukoencephalopathy.

I have the following suggestions for improvement of an excellent paper.

1. The key point to address is the need for a more nuanced understanding and discussion of microglial subpopulations and heterogeneous microglial cell states. Numerous subpopulations of microglia have been revealed by single cell sequencing work (Hammond et al 2018 Immunity) and the various disease-associated states that microglia can assume, with vastly different effects on myelin in different states. The senior author's own elegant postdoctoral work demonstrating different effects of microglia in trophic "M2" or proinflammatory "M1"-like states on demyelination should be discussed. Also worth discussing is recent work demonstrating the specific activation of white matter microglia after systemic exposure to the chemotherapy drug methotrexate and consequent impairment in oligodendrogenesis and myelination that are rescued with CSF1R inhibitor blockade-mediated microglial depletion (Gibson et al 2019 Cell, Geraghty 2019 Neuron).

2. Regarding the later point, CSF1R inhibitor therapy is in numerous clinical trials for a range of indications. Is there a safe window for microglial depletion to allow a "reset" of microglia in a disease-causing state? Or is there a threshold level of microglia required for maintenance of myelin integrity? This should be discussed.

3. It would add a great deal to the paper (and to future use of this mouse model) to understand at a

more granular level what myeloid cell populations are present, they state they are in, and how this changes with age. Single cell RNA-seq would be ideal, but other methods could also shed more light on the myeloid subpopulations that remain and how this evolves. Such characterization of the CNS myeloid subpopulations would be a valuable addition.

4. In Figure 4, in addition to the scatterplots, please also express the data in 4E and 4F (inner tongue thickness and myelin thickness) as mean per animal (n = 3-4). Are stats run on a per mouse basis or a per axon basis?

Minor:

5. "Short-term" memory is on the order of seconds to minutes. Barnes maze is more accurately described as testing memory "encoding", and the wording should reflect that

6. Line 140 - remove "short term", more accurate to say "learning and memory encoding". Here, Geraghty et al 2019 Neuron should also be cited (novel object recognition test performance depends on activity-regulated oligodendrogenesis)

7. Please state that ALSP leukoencephalopathy results from CSF1R gene mutations (somehow this point was omitted)

8. Although beyond the scope of this paper, do the authors have any thoughts as to the mechanisms that may underlie microglial-mediated maintenance of myelin integrity? Just a point for discussion.

Referee #2 (Remarks to the Author):

McNamara et al. Nature 2021

McNamara et al. describe in their manuscript a potential role for microglia in the maintenance and long-term integrity of myelin. Using the newly described FIRE KO mouse, which are selectively depleted of microglia, they demonstrate myelin degeneration and overall changes in myelin integrity (inner tongue and myelin thickness) at 6M of age, which coincides with deficits in spatial reversal learning. By further assessing the brains of ALSP patients, they could demonstrate that reductions in white matter microglia was also associated with loss of myelin integrity in humans.

Although overall the quality of the work was solid, the study remains highly descriptive and correlative. It lacks, in my view, further experimental evidence and mechanistic insights into how loss of microglia leads to changes in myelin integrity and cognitive deficits, or whether these observations could be explained by other factors.

Major comments:

1) At 1M of age the number of oligodendrocytes and myelinated axons are not affected, despite the lack of microglia. However, at 3-4 and 6M of age, the quantification of the number of oligodendrocytes is missing. Could demyelination be due to a loss of oligodendrocytes per se?

2) Mechanistic insight into why lack of microglia leads to decreased myelination is lacking. Do they

affect oligodendrocyte survival? Which factors are needed from the microglia to reconstitute the oligodendrocyte function?

3) Orthogonal experiments that demonstrate the validity of the observation/conclusions are lacking. Is chemical depletion of microglia (e.g. with CSF1R inhibitors), having a similar long-term impact on myelin compaction and myelin degeneration? Alternatively, could reconstitution of microglia rescue these myelin phenotypes?

4) For some of the regression analyses on myelin thickness and inner tongue thickness (e.g. 4E&F, 2M), the statistically significant difference seems marginal (i.e. the difference in slope). What's the biological impact of these differences seen in regression? Although the validity of this way of analyzing the data is not questioned, the binned approach as demonstrated in Fig.2F gives a better understanding of where the differences between the genotypes can be observed.

5) P.6 lines 149-154: can you be more specific of the impact of this observation? I.e. 'in a subset of mice perfused for ultrastructural analysis'. How large was this subset in comparison to all mice? What is the percentage increase in number of myelinated axons in the FIRE WT mice 6 weeks after learning (fig.3L)? Can you show this in representative images, highlighting unmyelinated axons and myelinated axons, because I cannot observe this difference in Fig.3K.

Are the trained and untrained mice used for this analysis of the exact same age? The age-range for mice tested for cognition is quite broad, i.e. 2-4M. Can you be more specific on the median age between your FIRE WT and FIRE KO groups (both for untrained and trained) at the day of sacrifice?

6) P.6 lines 152-154: I find this conclusion too strong. The increase in the number of myelinated axons is only observed in a 'subset of mice'. Also your data showing a 'lack of cognitive flexibility' is as strong as your data showing enhanced short term spatial memory (extended Fig.5B). Although this finding is mentioned in the text, no further attention is given to this, while the 'lack of cognitive flexibility' is stressed. How sure are you that this reduction in the number of myelinated axons underlies these (perhaps serendipitous) findings in behavior? Do mice with the lowest number of myelinating axons also perform the worst? Is there a way to confirm these findings, i.e. by reducing or increasing the number of myelinated axons (e.g. chemically) and assess the impact on cognition? To what extent can these cognitive changes truly be linked to impaired myelination? The lack of microglia may also impact neuronal function and cognition independently of the effect on myelin/oligodendrocytes.

7) Fig.5D, E: Can you explain why ALSP patients have much thicker axons than healthy controls? Can this skew your observation on myelin thickness?

8) Regarding the axonal spheroids: How often are these present (e.g. in every mouse Are these axonal spheroids never observed in WT mice? How often are they observed in the FIRE KO mice? I.e. which percentage of axons have these spheroids?

Minor comments:

1) I am missing the figure legends on the extended figures, making it hard to interpret some.

2) It would be easier for comparison purpose if all plots on myelin thickness were on the same scale (Y-axis). Although linear regression is the correct statistical method for analyzing the data, it is difficult to get a feeling for the percentage of difference in myelin thickness across age and genotype. Is it possible to visualize this in this binned analysis similar to what is shown for the inner tongue thickness? Perhaps all ages and genotypes can be combined in 1 graph, just for visualization purposes (in appendix) and to make the reader understand the severity of the increased myelin

thickness.

I would however, suggest to perform a combined analysis of your regression slopes across the different ages and genotypes, to reduce the chance of a type I error.

3) Fig.1H and others: you perform a 2x2 ANOVA on Olig2+CC1- vs Olig2+CC1+ cells. However, the total of Olig2+CC1 plus Olig2+CC1+ is set at 100%. That means that if the proportion of CC1- cells drops, your proportion of CC1+ cells automatically increases, making these observations (CC1+ vs CC1-) dependent on each other. In 2x2 ANOVA you are assessing the impact of 2 independent factors. Thus, here it would suffice to perform a 1x ANOVA on e.g. only the CC1+ cells and compare this between the WT and KO mice. Please also revise this for all of the other graphs in this manuscript where you use similar sets of data (e.g. 2C, 2K etc).

Referee #3 (Remarks to the Author):

In this study, McNamara et al investigated how microglia regulate CNS myelin formation and maintenance. Bruce Appel's lab recently reported in zebrafish that microglia phagocytose myelin sheaths and microglia depletion leads to excessive and ectopic myelin sheath formation (Hughes and Appel. *Nat Neurosci* 2020). The authors took one step further here by documenting the role of microglia in myelin maintenance in mice and humans. They used the recently developed transgenic mouse model in which deletion of the FIRE super-enhancer of the *Csf1r* gene (FIRE Δ/Δ) leads to an absence of microglia from development (when they normally emerge) through adulthood, while other CNS macrophages, such as perivascular macrophage are present. Major advantage of this mouse model is that there is no acute massive cell death associated with *CSF1r* inhibitor nor early postnatal lethality. In addition to recapitulating Hughes and Appel's finding, namely the absence of microglia has no effect on oligodendrocyte (OL) lineage proliferation and maturation but leads to increased myelin thickness, McNamara et al found increased (44%) outfolds and unraveling myelin in the corpus callosum of FIRE Δ/Δ mice compared to controls (14%). Furthermore, they observed myelin degeneration at 6 months of age. Significantly, they extended their observation to rare human leukoencephalopathy ALSP (Adult-onset Leukoencephalopathy with axonal Spheroids and Pigmented glia) by showing decreased number of microglia and myelin outfoldings and unravelling, thicker myelin, and enlarged inner tongues in young adult brains and progressive demyelination in older brains. Given the importance of microglia in brain homeostasis and neurodegeneration, the study results presented are of immediate interest to many people in broader neuroscience field. However, there are several major concerns hinder the impact of this manuscript. Specifically,

1. The authors contributed all myelin-related phenotypic observations to disruption of myelin integrity. This is problematic. The phenotypes are two folds, excessive myelin sheath growth that leads to hypermyelination (P25-30 and 3-4 months) and loss of myelin integrity (6 months of age). First, in mice, myelination peaks around P30, but is not complete until ~P60 (Bercery K, Macklin WB Dynamics and mechanisms of CNS myelination. *Dev Cell* 32:447–458, 2015). There were striking phenotypes—outfolds, unraveling, impaired compaction of myelin, and increased myelin thickness (fig. 2A-H)—at P25-30, a timepoint when developmental myelin growth is occurring. Secondly, the growth of the myelin sheath in vivo was associated with a larger inner tongue when compared to later stages of development (Snaidero et al. *Cell* 2014). The documented inner tongue phenotype

likely indicates excessive myelin sheath growth, which is supported by the observation of increased myelin thickness (Fig. 2N and O; extended data Fig 3). Thirdly, absence of microglia prevents the increase in myelination that normally occurs with consolidation of new spatial information (Fig. 3K-L).

2. For the reason discussed above, the statement that “microglia are dispensable for developmental myelin formation” is not fully supported.

3. The study is largely descriptive and lacks mechanistic exploration. Literature suggested that microglial sheath pruning corrects myelin targeting and refines myelination by oligodendrocytes, which accounts why depleting microglia leads to increased myelin sheath growth (Hughes and Appel. *Nat Neurosci* 2020). It would be desirable to present data to either concur such mechanism or support a new one.

4. To conclusively evaluate myelin maintenance, it would be desirable to delete FIRE at a time when the active growth phase of myelin has passed, which would be P60 in mice (Bercury K, Macklin WB Dynamics and mechanisms of CNS myelination. *Dev Cell* 32:447–458, 2015).

5. Activating mTOR/AKT pathway leads to CNS hypermyelination (Snaidero et al *Cell* 2014; Flores et al *J Neurosci* 2008; Narayanan...Macklin. *J Neurosci* 2009). Does depleting microglia results in increased p-mTOR, p-P70S6K, p-S6RP? Chronic inhibiting AKT signaling by rapamycin treatment correct the hypermyelination phenotype in mice that express constitutive active Akt (Narayanan et al *J Neurosci* 2009). Could rapamycin treatment ameliorates FIRE Δ/Δ hypermyelination phenotype?

6. Hypermyelination could lead to demyelination in PNS (Adlkofer et al *Nat Genetics* 1995) and CNS (Hu et al *BioRxiv* 2020, <https://doi.org/10.1101/2020.11.10.377226>). Was the myelin degeneration here due to hypermyelination or other mechanism?

7. The authors showed severe demyelination phenotype at 6 months of age. It would be informative to know when the demyelination starts to occur. In addition to TEM data. MBP western blot of the corpus callosum would be informative.

8. The overall statistical tests are appropriate. However, it would be desirable to provide actual P value instead of “non-significant” for Fig 1K and to present quantitative data and statistical analysis on the fluorescent intensity for Fig. 1I, Extended data Fig 2 and Extended data Fig 4A.

We appreciate positive comments from all 3 reviewers that our manuscript ‘*adds a great deal to our understanding of myeloid cell functional diversity in the CNS*’, is of ‘*immediate interest to many people in the broader neuroscience field*’, and the ‘*quality of the work was solid*’. We have now addressed the reviewers’ comments with a substantial amount of new data, which has significantly improved the mechanistic insight of our study, by performing single cell RNA sequencing and validation, lipidomics, 4 new *in vivo* experiments, and further characterization of our mouse model, resulting in 7 new figures and new data added to another 6 figures. We build on our original submission which identified that microglia are required for regulation of myelin growth and integrity, by now showing that the loss of myelin health in the absence of microglia is associated with the appearance of an oligodendrocyte state with altered lipid metabolism and is regulated by disruption of TGFβ1 signalling. These experiments are summarized below, followed by a point-by-point response to reviewer comments. We have highlighted the changes in the manuscript and extended data file in yellow to facilitate review.

1) Summary of new data:

- **Identification of new cellular and molecular mechanisms by which microglia regulate myelin growth and integrity:**
 - **Microglia regulate the appearance of an oligodendrocyte state:** Single cell RNA sequencing of FIRE^{ΔΔ} mice vs wildtype control revealed the appearance of a new oligodendrocyte state in the absence of microglia, validated by immunostaining to be almost exclusively present in FIRE^{ΔΔ} mouse white matter. Interestingly, this state shows similarities with one recently shown to appear in other mouse models of CNS injury.
 - **Microglia regulate lipid metabolism pathways in the oligodendrocyte state:** Dysregulated cholesterol/lipid metabolism were identified as pathways differentially regulated in the FIRE^{ΔΔ} mouse oligodendrocyte state, validated by lipidomics analysis. Results were consistent with hypermyelination in FIRE^{ΔΔ} mice.
 - **The TGFβ1-TGFβR1 axis is dysregulated in the absence of microglia:** TGFβ1 was bioinformatically predicted to be an upstream regulator of the oligodendrocyte state in FIRE^{ΔΔ} mice. TGFβ1 was significantly reduced in FIRE^{ΔΔ} mouse white matter, and TGFβR1 expression was decreased in FIRE^{ΔΔ} mouse oligodendrocytes.
 - **TGFβ receptor signalling in oligodendrocytes is required for myelin health:** Conditional knockout of TGFβR1 in mature oligodendrocytes revealed that loss of this receptor was sufficient to cause hypermyelination, mirroring results in FIRE^{ΔΔ} mice and revealing a new molecular mechanism by which myelin growth is regulated.
 - **Myelin health is rescued in the absence of microglia by activating TGFβR1 downstream signalling:** A small molecule activator of TGFβ receptor signalling rescued myelin pathology in FIRE^{ΔΔ} mice, such that myelin profiles aligned with those in wildtype control.
- **Assessment of impact of microglia depletion on myelin in adulthood:**
 - Depletion of microglia in adulthood, once myelination is already complete, mimicked the myelin pathology observed in FIRE^{ΔΔ} mice, including axon diameter- and age-dependent effects. This indicates an impact of microglia on myelin maintenance once it is already formed rather than an influence on developmental myelination.
- **Further characterization of demyelination occurring in FIRE^{ΔΔ} mice:**
 - Demyelination is initiated at 4.5 months of age.
 - Demyelination is not associated with oligodendrocyte loss at all ages assessed.
 - Demyelination occurs in small focal areas in a patchy pattern, which is only observable by ultrastructural analysis.
 - Demyelination is more prevalent in larger diameter axons in FIRE^{ΔΔ} mice, the same axon group that shows the most robust hypermyelination, suggesting that hypermyelination may precede demyelination.
- **Further characterization of perivascular macrophages:**
 - No change in perivascular macrophage number at all ages assessed.

- Perivascular macrophages downregulate CSF1R in FIRE^{ΔΔ} mice.
- **Further characterization of axonal changes in FIRE^{ΔΔ} mice:**
 - No overt axonal pathology: Phosphorylated neurofilament is not significantly affected and axonal spheroids are observed in <0.1% of axons at 3-6 months of age.

2) Response to reviewer comments:

Reviewer 1:

1.1: The key point to address is the need for a more nuanced understanding and discussion of microglial subpopulations and heterogeneous microglial cell states. Numerous subpopulations of microglia have been revealed by single cell sequencing work (Hammond et al 2019 Immunity) and the various disease-associated states that microglia can assume, with vastly different effects on myelin in different states. The senior author's own elegant postdoctoral work demonstrating different effects of microglia in trophic "M2" or proinflammatory "M1"-like states on demyelination should be discussed. Also worth discussing is recent work demonstrating the specific activation of white matter microglia after systemic exposure to the chemotherapy drug methotrexate and consequent impairment in oligodendrogenesis and myelination that are rescued with CSF1R inhibitor blockade-mediated microglial depletion (Gibson et al 2019 Cell, Geraghty 2019 Neuron).

The reviewer raises an important point regarding the recent discoveries of the impact of microglia on myelin, and the heterogeneity of microglia in the white matter. As suggested, we have included reference to these studies in the discussion. We also discuss whether distinct microglia states are required to regulate myelin integrity. Please see text excerpts below. Follow-on work to our study will address whether one or more specific microglia states regulate myelin integrity, which first requires microglia heterogeneity in young adult white matter to be characterized.

'We linked structural changes in myelin in the absence of microglia with poor cognitive flexibility, compounded by impaired de novo myelination which normally underpins long-term memory consolidation (32, 52). This builds on previous work revealing that microglia dysregulation (in response to chemotherapy) is sufficient to disrupt myelin structure and cognitive function (10, 34), as we now uncover the requirement for healthy microglia in preventing these pathologies. Given the close relationship between myelin structure and neuronal activity in regulating cognitive function (53-56), our findings also raise the possibility of a role for microglia in influencing adaptive myelination to reinforce cognitive circuits. [...]

'We previously discovered that a subset of microglia expressing a TGFβ superfamily member activin-A regulates remyelination efficiency (22, 72). Taken together with the transcriptomic heterogeneity of microglia during development, homeostasis, demyelination, remyelination, and ageing (57, 72-77), we are now poised to ask whether specific microglia states are required to regulate myelin growth and integrity. This would complement the recent identification of microglia states associated with white matter, with roles in phagocytosis of dying cells in development (77) or myelin debris in ageing (57), and of a shift in functional microglial states underpinning their capacity to support remyelination (72, 73). Whether the decrease of microglia heterogeneity with ageing (74) is associated with a loss in supportive states and progressive myelin pathology needs to be investigated.'

1.2: Regarding the later point, CSF1R inhibitor therapy is in numerous clinical trials for a range of indications. Is there a safe window for microglial depletion to allow a "reset" of microglia in a disease-causing state? Or is there a threshold level of microglia required for maintenance of myelin integrity? This should be discussed.

We thank the reviewer for raising the potential issues surrounding CSF1R inhibitor therapy. We have performed additional experiments to address this point, by administering a CSF1R inhibitor (PLX5622) to wildtype mice for 1 month at different ages in adulthood (new Extended

Data Fig.7). We observed a loss of myelin integrity, the severity of which depended on the age of the mice. We treated 2 month-old mice (as myelination is complete at this age) for 1 month, until 3 months of age, and observed hypermyelination which mirrored the results in the 3 month-old FIRE $\Delta\Delta$ mice (Extended Data Fig.7A-G). Mice treated at 5 months of age (until 6 months) showed patches of demyelination, similar to that observed in 6 month-old FIRE $\Delta\Delta$ mice (Extended Data Fig.7J-L). Altogether, these findings indicate that microglia depletion with CSF1R inhibition for just 1 month is sufficient to disrupt myelin integrity, and this worsens with increasing age.

PLX5622-treated mice had a ~50% depletion of microglia (Extended Data Fig.7B-C); the mirroring of loss of myelin integrity observed in FIRE $\Delta\Delta$ mice suggests that it is the reduction in the number microglia that leads to loss of myelin integrity, more so than dysregulation of remaining microglia. This likely indicates that there is a threshold number of microglia needed for healthy myelin integrity (more than half of baseline levels). ALSP patients also show an incomplete depletion of white matter microglia, indicating the potential translational relevance of this finding (Fig.4A-B). These results have implications for possible off-target effects of CSF1R therapy on myelin health, which we have now referenced in the discussion, as indicated in the excerpt below:

'Our work suggests that a threshold number of microglia is needed to maintain myelin health, as even just a 50-60% decrease in white matter microglia in mouse or human is associated with loss of myelin integrity. Altogether, these findings suggest prudence in the current trialling of CSF1R inhibitors to deplete microglia in cancer or neurological conditions (50, 51), warranting monitoring of potential off-target effects on myelin health.'

1.3: It would add a great deal to the paper (and to future use of this mouse model) to understand at a more granular level what myeloid cell populations are present, they state they are in, and how this changes with age. Single cell RNA-seq would be ideal, but other methods could also shed more light on the myeloid subpopulations that remain and how this evolves. Such characterization of the CNS myeloid subpopulations would be a valuable addition.

We agree with the reviewer that it is useful to the field to understand what myeloid cell populations are present and how they may change in the FIRE $\Delta\Delta$ mice. Previous work in the FIRE $\Delta\Delta$ mice determined that there is no reduction in number of circulating monocytes (Rojo et al., 2019, *Nature Communications*). With respect to other CNS-resident macrophages, perivascular macrophages, meningeal macrophages, and most choroid plexus macrophages are maintained at the ages used in our study and beyond (Kiani-Shabestari et al., 2022, *Cell Reports*; Munro et al., 2020, *Development*). Only a very small number of macrophages floating in the CSF (intracerebroventricular macrophages) are lacking (Munro et al., 2020, *Development*). With relevance to the white matter, we have now confirmed that perivascular macrophage densities are unchanged in FIRE $\Delta\Delta$ mice at the ages investigated in our study: 1 month (Fig.1C-E), 3-4 months and 6 months (new Extended Data Fig.1B-C).

To further assess BAMs in FIRE $\Delta\Delta$ mice, we performed single cell RNA sequencing of FIRE $\Delta\Delta$ and wildtype brains; however, the BAM cluster was too small to reliably indicate any transcriptomic changes, which is a common issue in CNS myeloid cell sequencing studies (although we were able to use this data to assess oligodendrocyte changes; new Fig.5). Instead, we performed flow cytometric analysis of non-microglial myeloid cells (including BAMs) and found that their expression of CSF1R is reduced (new Extended Data Fig.1D-F), in line with FIRE known to regulate expression of CSF1R (Rojo et al., 2019, *Nature Communications*).

1.4: In Figure 3, in addition to the scatterplots, please also express the data in 3E and 3F (inner tongue thickness and myelin thickness) as mean per animal (n = 3-4). Are stats run on a per mouse basis or a per axon basis?

The scatterplots statistics were run on a per axon basis. As suggested, we have now also expressed the data as mean per animal (per mouse basis) (new Extended Data Fig.6D).

We have also done this for all other analyses of the inner tongue and myelin thickness throughout the paper (new Extended Data Figs. 8E,F; 9E-G; 10C-H). Please note that what we find as statistically significant in the scatterplots does not always translate to significance in the binned axon diameter graphs, as the bins are rather arbitrary (1 μm), and myelin changes likely occur more on a sliding scale. However, we still find it helpful to show this data as a distribution according to axon size. It could be that choosing different bin sizes could show significance, eg. as it did for assessing the association between medium-to-large diameter axon hypermyelination and subsequent demyelination (below or above 0.6 μm ; Extended Data Fig. 6L), yet we did not think it was appropriate to choose the bins for the rest of the data with the purpose of observing significance.

Minor comments:

1.5. “Short-term” memory is on the order of seconds to minutes. Barnes maze is more accurately described as testing memory “encoding”, and the wording should reflect that.

We appreciate the reviewer’s suggestion to more accurately describe the cognitive testing and have adjusted the text to ‘memory encoding’ as suggested.

1.6. Line 140 - remove “short term”, more accurate to say “learning and memory encoding”. Here, Geraghty et al 2019 Neuron should also be cited (novel object recognition test performance depends on activity-regulated oligodendrogenesis).

As suggested, we have replaced ‘short term’ with ‘learning and memory encoding’ and have cited Geraghty et al., 2019, Neuron.

1.7. Please state that ALSP leukoencephalopathy results from CSF1R gene mutations (somehow this point was omitted).

We apologize for this omission and have now indicated in the text that ALSP results from heterozygous CSF1R gene mutations (line 220).

1.8. Although beyond the scope of this paper, do the authors have any thoughts as to the mechanisms that may underlie microglial-mediated maintenance of myelin integrity? Just a point for discussion.

As the other reviewers requested a mechanism by which microglia regulate myelin integrity, we have now addressed this in 5 experiments as described below. In summary, we discovered that the loss of myelin integrity in the absence of microglia is associated with the appearance of an oligodendrocyte state with dysregulated lipid metabolism, and is regulated by disruption of the TGF β 1-TGF β R1 axis.

1) **Single cell sequencing reveals FIRE $\Delta\Delta$ mouse-enriched oligodendrocyte state:** To identify cellular and molecular changes underpinning loss of myelin integrity in the absence of microglia, we performed single cell RNA sequencing of FIRE $\Delta\Delta$ and wildtype mice. This revealed an oligodendrocyte state almost exclusively found in the FIRE $\Delta\Delta$ mouse white matter, identified by high expression of *C4b* and *Serpina3n* (new Fig. 5A-G). This is of interest as an oligodendrocyte cluster expressing the same markers appears in other models with myelin pathology (Kenigsbuch et al., 2022, Nature Neuroscience; Shen et al., 2021, Cell Reports; Zhou et al., 2020, Nature Medicine; Falcao et al., 2018, Nat Med). Our results suggest that healthy microglia may normally suppress the appearance of this state. We validated our finding at the protein level and found that SERPINA3n+ Olig2+ cells were present in the white matter tracts of FIRE $\Delta\Delta$ mice, but not in wildtype mice, nor in grey matter of either genotype (new Fig. 5F-G).

2) **Lipidomics demonstrates dysregulated lipid pathway in FIRE $\Delta\Delta$ mice:** Pathway analysis on the FIRE $\Delta\Delta$ mouse-specific oligodendrocyte state revealed dysregulated cholesterol/lipid metabolism-associated pathways (Table removed from figure and now just mentioned in the text). As cholesterol is a major component of myelin which regulates myelin growth, this finding is consistent with the hypermyelination in FIRE $\Delta\Delta$ mice. Interestingly, a previous study identified dysregulation of

genes associated with cholesterol transport in ALSP tissue (*Kemphorne et al., 2020, Acta Neuropathologica Communications*). We validated our finding by lipidomics analysis of FIRE^{Δ/Δ} mouse corpus callosum, which indicated significantly increased cholesterol esters and reduced triglycerides, indicative of excess cholesterol and impaired lipid export, respectively (new Fig.5H).

3) **Dysregulated TGFβ receptor pathway in FIRE^{Δ/Δ} mice:** To determine how the absence of microglia could lead to this dysregulated oligodendrocyte profile, we identified predicted upstream regulators based on the genes/pathways that were significantly regulated in this FIRE^{Δ/Δ} mouse oligodendrocyte state. TGFβ1 was identified as a top predicted upstream regulator (Table was removed from figure and now just mentioned in text), of interest as it is primarily expressed by microglia in the CNS and is known to regulate lipid pathways. TGFβ1 was significantly downregulated in FIRE^{Δ/Δ} mouse white matter (new Fig.6A), and the receptor TGFβR1 was downregulated by oligodendrocyte lineage cells in FIRE^{Δ/Δ} mice (new Fig.6B-D), consistent with TGFβ1 known to regulate the expression of TGFβR1 in other cell types (*Tu et al., 2018, Immunity*).

4) **Conditional knockout of *Tgfb1* in mature oligodendrocytes is sufficient to cause loss of myelin integrity:** As *Tgfb1* knockout in the CNS results in a confounding decrease in microglia number, loss of microglia homeostasis, and monocyte infiltration (*Butovsky et al., 2014, Nature Neuroscience*), we aimed to instead target the receptor on oligodendrocytes. Having observed that TGFβR1 is downregulated on FIRE^{Δ/Δ} mouse oligodendrocytes, we then asked whether this is sufficient to cause loss of myelin integrity. We generated a conditional knockout of *Tgfb1* in mature oligodendrocytes (*Plp-CreERT: Tgfb1^{fl/fl}*), induced recombination by tamoxifen administration from P14 to P18, and assessed the mice at P28. Compared to tamoxifen-treated controls, conditional knockouts had enlarged inner tongues and thicker myelin on smaller and larger diameter axons, respectively, which mirrored the hypermyelination in 1 month-old FIRE^{Δ/Δ} mice (new Fig.6G-I).

5) **Rescue of FIRE^{Δ/Δ} mouse myelin pathology by stimulating TGFβR downstream signalling:** We asked whether reinstating TGFβR1 signalling in oligodendrocytes in FIRE^{Δ/Δ} mice would be sufficient to rescue myelin pathology. In the absence of sufficient expression of the receptor on oligodendrocytes in FIRE^{Δ/Δ} mice, and the inability of TGFβ1 to cross the blood-brain barrier for long-term treatment, we used a small molecule activator of downstream signalling via the Smad2/3 pathway (SRI-011381) which bypasses the need to stimulate TGFβR1. This has been successfully used to preserve myelin in the demyelinating model EAE (*Wu et al., 2021, Theranostics*). We administered SRI-011381 from 2 months of age for 1 month, to assess the impact on the significant myelin pathology observed in FIRE^{Δ/Δ} mice at 3 months of age (Fig.6J). SRI-011381 significantly reversed the myelin pathology in FIRE^{Δ/Δ} mice, reducing the inner tongue enlargement and myelin thickness compared to vehicle control-treated FIRE^{Δ/Δ} mice, such that profiles completely overlapped with those of wildtype controls (new Fig.6K-N).

Altogether, these experiments demonstrate that microglia normally suppress the appearance of an oligodendrocyte state with a dysregulated lipid profile, limiting myelin overgrowth via regulation of TGFβR1 signalling in oligodendrocytes.

Reviewer 2:

2.1 At 1 month of age the number of oligodendrocytes and myelinated axons are not affected, despite the lack of microglia. However, at 3-4 and 6 month of age, the quantification of the number of oligodendrocytes is missing. Could demyelination be due to a loss of oligodendrocytes per se?

We thank the reviewer for this interesting question. We have now assessed oligodendrocyte numbers and proportions at 3-4 months and 6 months of age, and found that oligodendrocytes (CC1+Olig2+) were not affected at either time point (new Extended Data Fig.6E-H). This suggests that the demyelination is not due to loss of oligodendrocytes.

2.2 Mechanistic insight into why lack of microglia leads to decreased myelination is lacking. Do they affect oligodendrocyte survival? Which factors are needed from the microglia to reconstitute the oligodendrocyte function?

As addressed in point 2.1, our data did not point to a mechanism involving oligodendrocyte survival, but rather an impact on oligodendrocyte function. To address this, we identified the mechanism by which microglia impact myelin integrity via 5 new experiments, as described below. In summary, we discovered that the loss of myelin integrity in the absence of microglia is associated with the appearance of an oligodendrocyte state with dysregulated lipid metabolism, and is regulated by disruption of the TGF β 1-TGF β R1 axis.

1) **Single cell sequencing reveals FIRE $\Delta\Delta$ mouse-enriched oligodendrocyte state:** To identify cellular and molecular changes underpinning loss of myelin integrity in the absence of microglia, we performed single cell RNA sequencing of FIRE $\Delta\Delta$ and wildtype mice. This revealed an oligodendrocyte state almost exclusively found in the FIRE $\Delta\Delta$ mouse white matter, identified by high expression of *C4b* and *Serpina3n* (new Fig.5A-G). This is of interest as an oligodendrocyte cluster expressing the same markers appears in other models with myelin pathology (Kenigsbuch et al., 2022, *Nature Neuroscience*; Shen et al., 2021, *Cell Reports*; Zhou et al., 2020, *Nature Medicine*; Falcao et al., 2018, *Nat Med*). Our results suggest that healthy microglia may normally suppress the appearance of this state. We validated our finding at the protein level and found that SERPINA3n+ Olig2+ cells were present in the white matter tracts of FIRE $\Delta\Delta$ mice, but not in wildtype mice, nor in grey matter of either genotype (new Fig.5F-G).

2) **Lipidomics demonstrates dysregulated lipid pathway in FIRE $\Delta\Delta$ mice:** Pathway analysis on the FIRE $\Delta\Delta$ mouse-specific oligodendrocyte state revealed dysregulated cholesterol/lipid metabolism-associated pathways (Table removed from figure and mentioned in text). As cholesterol is a major component of myelin which regulates myelin growth, this finding is consistent with the hypermyelination in FIRE $\Delta\Delta$ mice. Interestingly, a previous study identified dysregulation of genes associated with cholesterol transport in ALS tissue (Kempthorne et al., 2020, *Acta Neuropathologica Communications*). We validated our finding by lipidomics analysis of FIRE $\Delta\Delta$ mouse corpus callosum, which indicated significantly increased cholesterol esters and reduced triglycerides, indicative of excess cholesterol and impaired lipid export, respectively (new Fig.5H).

3) **Dysregulated TGF β receptor pathway in FIRE $\Delta\Delta$ mice:** To determine how the absence of microglia could lead to this dysregulated oligodendrocyte profile, we identified predicted upstream regulators based on the genes/pathways that were significantly regulated in this FIRE $\Delta\Delta$ mouse oligodendrocyte state. TGF β 1 was identified as a top predicted upstream regulator (Table removed from figure and mentioned in text), of interest as it is primarily expressed by microglia in the CNS and is known to regulate lipid pathways. TGF β 1 was significantly downregulated in FIRE $\Delta\Delta$ mouse white matter (new Fig.6A), and the receptor TGF β R1 was downregulated by oligodendrocyte lineage cells in FIRE $\Delta\Delta$ mice (new Fig.6B-D), consistent with TGF β 1 known to regulate the expression of TGF β R1 in other cell types (Tu et al., 2018, *Immunity*).

4) **Conditional knockout of *Tgfb1* in mature oligodendrocytes is sufficient to cause loss of myelin integrity:** As *Tgfb1* knockout in the CNS results in a confounding decrease in microglia number, loss of microglia homeostasis, and monocyte infiltration (Butovsky et al., 2014, *Nature Neuroscience*), we aimed to instead target the receptor on oligodendrocytes.

Having observed that TGF β R1 is downregulated on FIRE $\Delta\Delta$ mouse oligodendrocytes, we then asked whether this is sufficient to cause loss of myelin integrity. We generated a conditional knockout of *Tgfb1* in mature oligodendrocytes (*Plp-CreERT:Tgfb1^{fl/fl}*), induced recombination by tamoxifen administration from P14 to P18, and assessed the mice at P28. Compared to tamoxifen-treated controls, conditional knockouts had enlarged inner tongues and thicker myelin on smaller and larger diameter axons, respectively, which mirrored the hypermyelination in 1 month-old FIRE $\Delta\Delta$ mice (new Fig.6F-I).

5) Rescue of FIRE $\Delta\Delta$ mouse myelin pathology by stimulating TGF β R downstream signalling: We asked whether reinstating TGF β R1 signalling in oligodendrocytes in FIRE $\Delta\Delta$ mice would be sufficient to rescue myelin pathology. In the absence of sufficient expression of the receptor on oligodendrocytes in FIRE $\Delta\Delta$ mice, and the inability of TGF β 1 to cross the blood-brain barrier for long-term treatment, we used a small molecule activator of downstream signalling via the Smad2/3 pathway (SRI-011381) which bypasses the need to stimulate TGF β R1. This has been successfully used to preserve myelin in the demyelinating model EAE (*Wu et al., 2021, Theranostics*). We administered SRI-011381 from 2 months of age for 1 month, to assess the impact on the significant myelin pathology observed in FIRE $\Delta\Delta$ mice at 3 months of age (Fig.6J). SRI-011381 significantly reversed the myelin pathology in FIRE $\Delta\Delta$ mice, reducing the inner tongue enlargement and myelin thickness compared to vehicle control-treated FIRE $\Delta\Delta$ mice, such that profiles completely overlapped with those of wildtype controls (new Fig.6K-N).

Altogether, these experiments demonstrate that microglia normally suppress the appearance of an oligodendrocyte state with a dysregulated lipid profile, limiting myelin overgrowth via regulation of TGF β R1 signalling in oligodendrocytes.

2.3 Is chemical depletion of microglia (e.g. with CSF1R inhibitors), having a similar long-term impact on myelin compaction and myelin degeneration? Alternatively, could reconstitution of microglia rescue these myelin phenotypes?

We thank the reviewer for suggesting these interesting experiments. As reconstitution of microglia in FIRE $\Delta\Delta$ mice is technically challenging due to the mixed background requiring microglia to be isolated from wildtype littermates, and the FIRE $\Delta\Delta$ mouse breeding being very poor, we opted to perform the first requested experiment. We tested the long-term impact of chemical depletion of microglia using a CSF1R inhibitor (PLX5622) on myelin compaction and myelin degeneration (new Extended Data Fig.7A-C). PLX5622 administration to wildtype mice for 1 month recapitulated the loss of myelin integrity in the FIRE $\Delta\Delta$ mice, and the age of the mice dictated the severity of pathology. PLX5622 treatment of 2 month-old mice – when myelination is complete – until 3 months of age caused the same changes in myelin compaction and thickness observed in the 3 month-old FIRE $\Delta\Delta$ mice (new Extended Data Fig.7D-G). When we treated 5 month-old wildtype mice until 6 months of age, this caused patchy demyelination, as observed in 6 month-old FIRE $\Delta\Delta$ mice (new Extended Data Fig.7J-L). Although CSF1R inhibitors also deplete other CNS macrophage populations, the mirroring of results obtained in the FIRE $\Delta\Delta$ mice suggests that the PLX5622 results are likely a consequence of microglia depletion. As we depleted microglia in adulthood after myelination is complete, our results also indicate that microglia regulate myelin maintenance once it is already formed, rather than influencing developmental myelination. Altogether, our findings indicate that microglia number is a critical determinant of myelin health in adulthood.

2.4 For some of the regression analyses on myelin thickness and inner tongue thickness (e.g. 3E&F, 2M), the statistically significant difference seems marginal (i.e. the difference in slope). What's the biological impact of these differences seen in regression? Although the validity of this way of analyzing the data is not questioned, the binned approach as demonstrated in Fig.2F gives a better understanding of where the differences between the genotypes can be observed.

Our observations of changes in slopes for myelin thickness and inner tongue thickness are in line with the magnitude of what is normally observed in the field. Our finding of impaired cognitive flexibility in FIRE $\Delta\Delta$ mice is consistent with previous studies showing that myelin

structural changes of a similar magnitude impact cognitive flexibility (also known as ‘reversal learning’) (Hiramoto et al., 2021, *Mol Psychiatry*; Silva et al., 2019, *Nat Commun*; Yermakov et al., 2019, *Beh Brain Res*; Taib et al., 2017, *PLoS One*; Inagawa et al., 1988, *Behav Neural Biol*). We would like to note that biological impact on neuronal responses is being independently characterized by other researchers in separate studies (unpublished), and we feel it is beyond the scope of the study to duplicate these findings in our paper.

As requested, we have now binned the data as shown in new Extended Data Figs. 3E,F; 6B-C,H; 8E-G; 10C-H; REDACTED. Please note that what we find as statistically significant in the scatterplots does not always translate to significance in the binned axon diameter graphs, as the bins are rather arbitrary (1 μm), and myelin changes likely occur more on a sliding scale. However, we still find it helpful to show this data as a distribution according to axon size. It could be that choosing different bin sizes could show significance, eg. as it did for assessing the association between medium-to-large diameter axon hypermyelination and subsequent demyelination (below or above 0.6 μm ; Extended Data Fig. 6L), yet we did not think it was appropriate to choose the bins for the rest of the data with the purpose of observing significance.

2.5 P.6 lines 149-154: can you be more specific of the impact of this observation? I.e. ‘in a subset of mice perfused for ultrastructural analysis’. How large was this subset in comparison to all mice? What is the percentage increase in number of myelinated axons in the FIRE WT mice 6 weeks after learning? Can you show this in representative images, highlighting unmyelinated axons and myelinated axons, because I cannot observe this difference.

We apologize for the lack of clarity regarding using a subset of mice. Following the behavioural testing, we perfused some of the mice immediately for assessment of new oligodendrogenesis, and 6 weeks later perfused the remainder of the mice for ultrastructural analysis of myelination (following time points of oligodendrogenesis and myelination shown in previous studies of learning and memory). As these assessments require different fixatives and tissue processing, it was not possible to include all the mice for both assessments. Nonetheless, we were still statistically powered to detect differences in myelination with an $n=6$ for FIRE $\Delta\Delta$ mice and $n=4$ for control mice. Therefore, we observed consistent responses in FIRE $\Delta\Delta$ mice for each assay.

As requested, we have now also determined the percentage increase in number of myelinated axons in wildtype and FIRE $\Delta\Delta$ mice 6 weeks after learning (new Extended Data Fig. 5G). We have also highlighted unmyelinated axons in the representative images in Extended Data Fig. 5E.

2.6 Are the trained and untrained mice used for this analysis of the exact same age? The age-range for mice tested for cognition is quite broad, i.e. 2-4M. Can you be more specific on the median age between your FIRE WT and FIRE KO groups (both for untrained and trained) at the day of sacrifice?

We apologize for the lack of clarity on the median ages of the mice. At the time of sacrifice, the median ages were:

- untrained FIRE $^{+/+}$: 118 days old
- untrained FIRE $\Delta\Delta$: 118 days old
- trained FIRE $^{+/+}$: 119 days old
- trained FIRE $\Delta\Delta$: 120 days old.

We have now noted this information in the methods section. We included a range of ages (2-4 months) during which myelin structural integrity is initially affected in FIRE $\Delta\Delta$ mice, without confounding demyelination which we already know would impact cognitive function; demyelination starts at 4.5 months in FIRE $\Delta\Delta$ mice (new Extended Fig. 6I-J). In addition, a range of ages was chosen because FIRE $\Delta\Delta$ mice are poor breeders, and it was unlikely we would ever have enough knockout mice of the same age to use simultaneously for behavioural testing. Rather than perform numerous behavioural assays (with approx. 1-2 knockouts per

litter) repeatedly when the mice were of a specific age, we aimed to perform the behavioural assays in as few batches as possible (2 rounds) to control for testing conditions and to reduce the inherent variability in these experiments as much as possible.

2.7 P.6 lines 152-154: I find this conclusion too strong. The increase in the number of myelinated axons is only observed in a ‘subset of mice’.

We apologize for the lack of clarity. The term ‘subset’ referred to some of the mice used in the behavioural testing being subsequently assessed for myelination, as explained in point 2.5, and not to only some mice showing an increase in myelination. We have removed the term in the text to avoid confusion.

2.8. Your data showing a ‘lack of cognitive flexibility’ is as strong as your data showing enhanced short term spatial memory (extended Fig.4C). Although this finding is mentioned in the text, no further attention is given to this, while the ‘lack of cognitive flexibility’ is stressed.

We thank the reviewer for pointing this out. We would like to clarify that for the 1 h probe test of short-term spatial memory (which we have re-termed ‘memory encoding’ as per Reviewer 1’s suggestion), there was only significance for number of nose pokes in the target hole and not for any other measures. This did not result in significantly more time being spent in the target quadrant. In contrast, during the reversal learning, we saw significant differences in primary errors at multiple time points.

Nonetheless, the 1h probe test result indicates a transient increase in spatial reference memory in the FIRE^{Δ/Δ} mice, which is however then resolved at 3 days and in the reversal phase. This change reflects the abilities of the mice to remember the target quadrant where the escape hole is located, so their spatial reference memory remains intact or even improved initially, but their ability to do this accurately is impaired (i.e. number of errors being increased reflects impaired working memory/ cognitive flexibility). The statistical significance in primary errors in the reversal phase (Extended Data Fig.3I) on reversal days 1 and 2 was more robust, where the error bars are non-overlapping with those from wildtype. Reversal learning is important as this most closely maps to white matter connectivity. We have now adjusted the text to reflect these points, shown in the excerpt below:

‘Second, memory encoding was tested by removing the escape chamber and probing 1 h and 3 days later. The percentage of time spent in the target quadrant (Fig.3C) and the number of nose pokes in and around the target hole were similar between genotypes (Extended Data Fig.5B-C); FIRE^{Δ/Δ} mice even explored the target hole more than FIRE^{+/+} mice during the 1 h probe test (Extended Data Fig.5B), although this did not lead to significance for time spent in the target quadrant (Fig.3C). These findings suggest a transient increase in spatial reference memory only at 1 h, which is then resolved at 3 days; nonetheless, this indicates no memory encoding deficit in FIRE^{Δ/Δ} mice.’

2.9 How sure are you that this reduction in the number of myelinated axons underlies these (perhaps serendipitous) findings in behaviour? Do mice with the lowest number of myelinating axons also perform the worst? Is there a way to confirm these findings, i.e. by reducing or increasing the number of myelinated axons (e.g. chemically) and assess the impact on cognition? To what extent can these cognitive changes truly be linked to impaired myelination?

We thank the reviewer for this interesting question. As requested, we assessed myelinated axon number and cognitive performance, yet found no correlation (new Extended Data Fig.5H). This may reflect that a common threshold number of myelinated axons is needed for cognitive flexibility (i.e. >~1.5 million/mm²), or that the structural integrity of the myelin may be more important than myelinated axon number for performance. Consistent with the latter postulate, previous studies showed that changes in compaction or thickness impact cognitive flexibility (Hiramoto et al., 2021, *Mol Psychiatry*; Silva et al., 2019, *Nat Commun*; Yermakov et al., 2019, *Beh Brain Res*; Taib et al., 2017, *PLoS One*; Inagawa et al., 1988, *Behav Neural*

Bio). In addition, myelinated axon number may be more important for long-term memory consolidation (which we did not test), as previously shown by *Steadman et al., 2020, Neuron*. In this study, as in ours, myelination was measured 6 weeks after the initiation of behavioural training. We consider this an interesting and important question to address in follow-on work.

While we appreciate the suggestion to alter the number of myelinated axons by chemical means to assess impact on cognition, we could not foresee how to achieve this without confounding influences. Previous studies have increased myelination to influence cognition by Clemastine-mediated oligodendrogenesis in contexts where it is impaired (e.g. ageing, Alzheimer's disease model; *Wang et al., 2020, Nature Neuroscience*; *Chen et al., 2021, Neuron*). However, we observed no impairment in oligodendrogenesis during learning in FIRE $\Delta\Delta$ mice (**Extended Data Fig.5A-D**). Additionally, recent work has revealed that Clemastine can surprisingly induce oligodendrogenesis while reducing myelination (*Palma et al, 2022, Front Cell Dev Biol*) and either increasing or decreasing myelin thickness (*Palma et al, 2022, Front Cell Dev Biol*; *Lee et al., 2021, Scientific Reports*), all of which could confound interpretation of results. Clemastine may also decrease a microglia subpopulation (Cd11c+) in our wildtype controls (*Palma et al, 2022, Front Cell Dev Biol*) which has been associated with developmental white matter (*Wlodarczyk et al, 2017, EMBO J*). Conversely, decreasing the number of myelinated axons would be challenging to achieve without targeting oligodendrogenesis or inducing demyelination, both of which would alter the microenvironment independently of changes in myelination.

2.10 The lack of microglia may also impact neuronal function and cognition independently of the effect on myelin/oligodendrocytes.

We agree with the reviewer that there is the possibility that cognition is also affected by the impact on other cell types including neurons. We attempted to investigate potential neuronal changes in FIRE $\Delta\Delta$ mice by single cell RNA sequencing of all brain cells, yet neuronal clusters were not well represented thereby preventing further analysis. Nonetheless, previous microarray analysis (*Rojo et al., 2019, Nature communications*) and follow-up bulk RNA sequencing (unpublished) of FIRE $\Delta\Delta$ mouse brain have not identified significant changes in genes related to neurons, axons, or synaptic function. Other researchers have been independently investigating if there are changes in FIRE $\Delta\Delta$ mouse neurons, synapses, and electrophysiology, and 2 manuscripts are in preparation for submission; we were not involved in these studies and would consider it outside the scope of our manuscript to duplicate this work here. We have however indicated the potential impact on neurons in discussion, as indicated in the excerpt below:

'Given the close relationship between myelin structure and neuronal activity in regulating cognitive function (48-51), our findings also raise the possibility of a role for microglia in influencing adaptive myelination to reinforce cognitive circuits. However, understanding the impact of the absence of microglia on neuronal activity and synaptic health requires further investigation.'

2.10 Fig.5D, E: Can you explain why ALSP patients have much thicker axons than healthy controls? Can this skew your observation on myelin thickness?

We thank the reviewer for highlighting this interesting observation. The larger axon sizes in ALSP are a result of axonal swellings, which has been previously documented to occur in myelinated axons (*Oyanagi et al., 2017, Brain Pathology*; *Lin et al., 2010, Int J Clin Exp Pathol*; *Yazawa et al., 1997, J Neurol Sci*; *Kinoshita et al., 2012, J Neurol Sci*). Indeed, axonal spheroids are a common pathological feature of this disorder, as it is in the name 'Adult onset leukoencephalopathy with axonal spheroids and pigmented glia'. We have now indicated the reason for larger axons in ALSP in the text, as indicated in the excerpt below:

'Larger axon diameters were noted in ALSP versus controls, consistent with axonal swelling being a typical pathological feature of this disorder (Fig.5D,E; Extended Data Fig.9E), and yet myelin was still thicker than would be expected of these axon diameters.'

2.11 Regarding the axonal spheroids: How often are these present (e.g. in every mouse Are these axonal spheroids never observed in WT mice? How often are they observed in the FIRE KO mice? I.e. which percentage of axons have these spheroids?)

We only observed the axonal spheroids occasionally in FIRE Δ/Δ mice and never in the wildtype mice. In FIRE Δ/Δ mice, we observed that <0.1% of axons have spheroids at 3+ months of age, and not in every mouse. We have now indicated this in the text.

Minor comments:

2.12 I am missing the figure legends on the extended figures, making it hard to interpret some.

The figure legends for the extended figures were/are included below the figures in the Extended Data PDF file.

2.13 It would be easier for comparison purpose if all plots on myelin thickness were on the same scale (Y-axis). Although linear regression is the correct statistical method for analyzing the data, it is difficult to get a feeling for the percentage of difference in myelin thickness across age and genotype. Is it possible to visualize this in this binned analysis similar to what is shown for the inner tongue thickness? Perhaps all ages and genotypes can be combined in 1 graph, just for visualization purposes (in appendix) and to make the reader understand the severity of the increased myelin thickness. I would however, suggest to perform a combined analysis of your regression slope s across the different ages and genotypes, to reduce the chance of a type I error.

As suggested we have provided the following (new Ext Data Fig.3):

- i) same scale for all plots of myelin thickness (up to 1 μm) and inner tongue thickness (up to 0.6 μm) throughout the paper.
- ii) binned analysis for myelin thickness and inner tongue thickness.
- ii) a combined data graph with all ages and genotypes for both myelin thickness and inner tongue thickness.

With regards to the combined statistical analyses, our statistical software would not allow us to perform regression analyses on 6 slopes (2 genotypes x 3 ages). However, other myelin studies have performed similar analyses combining several groups using an ANOVA. We therefore performed a one way ANOVA on the combined analyses of myelin thickness and inner tongue thickness and have provided the statistics in tables (new Extended Data Fig.3C,D).

2.14 ig.1H and others: you perform a 2x2 ANOVA on Olig2+CC1- vs Olig2+CC1+ cells. However, the total of Olig2+CC1 plus Olig2+CC1+ is set at 100%. That means that if the proportion of CC1- cells drops, your proportion of CC1+ cells automatically increases, making these observations (CC1+ vs CC1-) dependent on each other. In 2x2 ANOVA you are assessing the impact of 2 independent factors. Thus, here it would suffice to perform a 1x ANOVA on e.g. only the CC1+ cells and compare this between the WT and KO mice. Please also revise this for all of the other graphs in this manuscript where you use similar sets of data (e.g. 2C, 2K etc)

We thank the reviewer for this helpful suggestion. We have now performed a one way ANOVA on the CC1+ positive cells between the WT and KO mice, and similarly did not observe any statistical significance (P=0.9472). We have also done the same for other graphs in the manuscript (Fig.2C, 2K, REDACTED, 3D, 6E; Ext.Data Fig.6D, 6E, 7I).

Reviewer 3

3.1 The authors contributed all myelin-related phenotypic observations to disruption of myelin integrity. This is problematic. The phenotypes are two folds, excessive myelin sheath growth that leads to hypermyelination (P25-30 and 3-4 months) and loss of myelin integrity (6 months of age). First, in mice, myelination peaks around P30, but is not complete until ~P60 (Bercury K, Macklin WB Dynamics and mechanisms of CNS myelination. Dev Cell 32:447–458, 2015). There were striking phenotypes—outfolds, unraveling, impaired compaction of myelin, and increased myelin thickness (fig. 2A-H)—at P25-30, a timepoint when developmental myelin growth is occurring. Secondly, the growth of the myelin sheath in vivo was associated with a larger inner tongue when compared to later stages of development (Snaidero et al. Cell 2014). The documented inner tongue phenotype likely indicates excessive myelin sheath growth, which is supported by the observation of increased myelin thickness (Fig. 2N and O; extended data Fig 3). Thirdly, absence of microglia prevents the increase in myelination that normally occurs with consolidation of new spatial information (Fig. 3K-L).

We agree with the reviewer that our observations of myelin changes in the FIRE^{ΔΔ} mouse are threefold, first with hypermyelination, second with demyelination, and third with impaired myelination that normally occurs with spatial memory consolidation. We have now revised our title, abstract, results, and discussion to reflect this, referencing the changes in myelin structure as hypermyelination, and the loss of myelin integrity associated with demyelination.

3.2 For the reason discussed above, the statement that “microglia are dispensable for developmental myelin formation” is not fully supported.

We apologize that the phrasing of our statement was unclear; we were referring to microglia being dispensable with regards to myelin being initially formed in FIRE^{ΔΔ} mice, with no change in myelin protein expression or number of myelinated axons (Fig.1; Extended Data Fig.2). However, we understand the reviewer's comment that we assessed the initial impact of the absence of microglia in FIRE^{ΔΔ} mice while developmental myelination was still occurring (before P60), raising the question as to whether microglia impact the structural properties of myelin once it has already formed.

To address this, we depleted microglia after P60 and assessed the impact on myelin. Although the deletion of FIRE prevents microglia colonization of the CNS in early development, it is unknown if conditional FIRE deletion in existing microglia would lead to their depletion, and no FIRE conditional mouse exists. Therefore, we used a widely adopted strategy of depleting microglia in adulthood using CSF1R inhibitor (PLX5622) administration (new Extended Data Fig.7A-C). Treating P60 mice with PLX5622 for 1 month (to 3 months of age) led to the same hypermyelination profile observed in 3 month-old FIRE^{ΔΔ} mice: enlarged inner tongues on small diameter axons, and thicker myelin on large diameter axons (Extended Data Fig.7D-G). Treating 5 month-old mice with PLX5622 for 1 month (to 6 months of age) led to patchy demyelination, as observed in 6 month-old FIRE^{ΔΔ} mice (Extended Data Fig.7J-L). The mirroring of results when microglia were depleted after P60 to those in the FIRE^{ΔΔ} mice suggests that microglia regulate myelin maintenance after developmental myelination is complete.

3.3 The study is largely descriptive and lacks mechanistic exploration. Literature suggested that microglial sheath pruning corrects myelin targeting and refines myelination by oligodendrocytes, which accounts why depleting microglia leads to increased myelin sheath growth (Hughes and Appel. Nat Neurosci 2020). It would be desirable to present data to either concur such mechanism or support a new one.

As requested, we have now identified new cellular and molecular mechanisms by which microglia regulate myelin maintenance and integrity, via the 5 experiments described below. In summary, we discovered that the myelin pathology in the absence of microglia leads to the appearance of an oligodendrocyte state with dysregulated lipid metabolism, and is caused by disruption of the TGFβ1-TGFβR1 axis. This is distinct from Hughes and Appel's discovery that

microglia phagocytosis of myelin sheaths regulates sheath number in early development, as we now reveal that microglia prevent excessive growth of existing sheaths.

1) **Single cell sequencing reveals FIRE^{ΔΔ} mouse-enriched oligodendrocyte state:** To identify cellular and molecular changes underpinning loss of myelin integrity in the absence of microglia, we performed single cell RNA sequencing of FIRE^{ΔΔ} and wildtype mice. This revealed an oligodendrocyte state almost exclusively found in the FIRE^{ΔΔ} mouse white matter, identified by high expression of *C4b* and *Serpina3n* (new Fig.5A-G). This is of interest as an oligodendrocyte cluster expressing the same markers appears in other models with myelin pathology (*Kenigsbuch et al., 2022, Nature Neuroscience; Shen et al., 2021, Cell Reports; Zhou et al., 2020, Nature Medicine; Falcao et al., 2018, Nat Med*). Our results suggest that healthy microglia may normally suppress the appearance of this state. We validated our finding at the protein level and found that SERPINA3n+ Olig2+ cells were present in the white matter tracts of FIRE^{ΔΔ} mice, but not in wildtype mice, nor in grey matter of either genotype (new Fig.5F-G).

2) **Lipidomics demonstrates dysregulated lipid pathway in FIRE^{ΔΔ} mice:** Pathway analysis on the FIRE^{ΔΔ} mouse-specific oligodendrocyte state revealed dysregulated cholesterol/lipid metabolism-associated pathways (new Fig.5J-K) (Table removed from figure and mentioned in text). As cholesterol is a major component of myelin which regulates myelin growth, this finding is consistent with the hypermyelination in FIRE^{ΔΔ} mice. Interestingly, a previous study identified dysregulation of genes associated with cholesterol transport in ALSP tissue (*Kemphorne et al., 2020, Acta Neuropathologica Communications*). We validated our finding by lipidomics analysis of FIRE^{ΔΔ} mouse corpus callosum, which indicated significantly increased cholesterol esters and reduced triglycerides, indicative of excess cholesterol and impaired lipid export, respectively (new Fig.5H).

3) **Dysregulated TGFβ receptor pathway in FIRE^{ΔΔ} mice:** To determine how the absence of microglia could lead to this dysregulated oligodendrocyte profile, we identified predicted upstream regulators based on the genes/pathways that were significantly regulated in this FIRE^{ΔΔ} mouse oligodendrocyte state. TGFβ1 was identified as a top predicted upstream regulator (Table removed from figure and mentioned in text), of interest as it is primarily expressed by microglia in the CNS and is known to regulate lipid pathways. TGFβ1 was significantly downregulated in FIRE^{ΔΔ} mouse white matter (new Fig.6A), and the receptor TGFβR1 was downregulated by oligodendrocyte lineage cells in FIRE^{ΔΔ} mice (new Fig.6B-D), consistent with TGFβ1 known to regulate the expression of TGFβR1 in other cell types (*Tu et al., 2018, Immunity*).

4) **Conditional knockout of *Tgfb1* in mature oligodendrocytes is sufficient to cause loss of myelin integrity:** As *Tgfb1* knockout in the CNS results in a confounding decrease in microglia number, loss of microglia homeostasis, and monocyte infiltration (*Butovsky et al., 2014, Nature Neuroscience*), we aimed to instead target the receptor on oligodendrocytes. Having observed that TGFβR1 is downregulated on FIRE^{ΔΔ} mouse oligodendrocytes, we then asked whether this is sufficient to cause loss of myelin integrity. We generated a conditional knockout of *Tgfb1* in mature oligodendrocytes (*Plp-CreERT: Tgfb1^{fl/fl}*), induced recombination by tamoxifen administration from P14 to P18, and assessed the mice at P28. Compared to tamoxifen-treated controls, conditional knockouts had enlarged inner tongues and thicker myelin on smaller and larger diameter axons, respectively, which mirrored the hypermyelination in 1 month-old FIRE^{ΔΔ} mice (new Fig.6F-I).

5) **Rescue of FIRE^{ΔΔ} mouse myelin pathology by stimulating TGFβR downstream signalling:** We asked whether reinstating TGFβR1 signalling in oligodendrocytes in FIRE^{ΔΔ} mice would be sufficient to rescue myelin pathology. In the absence of sufficient expression of the receptor on oligodendrocytes in FIRE^{ΔΔ} mice, and the inability of TGFβ1 to cross the blood-brain barrier for long-term treatment, we used a small molecule activator of downstream signalling via the Smad2/3 pathway (SRI-011381) which bypasses the need to stimulate TGFβR1. This has been successfully used to preserve myelin in the demyelinating model EAE (*Wu et al., 2021, Theranostics*). We administered SRI-011381 from 2 months of age for 1 month, to assess the impact on the significant myelin pathology observed in FIRE^{ΔΔ} mice at 3 months of age (Fig.6J). SRI-011381 significantly reversed the myelin pathology in FIRE^{ΔΔ} mice, reducing the inner tongue enlargement and myelin thickness compared to vehicle

control-treated FIRE Δ/Δ mice, such that profiles completely overlapped with those of wildtype controls (new Fig.6K-N).

Altogether, these experiments demonstrate that microglia normally suppress the appearance of an oligodendrocyte state with a dysregulated lipid profile, limiting myelin overgrowth via regulation of TGF β R1 signalling in oligodendrocytes.

3.4 To conclusively evaluate myelin maintenance, it would be desirable to delete FIRE at a time when the active growth phase of myelin has passed, which would be P60 in mice (Bercury K, Macklin WB Dynamics and mechanisms of CNS myelination. Dev Cell 32:447–458, 2015).

We thank the reviewer for this helpful suggestion. We have addressed this with a new experiment; please see point 3.2 for details on how microglia depletion beyond P60 mimicked the effects on myelin seen in the FIRE Δ/Δ mice (new Extended Data Fig.7).

3.5 Activating mTOR/AKT pathway leads to CNS hypermyelination (Snaidero et al Cell 2014; Flores et al J Neurosci 2008; Narayanan...Macklin. J Neurosci 2009). Does depleting microglia results in increased p-mTOR, p-P70S6K, p-S6RP? Chronically inhibiting AKT signaling by rapamycin treatment corrects the hypermyelination phenotype in mice that express constitutively active Akt (Narayanan et al J Neurosci 2009). Could rapamycin treatment ameliorate FIRE Δ/Δ hypermyelination phenotype?

As requested, we investigated mTOR/AKT pathway activation in FIRE Δ/Δ mice during the hypermyelination phase by assessing p-S6RP, yet observed variable results which did not reach significance (see data below). In addition, mTOR/AKT signalling did not come up in the pathway analysis of FIRE Δ/Δ mouse oligodendrocytes. For these reasons, we did not proceed with rapamycin treatment. Rather, we discovered that the hypermyelination phenotype in FIRE Δ/Δ mice results from dysregulated TGF β R1 signalling in oligodendrocytes. We reproduced the hypermyelination profile in a conditional knockout of *Tgfb1* in mature oligodendrocytes, and rescued the hypermyelination in FIRE Δ/Δ mice by stimulating the pathway downstream of TGF β R1. Please refer to point 3.3. for details.

3.6 Hypermyelination could lead to demyelination in PNS (Adlkofer et al Nat Genetics 1995) and CNS (Hu et al BioRxiv 2020, <https://doi.org/10.1101/2020.11.10.377226>). Was the myelin degeneration here due to hypermyelination or other mechanism?

We thank the reviewer for raising this interesting point. Our previous and new experiments altogether suggest that hypermyelination may precede demyelination.

First, we had seen that the hypermyelination (1-3 months) precedes the demyelination in FIRE Δ/Δ mice (6 months; Fig.3).

Second, to identify which axons demyelinate first, we provide new data documenting when demyelination is initiated in FIRE Δ/Δ mice, which is at 4.5 months of age. At this age, we found that the average unmyelinated axon diameter was 0.73 μ m; we also know that medium-to-large diameter axons (>0.6 μ m) are the ones that have the most robust hypermyelination at

3-4 months of age (new Extended Data Fig.6I-L). Please note that unmyelinated axons of a medium-to-large size were so rarely observed in the wildtype mice at 4.5 months of age that we did not quantify the size of these axons. These data suggest that hypermyelination could lead to demyelination.

Third, assessment of the human disorder ALSP indicated extremely thick myelin sheaths which were starting to unravel (Fig.4).

We have indicated these findings in the results and discussion, as shown in the excerpts below:

Results:

'We found that demyelination was initiated as early as 4.5 months of age in FIRE^{Δ/Δ} mice (Extended Data Fig.7F-G); unmyelinated axons were of a medium-to-large calibre (average $0.73 \pm 0.1 \mu\text{m}$) (Extended Data Fig.7I), indicating that these axons underwent demyelination first. As medium-to-large diameter axons were the ones that showed hypermyelination just prior to demyelination, at 3-4 months of age (Extended Data Fig.7H), this suggests that hypermyelination may precede demyelination.'

Discussion:

'Our data suggests that hypermyelination may precede demyelination, raising the question as to whether this sequence of events underpins myelin damage in ageing and neurodegenerative disease.'

3.7 The authors showed severe demyelination phenotype at 6 months of age. It would be informative to know when the demyelination starts to occur. In addition to TEM data, MBP western blot of the corpus callosum would be informative.

We thank the reviewer for this interesting question. Our initial submission indicated that demyelination starts to occur in FIRE^{Δ/Δ} mice between 3-4 months of age (no demyelination) and 6 months of age (demyelination apparent). As indicated in point 3.6, we now include data demonstrating mild demyelination as early as 4.5 months of age in the FIRE^{Δ/Δ} mice, as indicated by reduced numbers of myelinated axons (new Extended Data Fig.6I-J).

With regards to Western blots for MBP, we would not expect to see differences as the focal and patchy demyelination in the FIRE^{Δ/Δ} mice occurs in small areas which are not noticeable even using the very sensitive method of confocal imaging (new Extended Data Fig.6A). This suggests limited sensitivity to detect the demyelination using methods other than electron microscopy.

3.8 The overall statistical tests are appropriate. However, it would be desirable to provide actual P value instead of "non-significant" for Fig 1K and to present quantitative data and statistical analysis on the fluorescent intensity for Fig. 1I, Extended data Fig 2 and Extended data Fig 4A.

As requested, we have now provided the p-value for Fig1K and for all statistical comparisons throughout the paper. Additionally, we have provided quantification and statistical analysis of fluorescence intensity for MBP and other myelin proteins (Extended Data Fig.2A,C,D) and neurofilament (Extended Data Fig.2G).

Reviewer Reports on the First Revision:

Referees' comments:

Referee #1 (Remarks to the Author):

The authors are to be congratulated on the great amount of new data produced during the revision period, which has substantially improved this manuscript. My comments have been thoroughly addressed.

It is an excellent and important paper that provides several key new insights.

Referee #2 (Remarks to the Author):

The new data are very exciting and have brought this paper really to a new level. The identification of (the lack of) TGF β 1 as one of the drivers of the myelin phenotype is very interesting.

Said that there are some things that need to be fixed before this is published.

1. Especially the new figure 6, although very interesting indicating the presence of a unique olig cluster1 present in the fire delta/delta and not in the wild type, is not very clear. The usefulness of panel A is unclear to me, there is also no figure legend for the colours and I am not sure what exactly is demonstrated with this panel. It would be much more useful to illustrate how the single nuclei data set in extended fig 10 was reduced to the oligodendrocyte data set shown here. In addition, the figure legend to extended fig 10 is much too limited. The authors should provide a full explanation on how these experiments were performed.
2. The lipid changes documented in panel 6L are potentially very interesting but the number of mice is small and before believing what is measured here a statistical analysis is needed (with multiple testing correction).
3. The finding that TGF β 1 drives this phenotype is very interesting. However, does this translate to human microglia? Is TGF β 1 expressed by human microglia? (for instance in the samples of the ALS patients)
4. The authors might be interested in (and cite) a publication of Shen et al (Cell 2020) where the authors demonstrated an interesting induction of an "olig" response in AD mouse model which was opposite to a microglia/astrocyte response in this mice, very much in line with their proposals regarding the relevance of their work to AD.

Referee #3 (Remarks to the Author):

The authors have adequately addressed my concerns. As such, the revised manuscript provided novel mechanism underlying how microglia regulate myelin growth and integrity in both mice and humans. The findings will have a broader impact in the field.

Author Rebuttals to First Revision:

We appreciate the positive comments from all 3 reviewers that our revised manuscript provides 'a great amount of new data', is 'an excellent and important paper', 'the new data are very exciting', and the 'findings will have a broader impact in the field'. As reviewers 1 and 3 had no further suggestions, we have addressed reviewer 2's suggestions in a point-by-point response below and have highlighted the changes in the text in yellow in both the main text and extended data file to facilitate review.

Reviewer 2:

2. 1 Especially the new figure 5, although very interesting indicating the presence of an unique olig cluster1 present in the fire delta/delta and not in the wild type, is not very clear. The usefulness of panel A is unclear to me, there is also no figure legend for the colours and I am not sure what exactly is demonstrated with this panel. It would be much more useful to illustrate how the single nuclei data set in extended fig 10 was reduced to the oligodendrocyte data set shown here. In addition, the figure legend to extended fig 10 is much too limited. The authors should provide a full explanation on how these experiments were performed.

We thank the reviewer for these helpful suggestions. We agree that the panel 5A (and B) were not useful so we have removed these. Rather we have expanded Extended Data Figure 9 as suggested, to better illustrate how the oligodendrocyte subsetting was performed on the single cell RNA sequencing data. We now include the tSNE plots showing all the cell populations identified prior to oligodendrocyte subsetting, and have indicated that the oligodendrocyte cluster (identified by expression of myelin genes *Plp*, *Mag*, *Mog*, and *Mbp*) does not express markers associated with other cell type clusters (*Fgfr3* for astrocytes, *Cd68* for microglia/macrophages, *Pdgfra* for OPCs, and *Gpr17* for committed oligodendrocyte precursors) (new Extended Data Fig. 9A-D).

With regards to additional explanation as to how these experiments were performed, we now refer to the methods section in the figure legend, given the lengthy explanation required. The bioinformatics section from the methods is provided below for ease of reference. Please note that the full details to replicate the analysis pipelines can be found in code scripts available on Github, with the link provided below (highlighted) and in the methods.

*'Alignment to the reference genome, feature counting and cell calling was performed following the 10X Genomics Cell Ranger (v5.0.0) pipeline, using the default mm10 genome supplied by 10X Genomics (refdata-gex-mm10-2020-A). From the output, the filtered matrices were used for downstream analyses. Pre-processing was performed on the University of Edinburgh's compute cluster Eddie. The analysis was performed with R version 4.1.1. Full details to replicate the analysis pipelines described below can be found in code scripts available on GitHub (<https://github.com/Anna-Williams/Veronique-Firemice>). Cells were filtered using dataset-specific parameters on the basis of genes and UMIs per cell, the ratio between these two parameters, and percentage of mitochondrial gene reads per cell. Thresholds were computed with the *isOutlier* function from *scater*(v1.20.1) as batch 6 was of poorer quality than the other batches, with outlier values, the *subset* argument was used. Only genes that were detected in at least two cells were kept. With *scran* (v1.20.1), the data were normalised by deconvolution and the top 15% highly variable genes were selected. Following principal component*

analysis (PCA), 25 principal components (PCs) were kept for downstream analysis (cut-off selected by examination of an Elbowplot). Non-linear dimensional representation (UMAP) and t-SNE and gene expression variance explained by batch (computed with scater) revealed the need for batch correction. Batch correction was performed by mutual nearest neighbours with fastMNN, batchelor (v1.8.0). Finally, a graph-based clustering approach was used to cluster the cells using the clusterCells function from scran, with $k = 60$. Clusters with the highest expression of oligodendrocyte markers (Plp1, Mog, Mag and Mbp) and that did not express other cell type markers (for example, astrocyte, OPC or microglia markers) were subsetting to be analysed separately. Cell and gene quality control were further adjusted setting a stricter minimum UMI count threshold (5000 UMIs) and maximum percentage of mitochondrial gene reads per cell (10%). A small cluster of cells of lower quality based on the new thresholds was also excluded from the analysis (Extended Data Fig.10). Ultimately, we included a total of 19506 genes and 13583 cells. The normalisation, feature selection, dimensional reduction and batch correction were repeated with the subset dataset as described above. Clustering was performed at different resolutions, and after examination with clustree (v0.4.4), $k = 100$ was selected, and then merged into four clusters (Fig.6A, Extended Data Fig.10). Differential gene expression between cluster 1 (specific to the FIRE Δ/Δ mice) and the mean expression of all other cells was performed with FindMarkers from Seurat (v4.1.0)(Extended Data Sheet 1).

2. The lipid changes documented in panel 5H are potentially very interesting but the number of mice is small and before believing what is measured here a statistical analysis is needed (with multiple testing correction).

As requested, we have now performed statistical analysis on the lipidomics data. As the data is presented as Log2 fold change, it was not possible to perform an ANOVA (as we have one 'group'). Instead, we have performed a one-sample t-test against a value of 0 (a value which would indicate no change), for both the individual desaturation classes as well as for the total lipid class (i.e. all desaturation classes combined for a given lipid class Log2 fold change). This has indicated significantly different regulation of several lipid classes, including the cholesterol esters and triglycerides which have relevance for lipid accumulation and transport, respectively, consistent with our findings of hypermyelination in the FIRE Δ/Δ mice.

3. The finding that TGF β 1 drives this phenotype is very interesting. However, does this translate to human microglia? Is TGF β 1 expressed by human microglia? (for instance in the samples of the ALSP patients).

We agree with the reviewer that these are interesting questions. Previous work has demonstrated that human microglia express TGF β 1 at the RNA level (Galatro et al., 2017, Nature Neuroscience PMID: 28671693; www.brainrnaseq.org) and protein level (Bottcher et al., 2019, Nature Neuroscience PMID: 30559476), which we have now referenced in the text as follows:

'To determine how the absence of microglia could contribute to these findings, we assessed predicted upstream regulators of Oligo1 genes and identified TGF β 1 as a prime candidate (REDACTED; Extended Data Sheet 2), as it is predominantly expressed by microglia in both mouse and human brain (www.brainrnaseq.org) (43, 44), and is known to influence lipid metabolism (45).'

With regards to ALSP, microglia have never been specifically assessed. As with many secreted proteins, the mature protein of TGF β 1 is challenging to detect by immunostaining

and we had most success in mouse by using ELISA with fresh tissue, which unfortunately is not available for ALSP brain due to it being a rare disease. However, 2 studies have previously assessed transcriptomics of whole white matter areas and their findings regarding TGF β regulation are in line with our hypothesis. More specifically:

- The TGF β pathway is significantly dysregulated in ALSP white matter, but not in grey matter (Kempthorne et al., 2020, *Acta Neuropathol Commun*, PMID32430064)
- *TGFB1* is upregulated in a white matter tract which is preserved in ALSP (Berdowski et al., 2022, *Acta Neuropathol*, PMID: 35713703), which may indicate a protective compensatory mechanism.

We have now referenced these studies in the text as follows:

'Notably, transcriptomic analyses of ALSP brain has indicated a dysregulation of the TGF β pathway specifically in the white matter (40), and an upregulation of TGFB1 in the least affected white matter region (74).'

4. The authors might be interested in (and cite) a publication of Chen et al (Cell 2020) where the authors demonstrated an interesting induction of an "olig" response in AD mouse model which was opposite to a microglia/astrocyte response in this mice, very much in line with their proposals regarding the relevance of their work to AD.

We thank the reviewer for referring to this interesting paper. We have now cited it in the discussion as follows:

'In addition, our work has important implications for understanding cellular networks contributing to cognitive decline with ageing – where there is prominent hypermyelination (2, 4, 24), demyelination and impaired new myelination (6, 54), alongside microglia dysfunction (43, 59-61). This may also be relevant to dementia-associated neurodegenerative disease, given that in a mouse model of Alzheimer's disease, there are opposite changes in gene modules primarily associated with microglia (and astrocytes) vs. those related to oligodendrocytes and myelination (62). This study also identified an initial upregulation of the oligodendrocyte-associated module which was later downregulated in microenvironments with highest A β accumulation; whether this represents initial hypermyelination followed by demyelination remains to be determined.'

Reviewer Reports on the Second Revision:

Referees' comments:

Referee #2 (Remarks to the Author):

The authors addressed all my remaining issues, congratulations with a great paper